# The Expert Strikes Back: Interpreting Mixture-of-Experts Language Models at Expert Level

Jeremy Herbst [1]   Stefan Wermter [1]   Jae Hee Lee [1]

## Abstract

Mixture-of-Experts (MoE) architectures have become the dominant choice for scaling Large Language Models (LLMs), activating only a subset of parameters per token. While MoE architectures are primarily adopted for computational efficiency, it remains an open question whether their sparsity makes them inherently easier to interpret than dense feed-forward networks (FFNs). We compare MoE experts and dense FFNs using $k$-sparse probing and find that expert neurons are consistently less polysemantic, with the gap widening as routing becomes sparser. This suggests that sparsity pressures both individual neurons and entire experts toward monosemanticity. Leveraging this finding, we *zoom out* from the neuron to the expert level as a more effective unit of analysis. We validate this approach by automatically interpreting hundreds of experts. This analysis allows us to resolve the debate on specialization: experts are neither broad domain specialists (e.g., biology) nor simple token-level processors. Instead, they function as fine-grained task experts, specializing in linguistic operations or semantic tasks (e.g., closing brackets in LaTeX). Our findings suggest that MoEs are inherently interpretable at the expert level, providing a clearer path toward large-scale model interpretability. Code is available at: https://github.com/jerryy33/MoE_analysis.

## 1. Introduction

Mixture-of-Experts (MoE) architectures have emerged as the most efficient choice for scaling large language models (LLMs), demonstrating state-of-the-art performance across numerous benchmarks (Comanici et al., 2025; Yang et al., 2025a; Team et al., 2025; Li et al., 2025; Zeng et al., 2025; Liu et al., 2024). By activating only a fraction of their total parameters for any given token, MoEs achieve the performance of large models with the inference cost of smaller ones. However, while these models continue to increase in complexity, interpretability research has struggled to keep pace. Understanding how these models represent and process information is essential for debugging failures, ensuring alignment, and building trust in high-stakes deployments.

Interpreting LLMs is primarily challenged by *polysemanticity*, the phenomenon where individual units (neurons) activate for multiple, unrelated concepts. This is driven by superposition, a mechanism where networks represent more concepts than they have dimensions by storing them in nearly orthogonal directions (Elhage et al., 2022). While recent work has made strides in disentangling these representations using sparse coding (Bricken et al., 2023; Dunefsky et al., 2024), these methods require massive compute budgets to interpret every layer.

MoE models offer a promising, yet under-explored, alternative. Recent experiments on toy models suggest that increased sparse routing, defined as the fraction of experts active per token, can reduce superposition (Chaudhari et al., 2025). If this trend holds in large-scale models, it would suggest a useful synergy: the architectural sparsity that drives performance scaling (He, 2024; Team et al., 2025; Zhao et al., 2025) may also make these models more interpretable by design. This would reduce reliance on expensive post-hoc concept extraction. However, whether MoE experts in production-scale models are truly more interpretable than dense feed-forward networks (FFNs) remains an open empirical question.

In this work, we investigate this question across a wide range of models. Using probing (Alain & Bengio, 2016), we demonstrate that MoE experts exhibit reduced polysemanticity compared to dense FFNs. By comparing models we find that this is better explained by architectural sparsity than by total parameter count alone. Critically, we show that this gap widens as routing becomes sparser, suggesting that experts in models with very sparse routing approach a state

[1]Department of Informatics, University of Hamburg, Hamburg, Germany. Correspondence to: Jeremy Herbst <jeremy.herbst111@gmail.com>, Jae Hee Lee <jae.hee.lee@uni-hamburg.de>.

of monosemanticity.

Based on these findings, we *zoom out* from the individual neuron to the entire expert as the primary unit of analysis. This allows for the automatic interpretation of model components without the need for additional trained models like sparse autoencoders (Bricken et al., 2023). We validate this approach through automatic labeling and scoring of hundreds of experts, and provide strong attributional evidence supporting the generated labels.

Our analysis helps resolve the ongoing debate regarding expert specialization. While some work suggests experts specialize in broad domains (e.g., biology or coding) (Muennighoff et al., 2025; Liu et al., 2024; Dai et al., 2024), other work suggests experts respond primarily to token-level or syntactic features (Xue et al., 2024; Jiang et al., 2024). We find that both views are incomplete: experts often behave like fine-grained task specialists. An expert may be domain-restricted (e.g., LATEX), but its role is better described as a concrete computational operation (e.g., closing brackets in LATEX) rather than representing the domain as a whole.

In summary, our contributions are:

- We show that MoE experts' neurons are consistently less polysemantic than dense FFNs, and that monosemanticity increases as the degree of sparse routing increases. (cf. Section 4).

- We demonstrate that zooming out to the expert level is an effective and scalable method for interpreting MoEs, which we validate through causal attribution. (cf. Section 5).

- We provide empirical evidence that experts are neither broad domain specialists nor simple token processors, but rather specialized task experts performing linguistic and semantic operations (cf. Section 6).

## 2. Related Work

**Interpretability in Dense Transformers.** Interpretability research in dense models has largely focused on disentangling the polysemantic activations of neurons. Post-hoc methods like sparse autoencoders (SAEs) (Bricken et al., 2023) and Transcoders (Dunefsky et al., 2024) have become the standard for finding interpretable concepts in an unsupervised manner, but they remain computationally expensive, requiring large datasets and compute to train for every layer. Furthermore, they have been shown to have several limitations (Heap et al., 2025; Paulo & Belrose, 2026; Kantamneni et al., 2025; Minegishi et al., 2025). Other techniques such as probing (Alain & Bengio, 2016; Gurnee et al., 2023), Logit Lens (nostalgebraist, 2020) and Direct logit attribution (DLA) (Elhage et al., 2021) allow researchers to map internal activations to specific concepts or output tokens. These

methods have been a useful tool in many interpretability works (nostalgebraist, 2020; Zhong et al., 2024a; Chughtai et al., 2024; Conneau et al., 2018; Tenney et al., 2019).

**MoE as a Path to Interpretability.** The idea that MoE models might inherently facilitate interpretability has often been suggested in the literature (Sharkey et al., 2025; Elhage et al., 2022; Chaudhari et al., 2025). This speculation is driven by the belief that architectural sparsity naturally reduces the pressure for superposition, leading to cleaner, more modular representations. Recently, Chaudhari et al. (2025) used toy models to show that superposition decreases as routing sparsity increases. However, it remains an open question whether this trend holds in large-scale LLMs, where the router must balance millions of parameters and diverse data distributions. Recent work has begun to analyze expert dissimilarity (Lo et al., 2025) and semantic routing (Olson et al., 2025), but a unified understanding of what experts actually do is still missing.

**Architectures Aimed at Interpretability.** Beyond analyzing existing models, a growing line of research seeks to build inherently interpretable architectures, often leveraging MoE-style designs to enforce modularity. Recent work has explored using MoE layers or sparsity to promote monosemanticity (Oldfield et al., 2024; Zhong et al., 2024b; Yang et al., 2025b; Park et al., 2025; Kang et al., 2025; Gao et al., 2025). Our work complements these efforts by empirically validating that the experts in standard MoE models exhibit the properties these specialized architectures aim to induce.

**The Debate on Expert Specialization.** Current literature is divided between domain-level specialization (Muennighoff et al., 2025; Dai et al., 2024; Liu et al., 2024; Riquelme et al., 2021) where experts adapt to broad themes like coding, and token-level specialization (Xue et al., 2024; Jiang et al., 2024), where routing is determined by syntactic markers. Our work bridges this gap by demonstrating that experts function as fine-grained task experts.

## 3. Preliminaries

We analyze decoder-only transformer language models. We use the term *component* to refer to either a single MoE expert or a dense FFN sublayer.

**Transformers and the Residual Stream.** Each token position maintains a hidden state $r^{(l)} \in \mathbb{R}^d$. Layers are arranged such that each sublayer (attention or FFN/MoE) produces an update vector $\Delta r$ that is added back to the *residual stream*:

$$r^{(l+1)} = r^{(l)} + \Delta r_{\text{attn}} + \Delta r_{\text{ffn}}$$

The update $\Delta r_{\text{ffn}}$ is produced by either a dense FFN or an MoE layer. The additive structure is the key property

enabling attribution methods, as it allows us to decompose the final representation into the sum of contributions from every preceding component. The model produces logits for the next token by applying a final normalization and an *unembedding* matrix $W_U \in \mathbb{R}^{d \times |\mathcal{V}|}$, where $|\mathcal{V}|$ is the vocabulary size.

**Mixture-Of-Experts (MoE).** An MoE layer consists of $N$ independent expert networks $\{E_1, \ldots, E_N\}$. Each expert $E_i$ is a feed-forward network, typically using the SWiGLU architecture (Shazeer, 2020). For an input $x$, the expert computes an intermediate activation vector $\mathbf{h} \in \mathbb{R}^{d_{\text{ff}}}$

$$\mathbf{h} = \text{Swish}(W_{\text{gate}}x) \odot W_{\text{up}}x \qquad (1)$$

where $\odot$ denotes the element-wise product and $\text{Swish}(z) = z \cdot \sigma(z)$. The final output is then produced by a down-projection: $E_i(x) = W_{\text{down}} \mathbf{h}$. We refer to the individual components $\mathbf{h}_j$ of the vector $\mathbf{h}$ as the **neurons** of the expert. Note that we can also apply Equation (1) to dense FFNs.

For each input $x$, a router network $R$ produces scores $s = R(x) \in \mathbb{R}^N$, which determine how strongly each expert is activated. A subset of $N_A$ experts is selected based on these scores (e.g., via Top-$N_A$ selection). The corresponding routing weights $g_i$ are then derived from $s$ (for instance via a softmax over some or all experts), with $g_i > 0$ for selected experts and $g_i = 0$ otherwise. The layer output is computed as

$$y = \sum_{i=1}^{N} g_i \, E_i(x).$$

We define the routing sparsity as the ratio $N_A/N$, where smaller values indicate sparser routing.

**Probing.** To measure the polysemanticity of a component $c$, we use $k$-sparse probing (Gurnee et al., 2023). This technique trains a linear classifier to predict a binary concept $y \in \{0, 1\}$, using only $k$ dimensions of an activation vector $h$. By varying $k$, we can measure how "smeared" a concept is across neurons. Following Gurnee et al. (2023), we select these $k$ neurons by identifying those with the highest absolute difference in mean activations between positive and negative samples:

$$a_j = |\mathbb{E}[h_j \mid y = 1] - \mathbb{E}[h_j \mid y = 0]|,$$
$$\mathcal{S}_k = \text{TopK}_k \left( \{a_j\}_{j=1}^d \right),$$

where $\text{TopK}_k$ returns the indices of the $k$ largest values. We then train a logistic regression probe with $L2$ regularization on only the dimensions $h_{j \in \mathcal{S}_k}$. If a concept can be accurately predicted at $k = 1$, it suggests the component contains a monosemantic neuron for that concept.

**Logit-Space Projections and Attribution.** Both the Logit Lens (nostalgebraist, 2020) and Direct Logit Attribution (DLA) (Elhage et al., 2021) analyze how a component update $v \in \mathbb{R}^d$ influences output logits by projecting $v$ into vocabulary space using the unembedding matrix $W_U \in \mathbb{R}^{d \times |\mathcal{V}|}$.

Given $v^{(l)}$ at layer $l$, the Logit Lens maps the component to logits

$$\ell^{(l)} = v^{(l)} W_U,$$

providing a snapshot of the model's intermediate predictions.

DLA extends this idea to quantify how $v$ affects the logit of a target token $t$. Because layer normalization (LN) is nonlinear, a first-order linearization around the final residual state is used, yielding the approximate contribution

$$A_{v \to t} = \text{LN}_{\text{linear}}(v)^\top W_U[:, t],$$

where $W_U[:, t]$ is the unembedding vector for token $t$. This linearization makes contributions approximately additive, enabling direct comparison of how different components influence specific token logits.

**Automatic Interpretability.** We use an LLM-based explainer (Bills et al., 2023) to generate natural language hypotheses for MoE experts. The explainer is provided with text snippets and tasked to generate a label. A separate scorer LLM then evaluates these hypotheses on held-out examples to produce an interpretability score (Paulo et al., 2025).

## 4. Quantifying the Monosemanticity of MoE Experts

To determine if MoEs are inherently more interpretable, we compare their internal representations to those of dense FFNs using $k$-sparse probing. We hypothesize that the structural constraint of sparse routing ($N_A/N$) reduces the pressure for *superposition*, the mechanism where a network represents more concepts than it has neurons (Elhage et al., 2022). By measuring the degree to which representations are distributed, we can infer the severity of neuron-level polysemanticity within the model.

### 4.1. Probe Selection and Methodology

We evaluate 12 different models (Appendix B; Table 2) across 58 concepts spanning four categories: Part-of-Speech, LATEX, code, and natural language text (Appendix C; Tables 3 and 4). For each concept, we initially collect 5,000 token samples, balanced between positive and negative classes. For MoE models, we filter this dataset to include only the subset of tokens that were routed to the target expert. By excluding unrouted tokens, we explicitly measure the rep-

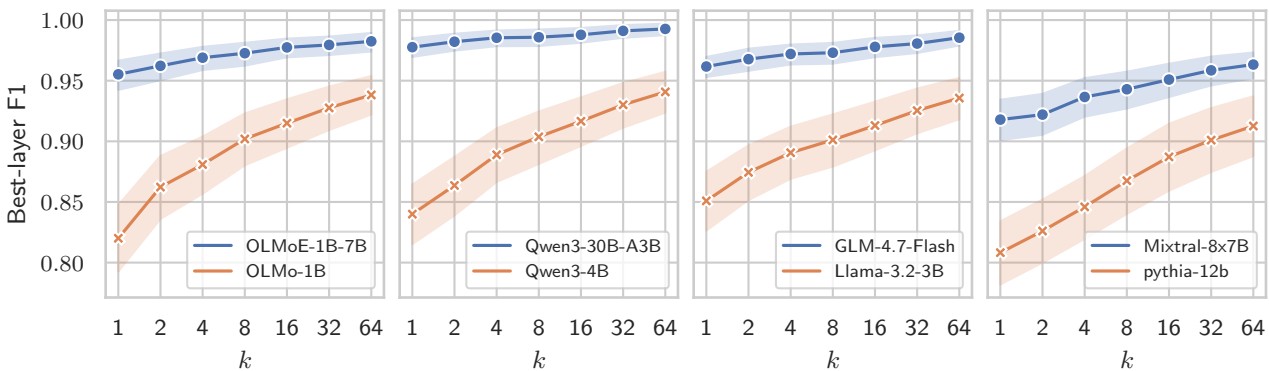

*Figure 1.* Best-layer F1 score for probes trained on MoE and dense models. Models are matched based on active parameter count, and if available from the same model family. Shaded regions represent 95% confidence intervals around the mean estimate over concepts at each $k$-value. Red lines represent dense models while blue lines represent MoE models. See Figure 8a in Appendix A for additional model comparisons.

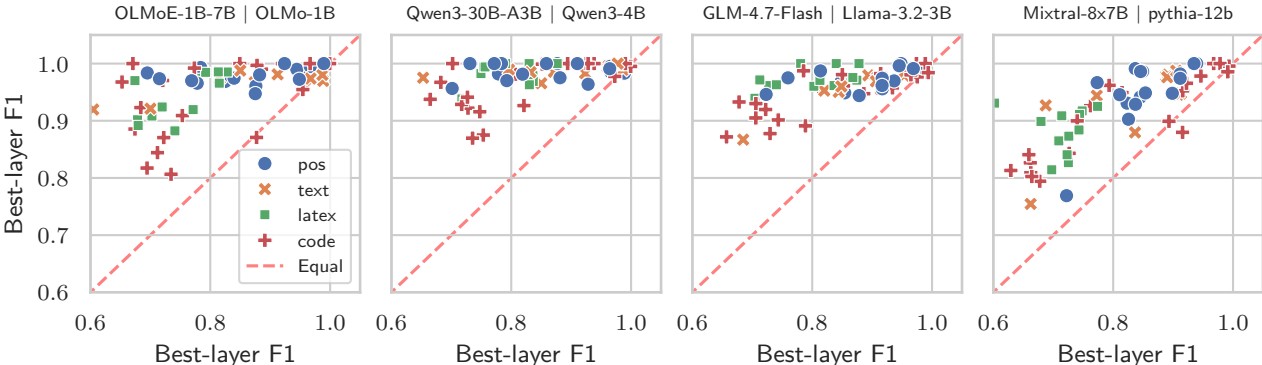

*Figure 2.* Comparison of best-layer probes trained on MoE experts against probes trained on dense models. MoE models are on the y-axis and dense models are on the x-axis. Models are matched based on active parameter count, and if available from the same model family. See Figure 8b in Appendix A for additional model comparisons.

resentation as it exists strictly within the expert's local subspace, acknowledging that the router has already acted as an initial coarse filter. All $k$-sparse probes are trained using a 75/25 train-test split on this filtered data. Consequently, all reported F1 scores reflect the probe's performance strictly on the held-out 25% test set.

To ensure we capture the model's maximum representational capacity for any given concept, we employ a best-layer selection strategy. For each concept and each value of $k \in \{1, 2, 4, 8, 16, 32, 64\}$, we train probes on *every* layer (and *every* expert for MoEs). Specifically, we probe the intermediate activation vector $\mathbf{h}$ (as defined in Equation (1)). We then identify the single best-performing layer for that concept across the entire model. For MoE models the best layer is selected based on the best expert's performance. This methodology allows us to compare the upper bound of interpretability for both architectures, ensuring that our findings are not an artifact of looking at the wrong layer. To ensure a fair comparison, we match MoE and dense mod-

els based on their active parameter count. Additionally, to control for the total parameter count, we perform a direct comparison within the OLMo family.

### 4.2. Experts Approach Monosemanticity

A key indicator of superposition is the degree to which a representation is distributed (Rumelhart et al., 1986). In a highly polysemantic dense model, a concept is typically smeared across dozens of neurons; thus, a probe at $k = 1$ (a single neuron) will often perform poorly, while performance only recovers as $k$ increases. Conversely, in a monosemantic representation, the concept is pinned to a single index $\mathbf{h}_j$ (neuron). If MoE experts were as polysemantic as dense FFNs, we would expect a significant performance gap between $k = 1$ and higher values of $k$.

However, as illustrated in Figure 1, MoE experts often achieve near-optimal F1 scores at $k = 1$, implying that the neurons $\mathbf{h}$ are dedicated to specific concepts. Across all

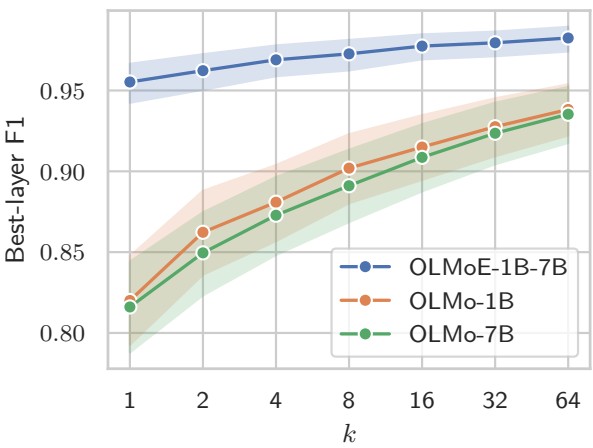

*Figure 3.* Comparison of best-layer probes across the OLMo family. Shaded regions represent 95% confidence intervals around the mean estimate over concepts at each $k$-value.

models, the gap in performance is largest at $k = 1$, where MoE experts often achieve near-perfect F1 scores while dense models struggle. This suggests that sparse routing encourages the model to assign monosemantic neurons to specific concepts, thereby reducing superposition. Furthermore, we observe that the variance in performance is much lower for MoE experts; while dense models struggle to represent certain concepts entirely, MoE experts represent the majority of probed concepts cleanly.

### 4.3. Consistency Across Concept Categories

To ensure these findings are not limited to specific types of knowledge, we analyze performance across four distinct categories: Part-of-Speech, LaTeX, Code, and Natural Language Text. As illustrated in Figure 2, MoE models (y-axis) consistently outperform dense models (x-axis) across all categories. Nearly every point lies above the equality line, demonstrating that MoE experts are better at representing diverse concepts monosemantically.

We also address the potential confounder that MoEs simply benefit from higher total parameter counts. By comparing the OLMo family (Figure 3), we find that OLMoE-1B-7B (1B active) significantly outperforms the OLMo-7B dense model. Despite the dense model having $7\times$ more active parameters per token, it still exhibits higher superposition. This provides strong evidence that sparse routing, rather than raw capacity, is the primary driver of reduced polysemanticity in our comparisons.

### 4.4. The Impact of Routing Sparsity

The results also show a systematic relationship between polysemanticity and the routing sparsity ($N_A/N$). As shown in

Figure 4, models with the highest degree of sparsity (lowest $N_A/N$) exhibit the cleanest representations. This trend is further validated by Mixtral-8x7B, which is the densest MoE in our study ($N_A/N = 0.25$). While Mixtral-8x7B still outperforms its dense counterparts, its interpretability scores are noticeably lower than those of sparser models like Qwen3-30B-A3B ($N_A/N \approx 0.06$). This supports the claim that the interpretability of MoEs scales with the degree of sparse routing: as routing sparsity increases, the internal units become increasingly monosemantic. This suggests that the industry trend toward models with more total experts and fewer active experts per token—a trend driven by performance scaling laws—is simultaneously making these models more monosemantic.

### 4.5. From Neurons to Experts

The finding that MoE neurons exhibit significantly lower polysemanticity than dense FFN neurons has a profound implication for model analysis. In dense models, the high degree of superposition means that any sub-layer or group of neurons is likely performing thousands of disparate computations simultaneously. Because the neurons fire on almost all tokens, their aggregate output is a superposition of concepts, making zoomed-out interpretability nearly impossible.

In MoE architectures, however, two distinct mechanisms work in tandem to create what we term *modular monosemanticity*. As demonstrated in our probing experiments, the architectural pressure of sparse routing is associated with individual expert neurons being less polysemantic. As established by prior work on MoE routing (e.g., (Muennighoff et al., 2025; Olson et al., 2025)), experts do not see the entire data distribution; the router acts as a filter, ensuring an expert is only activated for a restricted, semantically or syntactically related subset of tokens. If an expert is composed of mostly single-concept neurons, and it is only triggered by a coherent family of inputs, we should expect that the aggregate computation of that expert will reflect a higher-order task.

This synergy motivates a shift in our unit of analysis. Rather than relying on computationally expensive post-hoc methods to untangle individual neurons, we can zoom out. Because the constituent neurons are relatively monosemantic and their activations are contextually bound by the router, we can treat the entire expert as an interpretable module.

## 5. Automatically Interpreting MoE Experts

Leveraging the modular monosemanticity identified in Section 4, we now move from the neuron level to the expert level. This zooming out allows us to interpret the model's logic at a scale that is computationally infeasible with tra-

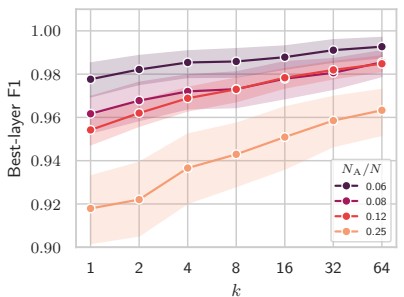

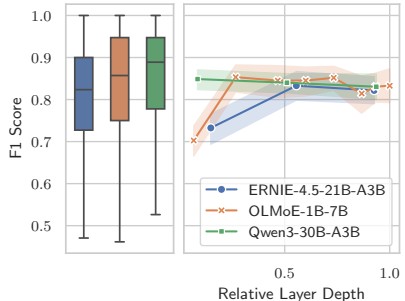

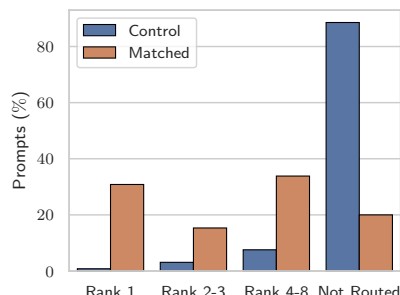

*Figure 4.* Comparison of the Best-layer F1 score for different $N_A/N$ ratios.

*Figure 5.* Automatic Interpretability F1 scores. (left) Distribution (right) Average per layer.

*Figure 6.* Percentage of prompts for which an expert achieved a high rank or did not get routed. Control prompts show the average rank on prompts designed for other experts.

ditional neuron-by-neuron or sparse coding-based methods. By treating each expert as a functional block, we can automatically generate natural language descriptions of their roles and validate these descriptions.

In this section, we describe our pipeline for automatic labeling and provide evidence that experts, unlike dense FFNs, behave as causally coherent units.

### 5.1. Automatic Labeling

To interpret MoE experts, we use a two-stage LLM-based pipeline consisting of an explainer and a scorer (Bills et al., 2023). For a target expert $E_i$, we identify text sequences from the pile-uncopyrighted dataset (Gulliver, 2023) where the expert is "highly active".

Identifying high-activating examples for an entire expert is not as straightforward as it is for a single neuron. We cannot simply rely on the router weight $g_i(x)$, because being selected by the router only means the expert was given the opportunity to process the token; it does not guarantee the expert actually performed a significant computation (i.e., it might output a vector near zero). Likewise, aggregating the scalar activations of internal neurons does not reflect the expert's final output.

In a transformer, the only way a component influences the model's final prediction is by writing an update vector to the residual stream. Therefore, to find examples where an expert is truly "active" and causally impactful, we must measure the magnitude of its contribution to the residual stream. For a given token $x$, this contribution is the expert's output vector $E_i(x)$ scaled by the router weight $g_i(x)$. We measure this using the L2 norm: $g_i(x) \|E_i(x)\|_2$. A larger norm geometrically corresponds to a larger shift in the model's internal representation.

We evaluate text snippets to provide the explainer LLM with sufficient context. To guarantee that a sequence contains

at least one prominent spike where the expert heavily influenced the residual stream, we score each sequence $s$ by finding the maximum score across its constituent tokens:

$$\text{score}(s, E_i) = \max_{x \in s} \ g_i(x) \|E_i(x)\|_2$$

We provide an LLM explainer with 20 top activating sequences. To help the explainer identify the expert's role, we use Logit Lens to find the top 3 tokens promoted by the expert at the moments of peak activation. The explainer is tasked with generating a concise, one-sentence hypothesis for the expert's computational role (see Appendix J for prompts and Appendix D for sequence selection details).

We list all labels produced by this procedure in Appendix K (Tables 9 to 11). To validate the labels, we use a separate LLM scorer. The scorer is given 10 positive examples (where the expert was active) and 10 negative examples (where other experts in the same layer were active). The scorer must detect if each example fits the generated label. We then calculate the F1 score over the choices the scorer made. For both the explainer and scorer model we use `Gemini 3 Flash Preview`.

### 5.2. Expert Interpretability

We apply our method to all experts in 8 layers of `OLMoE-1B-7B`, 3 layers of `ERNIE-4.5-21B-A3B`, and 3 layers of `Qwen3-30B-A3B`. As shown in Figure 5 (right), the resulting F1 scores are consistently high across all layers. Most experts achieve F1 scores above 0.8, with very few failing to be interpreted (see Appendix E for failure cases). This suggests that experts are not just clean at the neuron level, but also act as coherent units that can be described in natural language. This level of interpretability is maintained across the layers we analyze. We also observe a correlation between a model's routing sparsity ($N_A/N$) and the reliability of its automatic labels. As illustrated in Figure 5 (left), average interpretability scores are consistently higher for models

with sparser routing. Specifically, Qwen3-30B-A3B, the most sparse model in our study ($N_A/N \approx 0.06$), achieves the highest average F1 scores, frequently exceeding 0.9. This suggests that the cleaner representations we identified at the neuron level in Section 4 translate directly into more coherent units at the expert level. In contrast, denser MoEs like ERNIE-4.5-21B-A3B exhibit slightly lower and more variable scores, likely because their experts are still forced to balance multiple, overlapping semantic features.

### 5.3. Causal Attribution

While high F1 scores indicate that our labels are descriptive, they do not prove that the experts have a causal effect on the model's output. To verify this, we design a "Trigger-Target" experiment. For a given expert label, we generate new, synthetic test cases where we define a trigger word (where we expect the expert to activate) and a target word (a word which the expert should promote).[2] For example, in

"We need to address the elephant in the room"

the is the trigger word and room is the target word. Note that the test cases are solely generated based on the expert's automatic interpretability label from Section 5.1. For further examples see Appendix F.

We run a forward pass and measure the expert's ranking among all experts from the same layer in terms of its DLA contribution to the target word. We run this experiment for Layers 4, 9 and 14 of the OLMoE-1B-7B model and select 10 random experts from each layer for which we generate 20 test cases. Test cases are all generated by Gemini 3 Flash Preview (see Appendix J for an example prompt) and then checked manually.

As shown in Figure 6, the results provide strong attributional evidence for our labels:

**Matched Prompts.** In the majority of cases, the expert we identified was either the Top-1 or among the Top-8 contributors to the target word.

**Control Prompts.** When we checked the same experts on prompts designed for other experts from the same layer, they were almost never routed and had almost no attribution to the output.

In 80% of the cases, the specific expert was not even routed to control prompts, whereas it was consistently routed and highly influential for matched prompts. This confirms that our zooming out approach captures the true causal mechanics of the model: the expert level is a valid and effective unit of analysis.

---

[2]Because of tokenization it can happen that the target or the trigger are split into multiple tokens. In that case, we select the token with the highest routing weight for the trigger word and the first token for the target word.

## 6. Expert Specialization

The nature of expert specialization has long been a point of contention. One camp argues that experts specialize in broad semantic domains like coding or biology (Liu et al., 2024; Muennighoff et al., 2025; Dai et al., 2024), while another suggests they primarily handle surface-level syntactic concepts (Xue et al., 2024; Jiang et al., 2024). In this section, we show that both views are incomplete. By analyzing the specialization of experts across layers, we demonstrate that experts function as fine-grained task experts, performing precise computational operations that are often domain-restricted but functionally specific.

### 6.1. A Taxonomy of Expert Roles

Across models and layers, the labels from Section 5 cluster into a small set of roles (Table 1). In our examples, these roles form a loose hierarchy: early experts bind morphology, mid-layer experts stabilize syntax, deeper experts retrieve domain knowledge, and late experts enforce formatting constraints. This is consistent with viewing the residual stream as a communication channel (Elhage et al., 2021), where different components iteratively refine the next-token distribution. Many experts also behave like key–value memories (Geva et al., 2021). We denote an expert $E$ in layer $L$ of model $M$ as M-L0-E0. See Appendix H for in-depth case studies of individual experts.

**Morphological**: We observe experts that seem to be responsible for gluing text back together. Because LLMs process text as tokens (often sub-words), these experts focus on suffixes, prefixes, and stems. For example, OLMoE-L1-E57 activates on amine in glutamine and promotes subword continuations (e.g., iaz, endar, uba) to help the model construct rare chemical terms.

**Syntactic**: Some mid-layer experts behave like syntactic continuers: when they see coordinating conjunctions (e.g., and, or, for, but), they upweight likely completions. In boots and the like, ERNIE-L15-E0 activates on and and promotes alike, like. We see similar behavior in other coordination contexts.

**Semantic**: We find experts, mostly in mid-to-late layers, that represent closely what one would call a *domain* expert. For example, OLMoE-L4-E3 operates mostly in legal and patent related documents and promotes tokens that reinforce patent-style and legal-technical document continuation such as patents, applications, inventor.

**Operational**: We also find experts that primarily enforce local validity constraints. OLMoE-L15-E17 activates inside LaTeX formatting blocks (e.g., \mathbf{b} ) and strongly promotes closing delimiters such as }}.

**Hyper-specialized experts:** MoE routing can allocate ded-

*Table 1.* Categorization of MoE Experts Across Layers. The categorization is based on the labels from Section 5.

| Category | Functional Role | Representative Expert | Label | Target Tokens / Contexts |
|---|---|---|---|---|
| **Morphological** | Morphology & Tokenization | **OLMoE-L1-E57** | Chemical & Biological suffixes | `-amine, -ine, -ium, -ase` |
| **Syntactic** | Syntax & Grammar | **ERNIE-L15-E0** | Syntactic Coordination | `and, or, that, to, for` |
| **Semantic** | Domain knowledge | **OLMoE-L4-E3** | Patent/Legal citations | `Case v. State, U.S., claim, Pat.` |
| **Operational** | Structural validity & formatting constraints | **OLMoE-L15-E17** | Closing LaTeX environments | `\end{...}, \mathbf{b}, \sqrt{q}` |

icated capacity to very narrow concepts. For instance, `Qwen3-L44-E12` responds to Iranian administrative geography (Province → County → District), promoting location names and census-style boilerplate. `OLMoE-L14-E59` behaves like a role-playing-game mechanic completer: in D&D-style contexts it promotes rule-specific continuations (e.g., associates abbreviations such as `DR` with `Damage Reduction`).

### 6.2. Experts in the Output Embedding Space

While the natural language labels in the previous section provide a human-readable map of expert functions, they remain qualitative. To confirm that this modularity is a structural property of the architecture and not just a byproduct of our labeling process, we require a model-native, quantitative metric. We measure *expert specialization*: the degree to which an expert's behavior isolates specific functional or semantic domains.

What constitutes a *domain* for an LLM remains an open question. Rather than imposing external human categories, we define domains natively by performing unsupervised $k$-means clustering on the model's output embedding matrix (the unembedding). This matrix is known to be a semantically and syntactically rich map of the model's vocabulary (Grindrod & Grindrod, 2025; Dar et al., 2023; Mikolov et al., 2013; nostalgebraist, 2020). To ensure our findings are not artifacts of a specific granularity, we analyze expert behavior over $10^6$ tokens across multiple resolutions ($k \in \{10, 50, 100, 1000, 5000\}$). Low values of $k$ capture broad semantic topics (e.g., biology), while high values capture more granular themes (e.g., bees and honey). See Appendix G for cluster examples.

Because natural language is highly skewed, we cannot simply measure if an expert frequently processes a specific type of token, because a random sample of text will naturally be dominated by common function words. To demonstrate that an expert is specialized, we must measure how much it deviates from the base rate of the layer.

We quantify this deviation using Jensen-Shannon Divergence (JSD). A Specialization Score of 0 indicates the expert processes tokens in the exact same proportions as the layer average (no specialization). A score approaching 1 indicates the expert is hyper-focused on a narrow semantic niche that

the rest of the layer ignores. Furthermore, because MoE routing is unbalanced, experts can process vastly different volumes of tokens. To ensure high scores are not simply the result of small-sample statistical noise, we compare every expert against a simulated Random Expert Baseline. This baseline calculates the expected JSD if the expert had simply drawn its $N$ tokens randomly from the layer's base rate. See Appendix I for a complete mathematical formulation.

To understand how experts specialize, we apply the specialization measure to two distinct stages of the MoE computation, projecting both onto the same $k$ clusters for a direct comparison:

1. **Routing Specialization (Input):** We track the actual tokens the router assigns to the expert.

2. **Functional Specialization (Output):** We apply Logit Lens to the tokens promoted by the expert's output vector.

Early-layer representations are primarily engaged in feature-building; they are functionally distant from the output vocabulary. Consequently, Logit Lens projections in early layers are known to be noisy and inaccurate. We therefore rely primarily on routing specialization for early layers, and use functional specialization to analyze the mid-to-late layers where the residual stream aligns more cleanly with the vocabulary space. By applying these scores across different cluster sizes, we can quantitatively verify the taxonomy observed in Section 6.1.

**Domain Experts (Semantic):** These experts focus on broad topics. We expect them to show high specialization at low $k$. They should exhibit both high Routing Specialization (the router sends them domain-specific text) and high Functional Specialization (they promote domain-specific concepts).

**Task Experts (Operational/Morphological/Syntactic):** These experts perform precise structural operations (e.g., closing LaTeX brackets). They may read tokens from any semantic domain (yielding a low routing score), but they consistently promote specific tokens. We expect these experts to show sharp increases in Functional Specialization at high $k$.

As shown in Figure 7, plotting the specialization scores reveals a clear trajectory across the depth of a model. In the

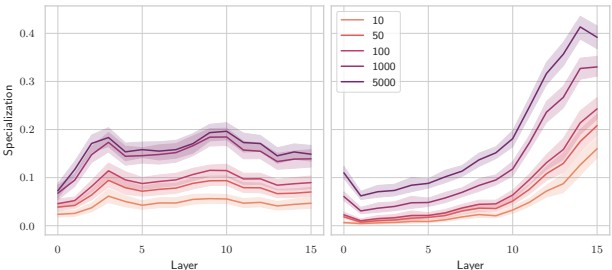

*Figure 7.* Expert specialization scores across layers for `OLMoE-1B-7B`. Scores reflect the expert's deviation from the layer's aggregate base rate. (Left) Routing Specialization. (Right) Functional Specialization.

Routing Specialization analysis, we observe a bimodal structure. The router begins sorting tokens early in the network, followed by a second, more intensive phase of semantic partitioning in the middle layers. This confirms that the router is highly selective about the data each expert receives, consistently deviating from the layer's base rate to filter specific inputs. However, the most dramatic shift occurs in Functional Specialization. In late layers, the degree to which experts promote unique vocabulary niches increases sharply. To put this magnitude in perspective, a JSD of 0.4 in a high-dimensional vocabulary space indicates that an expert is strongly biased away from the layer's com- mon base rate toward a narrow set of tokens.

Crucially, the $k$-sweep resolves the nature of this deep-layer specialization. If experts were broad domain specialists, the score for broad categories ($k = 10$) would be high. Instead, we see that the highest granularity ($k = 5000$) pulls dramatically ahead of the broad semantic lines. The fact that experts appear far more specialized at high $k$ provides strong quantitative support for our qualitative findings in Section 6.1: deep-layer MoE experts act primarily as granular Task Experts. They do not represent broad semantic domains; rather, they take in relatively general signals and apply a highly precise functional or syntactic transformation to the output space.

## 7. Discussion

**The Scaling of Sparsity.** A promising trend in recent MoE research is the move toward increasingly sparse configurations, exemplified by (He, 2024). Our findings suggest that this trend toward extreme sparsity ($N \gg N_A$) is not only beneficial for performance but may be the key to unlocking fully monosemantic models. If the relationship between sparse routing and interpretability holds at the limit, the next generation of models may be inherently transparent by design, potentially eliminating the interpretability tax that currently plagues dense architectures.

**Experts as Sub-Circuits.** Our findings support a shift in how we conceptualize Large Language Models. Rather than viewing them as monolithic thematic encyclopedias, our results suggest LLMs function as modular toolboxes. This is consistent with the circuits view of interpretability (Elhage et al., 2021; Ameisen et al., 2025; Lindsey et al., 2024), where model computation is seen as a graph of interacting functional units. In this framework, MoE experts act as discrete sub-routines for specific tasks like LATEX state resolution or genomic acronym completion. This modularity provides a clear path for future work to map the logic of the model by studying how the router sequences these experts into complex computational pipelines.

**Limitations.** While our results are consistent across 12 models, this study has limitations. Due to GPU memory and compute constraints, we were unable to include the largest current MoE models, such as `DeepSeek-V3` (Liu et al., 2024); however, given that these models use similar degrees of sparse routing, we expect our findings to hold. Furthermore, we do not claim that experts are entirely monosemantic. Some degree of superposition likely remains and superposition between experts could also be possible. However, our results suggest that experts are sufficiently monosemantic to be captured by functional natural language labels, providing a pragmatic and effective middle ground between uninterpretable neurons and computationally expensive concept extraction.

## 8. Conclusion

In this work, we demonstrated that Mixture-of-Experts (MoE) transformer architectures possess an inherent interpretability advantage over their dense counterparts. Using $k$-sparse probing, we provided empirical evidence that MoE neurons exhibit significantly lower polysemanticity, a property that is closely associated with the architectural constraint of sparse routing. By leveraging this relative monosemanticity, we showed that *zooming out* to the expert level provides a clearer, more scalable unit of analysis, allowing us to identify hundreds of specialized task experts that perform functional operations.

## Acknowledgements

Jae Hee Lee and Stefan Wermter were supported by the German Research Foundation (DFG), project number 551629603.

## Impact Statement

This paper studies interpretability in Mixture-of-Experts (MoE) language models at the level of individual experts. We provide empirical evidence that neurons inside MoE

experts are less polysemantic than neurons in dense feed-forward layers, and we show how this can be leveraged to assign functional descriptions and to test hypotheses via causal attribution. If these properties hold more broadly, they may reduce the cost of auditing and debugging large MoE systems and enable more targeted interventions (e.g., modifying a small set of experts) compared to neuron-level analyses.

Interpretability tools can also be dual use. The same ability to localize computations to particular experts may help malicious actors identify components to manipulate, bypass safety behaviors, or extract capabilities in a more targeted way than coarse fine-tuning. There is also a risk of over-trusting labels: a short natural-language description can hide important context such as prompt dependence, dataset artifacts, or interactions between experts. We therefore recommend treating expert labels as tentative summaries, validating them with counterfactual tests (including ablations and out-of-distribution checks), and not using interpretability alone as a safety guarantee. Our experiments analyze existing models rather than training new foundation models; the primary resource costs are forward passes and probe training.

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

# A. Further Probing Results

We provide additional probing results that support the comparisons in Section 4. Models are matched based on active parameter count where possible.

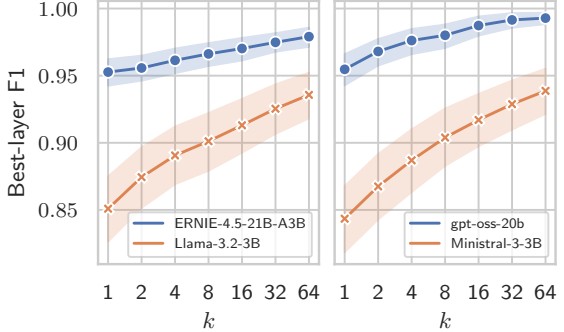

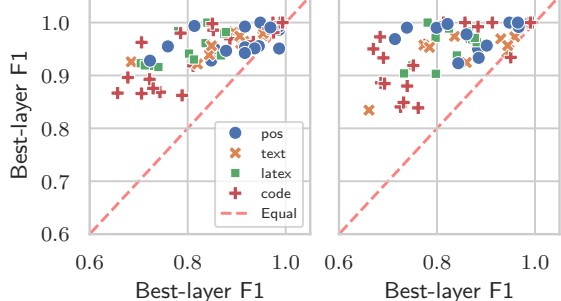

*(a)* Best-layer F1 score for $k$-sparse probes. Shaded regions represent 95% confidence intervals. Red lines represent dense models and blue lines represent MoE models.

*(b)* Comparison of best-layer probes trained on MoE experts against probes trained on dense models. MoE models are on the y-axis and dense models are on the x-axis.

*Figure 8*

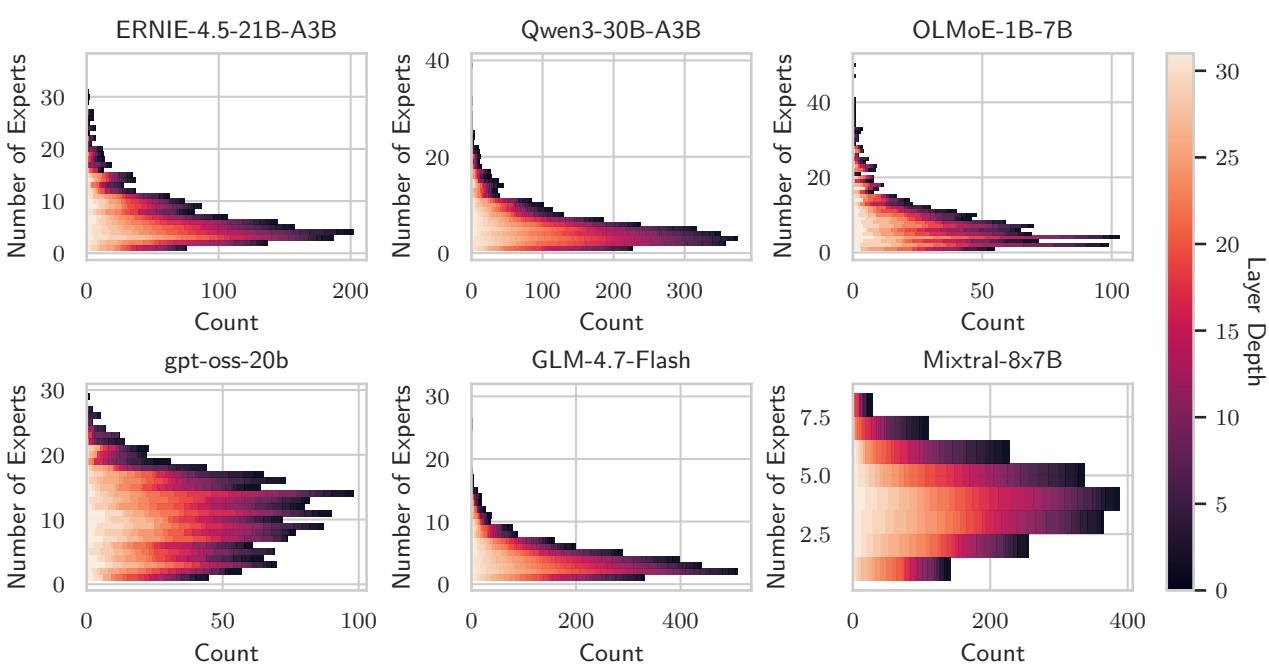

*Figure 9.* Estimated number of experts for each concept. For each concept and layer, experts whose F1 probe score is within 95% of the best expert are counted as active. The concept counts are stacked by layer.

## B. Model Selection

We list the dense and MoE models used in our probe comparisons, along with their routing configurations ($N$, $N_A$, $N_{SE}$) and architectural details needed to interpret parameter-matching choices. All model parameters were loaded using either `bfloat16` or `float16` numerical precision. We apply 8-bit quantization to `Mixtral-8x7B-v0.1`.

*Table 2.* MoE and dense language models for which we train probes. $N$ = Number of Total Experts, $N_A$ = Number of Active Experts, $N_{SE}$ = Number of Shared Experts

| Model | $N$ | $N_A$ | $N_{SE}$ | # Layers | FFN/Expert dim | FFN/Expert Style |
|---|---|---|---|---|---|---|
| OLMo-1B | - | - | - | 16 | 8192 | SwiGLU |
| OLMo-7B | - | - | - | 32 | 11008 | SwiGLU |
| LLama-3.2-3B | - | - | - | 28 | 8192 | SwiGLU |
| Qwen3-4B-Base | - | - | - | 36 | 9728 | SwiGLU |
| Ministral-3-3B | - | - | - | 26 | 9216 | SwiGLU |
| pythia-12b | - | - | - | 36 | 20480 | GELU |
| OLMoE-1B-7B | 64 | 8 | 0 | 16 | 1024 | SwiGLU |
| ERNIE-4.5-21B-A3B | 66 | 6 | 2 | 28 | 1536 | SwiGLU |
| Qwen3-30B-A3B | 128 | 8 | 0 | 48 | 768 | SwiGLU |
| gpt-oss-20b | 32 | 4 | 0 | 24 | 2880 | SwiGLU[3] |
| GLM-4.7-Flash | 65 | 4 | 1 | 47 | 1536 | SwiGLU |
| Mixtral-8x7B-v0.1 | 8 | 2 | 0 | 32 | 14336 | SwiGLU |

## C. Probing Datasets

We document the datasets and concept definitions used for $k$-sparse probing. The intent is to make the evaluation reproducible and to clarify which concepts are extracted via regex heuristics versus provided at the word level (Part-of-Speech tags). The regexes can be found in our published codebase.

*Table 3.* Probing datasets. We largely follow a similar approach as (Gurnee et al., 2023) and use some of their concepts. However, for the code category, we design our own fine-grained concepts.

| Category | Dataset | Total Concepts |
|---|---|---|
| Part-of-Speech | POS tagged Wikipedia (simple spacy subset) | 16 |
| LaTeX | pile-uncopyrighted (ArXiv subset) | 12 |
| code | pile-uncopyrighted (Github subset) | 20 |
| text | pile-uncopyrighted (All subsets) | 10 |

*Table 4.* Probing concepts. The token positions for latex, code and text concepts are extracted using regular expressions, while the Part-of-Speech concepts are available on the word level.

| Dataset | Concepts |
|---|---|
| Part-of-Speech | adjective, adposition, adverb, auxiliary, coordinating conjunction, determiner, noun, numeral, particle, pronoun, proper noun, punctuation, subordinating conjunction, symbol, verb, other |
| LaTeX | is_superscript, is_subscript, is_inline_math, is_display_math, is_math, is_denominator, is_numerator, is_frac, is_author, is_title, is_reference, is_abstract |
| code | is_function_def, is_function_call, is_assignment, is_class_def, is_import, is_comment, is_string_literal, is_control_flow, is_loop, is_conditional, is_exception_handling, is_array_literal, is_method_call, is_lambda, is_operator, is_constant, is_boolean, is_null, is_decorator, is_async, |
| text | leading_capital, leading_loweralpha, all_digits, is_not_ascii, contains_all_whitespace, all_capitals, is_not_alphanumeric, contains_whitespace, contains_capital, contains_digit |

---

[3]OpenAI's implementation is slightly different from the other models. They include clamping and a residual connection.

## D. Automatic Interpretability Data Selection

We extract activations over the pile-uncopyrighted dataset (Gulliver, 2023). For each document, we extract one random sequence containing 32 tokens until $2 \times 10^6$ tokens have been processed. This is intended to provide diverse examples from different pile-subsets for the explainer and scorer models. We then collect the top 40 examples for each expert based on the scores described in Section 5. We randomly select 20 examples for the explainer model, 10 for the scorer model as positive examples and 10 as negative examples for other experts.

## E. Automatic Interpretability Failure Cases

We analyze 5 failure cases of our automatic interpretability pipeline for MoE experts where the final F1 score was unusually low.

### E.1. OLMoE-L1-E2

| Text Examples |
|---|
| @ubuntu@http@ms-wbt-server@up**np**[A3C]ntrain_worker_num : 20 |
| Deals Product Information & CharacteristicsThe Otto **bed** by Joseph has been up**holstered** in Chocolate Brown faux leather. This outstanding bed features a high |
| create an "exchange of experiences" for the We**hrmacht** rear unit commanders. Participating officers were selected on the basis of their |

*Figure 10.* Text examples for `OLMoE-L1-E2`. Examples are taken from the data the explainer model saw. Highlighted words are tokens routed to this expert which also received a high score.

For `OLMoE-L1-E2` the generated label was *"Mid-word and terminal suffixes within proper nouns, brands, and technical terms"* (F1 score: 0.38). See Figure 10 for text examples for this expert.

This label failed because the expert actually activates on tokens that follow specific prefixes (e.g., up, off, Of, we, We). The explainer overfit because these prefixes frequently appeared in brands (e.g., WeWork), technical terms (e.g., upholstered, upnp) or proper nouns (e.g., Offenbach, Wehrmacht). Consequently, the scorer strictly evaluated based on the flawed hypothesis, correctly resulting in a low F1.

### E.2. OLMoE-L11-E5

| Text Examples |
|---|
| to reproduce the saturation properties of the nuclear matter. At the phase transition, we maintain strict thermodynamic conditions; **i.e.**, the Gibbs conditions\u201 |
| k, **ze**ta-**re**ceptors are reported to be involved in the non-opioid actions of the peptide, **i.e.** the inhibitory effect on |
| transition so that the energy produced by the transition can go predominantly into the photon; **i.e.** to produce light rather than heat. When the conduction and val |

*Figure 11.* Text examples for `OLMoE-L11-E5`. Examples are taken from the data the explainer model saw. Highlighted words are tokens routed to this expert which also received a high score.

For `OLMoE-L11-E5` the generated label was *"Activates on specific characters to predict the second half of common abbreviations."* (F1 score: 0.46). See Figure 11 for text examples for this expert.

The explainer model suffered from frequency bias. The most frequent word is i.e. (i predicting *e*) which appeared in 11 out of 20 examples and fit perfectly to the generated label. However, the explainer ignored other examples such as word beginnings (e.g., An predicting swers (Answers) or tic (Antic); j predicting ungle (jungle); N predicting issen (Nissen)). It also ignored file paths (e.g., D or C predicting :) and URLs (e.g., n predicting pr (npr.org)). These also show up in the scorer

examples and get completely ignored by the scorer (predicting 0) resulting in very low recall (0.3) but perfect precision (1.0).

### E.3. OLMoE-L15-E10

---

**Text Examples**

R Programming Assignment Help Our java assignment help is indicated for the trainees ==who== desire ==to== stand out ==and== find out.Locus RAGS provides Programming

---

Recently Deleted things. Peruse the rundown of documents. ==You can likewise== go to Settings > Restore Files. Following 30 days,

---

world. For a very long time, the ideologists of "free market" economics ==have== been able ==to successfully== conflate "democracy" with the control of

---

*Figure 12.* Text examples for OLMoE-L15-E10. Examples are taken from the data the explainer model saw. Highlighted words are tokens routed to this expert which also received a high score.

For OLMoE-L15-E10 the generated label was *"Predicts achievement and overcoming verbs following modal verbs, adverbs, and infinitives."* (F1 score: 0.18). See Figure 12 for text examples for this expert.

For this expert the pipeline fails because the explainer model correctly captures the syntactic structure but constructs an overly narrow semantic constraint (semantic overfitting). As a result, the scorer model attains perfect precision (1.0) but extremely low recall (0.1) by rejecting most valid instances.

The explainer correctly identifies the left-context pattern (*"following modals, adverbs, and infinitives"*) but introduces semantic bias. Influenced by a small subset of salient examples containing words like defeat, dominate, prevail, and successfully, it incorrectly concludes that the expert specializes in *"achievement"* or *"overcoming"* verbs, ignoring the majority of cases involving generic action verbs.

The scorer, adhering to this flawed specification, behaves consistently: it produces true positives only when verbs explicitly match the *"achievement/overcoming"* category (e.g., evade, subvert), yielding precision of 1.0, but generates numerous false negatives by rejecting valid examples containing ordinary verbs such as pull, do, call, and raise, leading to severely degraded recall.

### E.4. Qwen3-L24-E76

---

**Text Examples**

programs are running? In other words, who will ==watch the watchers==? A: Humans are watchers for tools like supervisor. There are 3rd party plugins

---

on Murphy's Law: Anything ==that can== go wrong, ==will== go ==wrong==. As we have grown, we've become a bit soft. We figure, why not

---

.], Socrates says that you can't seek ==what you== don't know, because ==you== don't know what to seek. Yet in [2], he

---

*Figure 13.* Text examples for Qwen3-L24-E76. Examples are taken from the data the explainer model saw. Highlighted words are tokens routed to this expert which also received a high score.

For Qwen3-L24-E76 the generated label was *"Syntactic elements and connectors within philosophical, legal, or logical propositions and laws."* (F1 score: 0.30). See Figure 13 for text examples for this expert.

For this expert the pipeline failed due to domain/genre overfitting by the explainer model. While it correctly identified the rhetorical structure (logical reasoning and propositions), it artificially constrained its hypothesis to specific academic domains after being misled by a few salient named entities. This led to the incorrect rule that the expert activates only on *"philosophical, legal, or logical propositions"*, causing the scorer model to reject valid examples from everyday and technical

contexts and resulting in very low recall (0.2). In addition, the expert promoted almost exclusively Chinese tokens[4], which could have been the cause of the misleading label.

In reality, the expert exhibits a broad activation pattern across explanatory and causal reasoning, firing on general truths, hypothetical scenarios, and logical explanations regardless of topic. This pattern is consistent across ground truth positives spanning technical, everyday, business, and moderation contexts. The explainer failed because it fixated on prominent references such as `Murphy's Law`, `Zawinski's Law`, the `Böckenförde dilemma`, and *Socratic statements*, mistakenly inferring a domain-specific rule instead of recognizing the underlying function. Consequently, the scorer followed this flawed constraint, correctly identifying only explicitly philosophical or legal cases, but rejecting the majority of valid examples as false negatives simply because they did not match the imposed domain restriction.

### E.5. ERNIE-L15-E54

---

**Text Examples**

this important? Bitso buying Unisend is an effort to achieve scale in the burgeoning bitcoin market in Mexico. It is also a sign of further

fewer components than the pitch-circle-disc type of planetary gear assembly. This aspect is very important because an assembly having fewer parts to assemble is easier to mass

multiple places you mitigate your risk if one or two of your holdings crash. This is also the case with an economy; if a state has a diversified

---

*Figure 14.* Text examples for `ERNIE-L15-E54`. Examples are taken from the data the explainer model saw. Highlighted words are tokens routed to this expert which also received a high score.

For `ERNIE-L15-E54` the generated label was *"Syntactic structures expressing logical explanation, definition, or significance after a demonstrative pronoun."* (F1 score: 0.18). See Figure 14 for text examples for this expert.

For this expert the pipeline failed because the explainer model constructed a strict syntactic prerequisite. In 12 of 20 examples, the subject was a demonstrative pronoun (`This`/`That`), e.g., `This is also the case`. The explainer hypothesized that a demonstrative pronoun was required, correctly identifying the semantic function but falsely restricting the trigger. Therefore, the scorer model rejected nearly all valid examples, resulting in perfect Precision (1.0) but very low Recall (0.1).

---

[4]Qwen3-30B-A3B is a Chinese model and likely trained on huge amounts of Chinese text.

## F. Additional Test Case Examples for Causal Attribution

We provide more examples for test cases used in Section 5.3. Note that the target word is not necessarily included in the text itself, we denote the target word in brackets after the actual text if that is the case. For a complete list take a look at the published codebase.

---

**Test Case Examples**

`OLMoE-L4-E46`: The study found a statistically significant correlation between the variables.

`OLMoE-L4-E46`: The researchers reported a p-value of less than 0.05 for the primary endpoint.

`OLMoE-L4-E46`: The researchers investigated the causal relationship between the two phenomena.

`OLMoE-L9-E60`: The bustling streets of Tokyo, Japan.

`OLMoE-L9-E60`: Prime Minister Narendra Modi visited the site.

`OLMoE-L9-E60`: The heavy traffic in Dhaka, Bangladesh.

`OLMoE-L14-E0`: After being treated unfairly, he decided to (retaliate)

`OLMoE-L14-E0`: Facing constant setbacks, the entrepreneur still (persisted)

`OLMoE-L14-E0`: Enraged by the unexpected betrayal, the king (executed)

`OLMoE-L14-E59`: To calculate the melee damage, add your Strength (modifier)

`OLMoE-L14-E59`: The boss has a high resistance to physical (damage)

`OLMoE-L14-E59`: Drinking the blue elixir will restore 50 (mana)

---

*Figure 15.* Test case examples from the DLA trigger-target experiment. Trigger words are highlighted in red, while target words are highlighted in blue.

## G. Cluster Examples

We present some example clusters from the $k$-means clustering in Section 6.2.

*Table 5.* Example clusters from the output embedding matrix of `OLMoE-1B-7B`. Most clusters form either a syntactic or semantic group of related tokens.

| k | Cluster ID | Cluster Name | Token Examples |
|---|---|---|---|
| 10 | 0 | Subword stems | analys, synth, correl, estim, walked, argued, incre, determ |
| 50 | 10 | 3-digit numbers | 199, 128, 125, 255, 999, 509 |
| 100 | 65 | All-caps subword n-grams | ER, IN, AT, ST, CON, AND, AAAA |
| 1000 | 475 | Economic terminology | econom, economic, capitalism, shortages, capital, recession |
| 1000 | 108 | Computers, software, and digital technology | software, simulation, programming, desktop, online, laptop |
| 5000 | 570 | GPU, shaders and textures | texture, GPU, gpu, Shader, TEXTURE, textures |
| 5000 | 3694 | Spatial + abstract intersection | overlap, intersection, confluence, interplay, junctions |
| 5000 | 1952 | Bees | bee, bees, Honey, hone, Bee |
| 5000 | 4459 | Supernatural entities | ghost, devil, wizard, vampire, witch |

# H. Expert Case Studies

We extend Section 6.1 with 3 case studies of specific experts that illustrate notable specialization patterns.

## H.1. LaTeX Bracket Closer

This expert is `OLMoE-L15-E17`. The label generated by the explainer model is: *"Closes LaTeX mathematical environments by predicting closing braces and formatting markers."*. The scorer model achieved a perfect F1 score of 1.0 on this hypothesis. See Figure 16 for text examples.

---

**Text Examples**

```
\prod_{\tau \in {\mathcal{T}}}{\mathrm{GL}}_{a_\tau^+}$. We define $ {{\tilde{\ell}_{{\
    mathrm{can}}}}}^
```

---

```
_{{\mathbf{k}} \alpha} {\mathbf{I}},$ where ${\mathbf{H}}_\alpha$ is independent of ${\
    mathbf{k}} and ${\mathbf
```

---

```
{\bf k}}^{2}}$ and $\xi_{n {\bf k}}=\epsilon_{n {\bf k}}-\mu$ is the one-electron energy
    measured from
```

---

*Figure 16.* Text examples for `OLMoE-L15-E17`. Highlighted words are tokens routed to this expert which also received a high score.

Expert 17 activates broadly in contexts containing dense LaTeX mathematical notation, especially expressions with nested subscripts and superscripts, matrix/vector symbols (e.g., $\mathbf{C}, \mathbf{k}, \mathbf{H}$), and operators such as $\mathcal{O}, \partial$, or $\Gamma$. While many tokens in these regions are routed to the expert (including variables, formatting commands, and surrounding punctuation), the highest activations are concentrated on symbolic tokens, particularly indexed variables and single-letter identifiers like $C, k, H, R$. The promoted tokens are dominated by structural continuations such as `}}`, `}}^`, `}}_`, as well as `{}{`, `_{}{}`, indicating a strong bias toward extending and properly closing hierarchical LaTeX constructs.

Taken together, this suggests that the expert plays a primarily syntactic role, tracking the structure of mathematical expressions and promoting well-formed continuation and termination of nested symbolic notation rather than encoding domain-specific semantic content.

*Table 6.* Specialization scores for `OLMoE-L15-E17` across different granularities ($k$).

| k | Routing Specialization (input) | Functional Specialization (output) |
|---|---|---|
| 10 | 0.012 | 0.049 |
| 50 | 0.026 | 0.111 |
| 100 | 0.031 | 0.136 |
| 1000 | 0.051 | 0.294 |
| 5000 | 0.065 | 0.349 |

Our specialization scores are consistent with this analysis in Table 6. The scores are extremely low for routed tokens since the expert receives all kinds of tokens, even at the highest $k$ the score remains near 0. For functional specialization the scores tell a different story, at low $k$ the score is low, but at high $k$ the score increases drastically. This is likely because the expert predicts only a very small number of different tokens that are all very similar. The brackets the expert predicts do not belong to a single domain, instead they form a loose group which is syntactically similar. The Functional Specialization score can therefore only capture the expert's behavior at very high $k$.

## H.2. RPG Game Mechanics

This expert is `OLMoE-L14-E59`. The label generated by the explainer model is: *"predicting mechanics, stats, and character classes in tabletop and video game RPGs"*. The scorer achieved a F1 score of 0.82 on this hypothesis.

---

**Text Examples**

single target fights and On fights where there is a lot ==of== area damage, Demon==ology== warlocks, Frost DKs ==and possibly== Survival hunters

world... I miss not being part of a guild at the moment. I notice this especially ==when== we ==go into== one of the cities ==and== the trade channel sudd

Barbarian At 20th level, you embody the power of the wilds. ==Your== Strength ==and== Constitution scores increase by 4. ==Your maximum==

---

*Figure 17.* Text examples for `OLMoE-L14-E59`. Highlighted words are tokens routed to this expert which also received a high score.

Expert 59 is routed primarily on structurally common tokens such as conjunctions (e.g., and), prepositions (e.g., `of`, `into`), and other high-frequency connective words that occur in descriptive or explanatory passages. See Figure 17 for examples of this. Among these, the tokens receiving the highest activation scores are typically those that precede or connect segments rich in domain-specific content, indicating that the routing is sensitive to positions where specialized terminology is likely to follow. The tokens promoted by the expert are consistently highly specific and domain-bound, including abbreviations and jargon such as `instance`, `raid`, `CR`, `DR`, `spell`, `class` and `experience`, which are characteristic of role-playing game (RPG) mechanics and systems. The context in which this expert operates is therefore not general narrative text, but rather discussions involving structured gameplay, including combat mechanics, character progression, encounters, and system rules across both video games and tabletop RPGs.

The expert's role is to complete domain-specific RPG terminology by promoting related tokens. For example, in a list like `Demonology warlocks, Frost DKs and possibly Survival hunters` it suggests other class/spec terms such as `Ret` or `monks`. In rules-heavy contexts like `calculating CMB/CMD`, it promotes related system terms like `CR` or `Challenge`. When combat mechanics are mentioned (e.g., `bypass any DR`), it reinforces associated stats such as `AC` and `Damage`. It also links attributes to outcomes, promoting `hit` after `Strength and Constitution... maximum` and predicts terms like `instance` or `dungeon` from generic phrasing like `go into`. Overall, it acts as a domain-aware autocomplete for RPG mechanics and terminology.

*Table 7.* Specialization scores for `OLMoE-L14-E59` across different granularities ($k$).

| k | Routing Specialization (input) | Functional Specialization (output) |
|---|---|---|
| 10 | 0.028 | 0.106 |
| 50 | 0.054 | 0.146 |
| 100 | 0.060 | 0.156 |
| 1000 | 0.097 | 0.332 |
| 5000 | 0.102 | 0.417 |

Our specialization scores are consistent with this analysis in Table 7. The Routing specialization is very low since the routed tokens are mostly high-frequency connective words. The Functional specialization becomes only visible at high $k$ since RPG mechanics are a very niche topic. Overall, it is remarkable to find an entire expert being dedicated to RPG content, as the expert has to be hyper-specialized and concentrated on only a single topic.

## H.3. Asian and African Subword Detector

This expert is `OLMoE-L9-E60`. The label generated by the explainer model is: *"Proper names and locations from non-Western cultures, especially Asian and African."*. The scorer model achieved a F1 score of 0.88 on this hypothesis. See Figure 18 for text examples.

---

**Text Examples**

Minister Umar Naseer and current Police Commissioner Hussain Waheed commended Riyaz for his work. Home Minister Umar described Riy

on a tour of three African countries in 1997. Malawian governmental delegations led by Justin Malewezi to Malaysia in 2003 marvelled at the industrial

material published or available on GuruFocus.com, or relating to the use of, or inability to use, GuruFocus.com or any content,

---

*Figure 18.* Text examples for `OLMoE-L9-E60`. Highlighted words are tokens routed to this expert which also received a high score.

Expert 60 operates primarily in contexts containing proper nouns and transliterated foreign words, including personal names, place names, and organizational names from diverse linguistic regions (e.g., South Asian, African, Middle Eastern, and East Asian contexts). Tokens that are routed to this expert are typically subword fragments within these names, such as vowel-consonant clusters or recurring character sequences. The expert shows high activation scores on fragments like `iy`, `az`, `aw`, `kin`, and `awa`, which often appear inside distinctive orthographic patterns. The expert tends to promote other visually or structurally similar subword fragments, for instance, `iy` promoting `ahi` or `uru` promoting `ahan`, indicating a sensitivity to character-level similarity rather than semantic content.

Taken together, these patterns suggest that Expert 60 functions as a subword pattern detector, specializing in recognizing and generalizing recurring orthographic motifs within named entities and transliterated text.

*Table 8.* Specialization scores for `OLMoE-L9-E60` across different granularities ($k$).

| k | Routing Specialization (input) | Functional Specialization (output) |
|---|---|---|
| 10 | 0.112 | 0.033 |
| 50 | 0.122 | 0.049 |
| 100 | 0.165 | 0.056 |
| 1000 | 0.186 | 0.091 |
| 5000 | 0.176 | 0.130 |

Our specialization scores support this analysis in Table 8. The Routing specialization scores are among the highest in the entire model, indicating the selectivity of the router for this expert. The Functional Specialization stays relatively low as the predicted tokens are often diverse subwords.

# I. Mathematical Definition of Specialization Scores

We formalize the specialization scores used in Section 6.2. To quantify the degree of specialization for an expert $E_i$ in layer $L$, we measure the divergence between the expert's empirical distribution over vocabulary clusters and the aggregate distribution of the entire layer.

## I.1. Probability Distributions

Let $\mathcal{V}$ be the model's vocabulary and $C : \mathcal{V} \to \{1, \dots, k\}$ be a mapping that assigns each token to one of $k$ clusters. For a given expert $E_i$, we define its cluster distribution $P_i$ as a probability mass function over the $k$ clusters:

$$P_i(c) = \sum_{v \in \mathcal{V} : C(v) = c} f_i(v)$$

where $f_i(v)$ represents the relative frequency of token $v$ appearing in the context of expert $E_i$. In the case of **Routing Specialization**, $f_i(v)$ is derived from the tokens $x \in \mathcal{V}$ routed to $E_i$. In the case of **Functional Specialization**, $f_i(v)$ is derived from the top-$n$ tokens promoted by the expert's output vector $E_i(x)$ via Logit Lens projection.

We define the layer-wide **base rate** $Q_L$ as the expected distribution for any component within layer $L$. This is computed as the aggregate distribution of all experts in that layer:

$$Q_L(c) = \frac{1}{|\mathcal{E}_L|} \sum_{E_j \in \mathcal{E}_L} P_j(c)$$

where $\mathcal{E}_L$ is the set of all experts in layer $L$.

## I.2. Specialization Score

To measure the deviation of an expert from the layer's base rate, we use Jensen-Shannon Divergence (JSD). We compute the score $S_i \in [0, 1]$:

$$S_i = \text{JSD}(P_i \parallel Q_L) = \frac{1}{2} D_{KL}(P_i \parallel M) + \frac{1}{2} D_{KL}(Q_L \parallel M)$$

where $M$ is the midpoint distribution:

$$M = \frac{1}{2}(P_i + Q_L)$$

and $D_{KL}(P \parallel Q) = \sum_c P(c) \log_2 \frac{P(c)}{Q(c)}$ is the Kullback-Leibler divergence. A score of $0$ indicates that the expert is indistinguishable from the layer average, while a score of $1$ indicates a distribution that is entirely disjoint from the base rate.

## I.3. Random Expert Baseline

Because an expert's empirical distribution $P_i$ is estimated from a finite sample of $n_i$ tokens, a low sample count may cause a non-zero JSD even for a non-specialized expert. To isolate specialization from sampling variance, we calculate a **Random Expert Baseline** $\hat{S}_i$.

For an expert that has processed $n_i$ tokens, we define a random baseline expert $B_i$ whose cluster counts are drawn from a multinomial distribution parameterized by the layer-wide base rate $Q_L$:

$$\mathbf{X}_{B_i} \sim \text{Multinomial}(n_i, Q_L)$$

The baseline specialization score is then defined as the expected divergence of this random process:

$$\hat{S}_i = \mathbb{E}\left[\text{JSD}(\hat{P}_{B_i} \parallel Q_L)\right]$$

where $\hat{P}_{B_i}$ is the empirical distribution of the sampled counts $\mathbf{X}_{B_i}$. In our analysis and for our plots, we subtract this baseline from the raw score $S_i$ to ensure that our metrics reflect true specialization.

# J. Prompts

We present the prompt templates used for the explainer and scorer models in Section 5. It also includes the prompt we used to generate test cases for Section 5.3. We include them verbatim because small wording changes can affect label quality and evaluation behavior.

```
EXPLAINER SYSTEM PROMPT
<role>
You are an expert interpretability researcher analyzing a specific 'Expert' within a Mixture-of-Experts
(MoE) Transformer.
</role>

<task>
You will be provided with several text snippets (32 tokens long).
In each snippet, the specific Expert being analyzed was active for one or more tokens.
Your goal is to formulate a single, precise hypothesis explaining the computational role of this Expert.
The hypothesis should be a concise, one-sentence functional description of the expert's role (3-12 words).
</task>

<data_structure>
Each example consists of:
1. <snippet>: The raw text. Tokens routed to this expert are wrapped in double asterisks (e.g., **token**).
2. <top_activations>: A list of the top active tokens in that snippet (up to 5),
sorted by an importance score (Router Weight * Output L2 Norm).
- 'score': The obtained score for that token.
- 'token_str': The string representation.
- 'promoted_tokens': The top 3 tokens the expert predicted next (Logit Lens).
</data_structure>

<guidelines>
1. **Analyze Density:** Does the expert activate sporadically (specific entities) or
continuously (syntactic blocks)?
2. **Consult Logit Lens:** Use the 'promoted_tokens' to understand the *effect* of the expert.
If an expert activates on 'New', and promotes 'York', 'Zealand', 'Jersey', it is a named-entity completer.
3. **Generalize:** Do not overfit to a single example.
Find the common thread across all examples.
4. **Formatting:** Ignore the `**` markers when analyzing the natural flow of text;
they are only for highlighting.
</guidelines>
```

```
EXPLAINER USER PROMPT
<context>
Here are the maximal activating examples for Expert 17.
</context>

<data>
<example id="1">
<snippet>
and whistles" you need to create more complex** quizzes** and** surveys**.
These** are** the** features** used by some of** the** world's most popular** quizzes****,
**** diagnostics****,**** and**
</snippet>
<top_activations>
<item token_str="diagnostics" score="4.30" promoted_tokens="Repeat, ozo, repeat"/>
<item token_str="quizzes" score="2.39" promoted_tokens="Published, qb, visitor"/>
<item token_str="surveys" score="2.01" promoted_tokens="ritt, æīĵ, anging"/>
<item token_str="quizzes" score="1.93" promoted_tokens="markup, è®°å½·, quiz"/>
<item token_str="," score="1.41" promoted_tokens="éK̲L̲, corner, ol"/>
</top_activations>
</example>

<example id="2">
<snippet>
```

```
, Mahndra New** Cars**, Tata** New**** Cars**...

My**Car****D****ek****ho** is India's most popular** website** for** new**** car**** pricing**.
** New**** Cars**** details****,****New**
</snippet>
<top_activations>
<item token_str="pricing" score="4.29" promoted_tokens="ä¸Gèµ·, ourselves, èkĹ"/>
<item token_str="Cars" score="1.68" promoted_tokens="airs, æ´¢, nak"/>
<item token_str="details" score="1.59" promoted_tokens="iron, Brom, experimenting"/>
<item token_str="car" score="1.55" promoted_tokens="bl, è¶h, draft"/>
<item token_str="new" score="1.21" promoted_tokens="åį¤, è·Lç¦», èĥ½èåĪ°"/>
</top_activations>
</example>
...
</data>

<instruction>
Based strictly on the data above, analyze the <top_activations> and their context in the <snippet>.
Generate your <hypothesis> now.
</instruction>
```

```
SCORER SYSTEM PROMPT
<role>
You are an automated evaluator for interpretability hypotheses.
</role>

<task>
You will be given:
1. A **Hypothesis** describing the function of a specific MoE Expert.
2. A list of **Test Examples**. Each example contains a text snippet,
where active tokens are highlighted with double asterisks (e.g., **token**).

Your job is to determine: **Does the highlighted token pattern in the example match the Hypothesis?**
- If the highlighted tokens fit the hypothesis description: Output 1.
- If the highlighted tokens clearly violate the hypothesis or are unrelated: Output 0.
</task>

<constraints>
- You must evaluate strictly based on the provided Hypothesis.
- You must verify that the **Hypothesis specifically describes the **BOLDED tokens**,
not just the general topic of the sentence.
</constraints>
```

```
SCORER USER PROMPT
<hypothesis>
Nouns and technical terms within descriptive product, tool, or software metadata.
</hypothesis>

<examples>

<example id="1">
<snippet>
hmottestad
Ruter**

**It's the*** app** for** checking**** bus****/****boat****/sub****way**/tr**am****
tim****et****ables**** for**** Oslo** (where I
live in
</snippet>
</example>

<example id="2">
<snippet>
```

```
avis Poker** Timer**** is**** a** great** looking**** tournament**** poker**** timer**** for****
Windows** and** Mac**** OS**** X**** with**** the** emphasis on** ease**** of**** use**.
*** Setup** and** manage**** your**** game**** in** a
</snippet>
</example>

<example id="3">
<snippet>
website** design****.

**A** logo**** and**** favicon** can** be**** uploaded** in the second tab.** A**** text****
logo**** can** also** be**** chosen** if** you**** do** not want to use** an**** image**.
</snippet>
</example>
...
<instruction>
Evaluate the 20 examples above against the hypothesis.
First, perform your analysis.
Then, output the final list. Ensure it contains exactly 20 integers.
</instruction>
```

```
Prompt used to generate examples for causal attribution experiment in Section 5.3

Your task is to generate JSON examples for a specific label.
The label is the result of interpreting a Mixture-of-Experts (MoE) expert in an LLM,
it describes what the expert's computational role is in the model.

The JSON file should contain exactly 20 examples with the following structure:
{
"text": "A short text snippet where you would expect the MoE expert to be active.",
"trigger": "A single word, subword or a combination of adjacent words, where the expert is likely routed to.",
"target": "A target word that based on the label is likely being predicted or promoted by the expert"
}

The "trigger" needs to be a word that has a high probability of being routed to the expert and the "target"
needs to be a word for which the expert must be highly responsible, either by directly predicting it
or by promoting it implicitly.

Here is the label for the MoE expert:
"Proper names and locations from non-Western cultures, especially Asian and African."

Please generate the JSON file now.
```

# K. Automatic Interpretability Labels

We list all automatically generated expert labels for the three models we analyzed: `ERNIE-4.5-21B-A3B` (Table 9), `OLMoE-1B-7B` (Table 10), and `Qwen3-30B-A3B` (Table 11).

*Table 9.* `ERNIE-4.5-21B-A3B` Expert labels.

| | ERNIE-4.5-21B-A3B | | |
|---|---|---|---|
| **L-E** | **Label** | **L-E** | **Label** |
| L4-E0 | Activates on specific keywords used in configuration, header files, and build constraints. | L4-E1 | Sub-word segments within technical, medical, and programming-related multi-token terms. |
| L4-E2 | Syntactic delimiters and numerical separators in code, math, and metadata formats. | L4-E3 | Scientific suffixes and technical word fragments in biology, coding, and linguistics. |
| L4-E4 | Sub-word segments within chemical, geological, and alphanumeric identifiers. | L4-E5 | Mid-word tokens within multisyllabic proper nouns and specialized technical terms. |
| L4-E6 | Software components, methods, and types containing common suffixes like View, Table, or Tag. | L4-E7 | Syntactic punctuation and internal word segments in structured data and categories. |
| L4-E8 | Activates on initial nouns and punctuation in structured text or lists. | L4-E9 | Mid-word sub-tokens in multisyllabic scientific, geographical, and proper nouns. |
| L4-E10 | Software keywords and internal word fragments in programming and technical terms. | L4-E11 | Sub-word segments within technical identifiers and programming library names. |
| L4-E12 | Technical punctuation and alphanumeric suffixes in code, math, and acronyms. | L4-E13 | Mid-to-end word fragments of proper nouns, especially in international names and locations. |
| L4-E14 | Subword segments within chemical names, technical acronyms, and command-line keywords. | L4-E15 | Metadata and configuration tags in markup, code, and file system paths. |
| L4-E16 | Syntactic symbols and formatting markers in code, formulas, and technical equations. | L4-E17 | Tokens within technical strings like file paths, URLs, and mathematical expressions. |
| L4-E18 | Activates on specific suffixes within medical, scientific, technical, and proper nouns. | L4-E19 | Completes specific multi-token proper nouns, technical identifiers, and biological terms. |
| L4-E20 | Syntactic delimiters and keywords in programming and mathematical markup languages. | L4-E21 | Sub-word segments within proper nouns, specialized terminology, and legal symbols. |
| L4-E22 | Mathematical instruction keywords like Rearrange, Collect, and Express in algebra problems. | L4-E23 | Syntactic assignment and definition operators in various programming and configuration languages. |
| L4-E24 | Sub-word segments within taxonomic names and the Wikipedia metadata term 'created'. | L4-E25 | Capitalized technical keywords, identifiers, and protocols in source code and structured data. |
| L4-E26 | Newlines and indentations at the start of structured text blocks. | L4-E27 | Scientific suffixes, Latin technical terms, and Wikipedia disambiguation links. |
| L4-E28 | Mid-word subunits in specialized proper nouns, technical terms, and programming keywords. | L4-E29 | Activates on assignment operators and specific technical name suffixes in configuration strings. |
| L4-E30 | Mid-word tokens following 'S' in proper nouns and scientific terms. | L4-E31 | Morphemes within scientific, technical, or complex multi-token names and identifiers. |
| L4-E32 | Mid-word tokens within technical terms, URLs, and proper nouns. | L4-E33 | Sub-word segments within proper names, technical terms, and rare multisyllabic words. |
| L4-E34 | Activates on specific sub-word units within scientific names, brands, and technical terms. | L4-E35 | Coding syntax markers including comment delimiters, build tags, and test keywords. |
| L4-E36 | Technical or formal descriptive identifiers and their subsequent definitional continuations. | L4-E37 | Syntactic punctuation and operators in programming, configuration, and markup languages. |
| L4-E38 | Mid-word syllables in proper nouns and technical terms. | L4-E39 | Mid-word tokens in scientific, technical, or proper names beginning with 'M'. |
| L4-E40 | Mid-to-end word segments in multi-token technical terms and compound words. | L4-E41 | Acronyms, technical identifiers, and proper nouns' internal or final sub-tokens. |
| L4-E42 | Suffixes and character sequences in code, scientific names, and legal citations. | L4-E43 | Periods in abbreviations, legal citations, decimals, and URLs. |
| L4-E44 | Mid-word syllables and suffixes in technical, biological, and specialized proper names. | L4-E45 | Structural markers signaling the start of metadata or reference categories. |
| L4-E46 | Legal case citation separators, especially the adversarial indicator 'v' or 'vs'. | L4-E47 | Plural nouns and naming conventions in software development and technical configurations. |
| L4-E48 | Numerical components of version numbers, dates, and identifiers, particularly trailing digits. | L4-E49 | Activates at the start of new lines in lists and biographies. |
| L4-E50 | Mid-word fragments and technical symbols in code, markup, and identifiers. | L4-E51 | Uppercase acronym suffixes, technical abbreviations, and specific numeric sequence components. |
| L4-E52 | Mid-word morphemes within complex proper nouns, scientific terms, and specialized brands. | L4-E53 | Syntactic keywords and structural components in programming code and technical identifiers. |
| L4-E54 | Hyphens and punctuation within structured citation keys and code configuration strings. | L4-E55 | Newlines and structural markers separating translated text, math, or code blocks. |
| L4-E56 | Sub-word segments within scientific names, technical terminology, and complex proper nouns. | L4-E57 | Mid-word tokens in multi-syllabic biological, technical, or programming identifiers. |
| L4-E58 | Common word segments and sub-tokens in specialized names, lists, or identifiers. | L4-E59 | Software code attributes, property names, and method identifiers in structured text. |
| L4-E60 | Mid-word tokens within proper nouns, scientific terms, and specialized identifiers. | L4-E61 | Tokens within structured alphanumeric strings, technical identifiers, and punctuation-heavy text segments. |
| L4-E62 | Mathematical and scientific terminology components, including factors, digits, and biological sub-words. | L4-E63 | Technical terms and identifiers in code, configuration files, and metadata. |
| L15-E0 | Syntactic coordination and logical continuation sequences using conjunctions like 'and' or 'or'. | L15-E1 | Syntactic transition points in mathematical notation, code, and narrative descriptions. |
| L15-E2 | Punctuation and conjunctions that introduce subordinate or coordinate clauses. | L15-E3 | Nouns and phrases specifying physical, geographic, or system-internal locations. |
| L15-E4 | Adjectives and nouns describing scale, quantity, intensity, or magnitude. | L15-E5 | Adverbs and auxiliary verbs expressing probability, frequency, and typicality in general statements. |

## ERNIE-4.5-21B-A3B

| L-E | Label | L-E | Label |
|---|---|---|---|
| L15-E6 | Activates primarily on the preposition 'for' and digits in mathematical expressions. | L15-E7 | Nouns and concepts related to human biology, health, and social structures. |
| L15-E8 | Tokens indicating digital presence or spatial positioning within specific contexts. | L15-E9 | Verbs and processes related to data processing, administrative management, and technical workflows. |
| L15-E10 | Configuration keys and named parameters in code and formal descriptions. | L15-E11 | Adjectives and modifiers describing specific characteristics, rankings, or physical properties. |
| L15-E12 | Tokens expressing capability, intention, or requirement within infinitive and modal structures. | L15-E13 | Function words and prepositions in formal, academic, or mathematical contexts. |
| L15-E14 | Identifies proper nouns and specific surnames in attribution lines or legal documents. | L15-E15 | Numerical values and logical operators in programming code and mathematical equations. |
| L15-E16 | Numerical digits and individual characters within mathematical expressions and structured identifiers. | L15-E17 | Unique identifiers, hex strings, and possessive suffixes in structured data. |
| L15-E18 | Newlines and delimiters separating menu items, headers, and metadata in web layouts. | L15-E19 | Tokens within phrases expressing restrictive, cautionary, or definitive conditions and prohibitions. |
| L15-E20 | Tokens involving attendance, participation, or travel to specific events and locations. | L15-E21 | Proper nouns including geographical locations, technical systems, and political entities. |
| L15-E22 | Mathematical and structural symbols used as delimiters, operators, or naming prefixes. | L15-E23 | Activates on prepositions and pronouns, especially the word 'at' in prepositional phrases. |
| L15-E24 | Activates on punctuation and whitespace terminating logical segments across code and prose. | L15-E25 | Sequences involving structured data, mathematical expressions, and passive auxiliary verb constructions. |
| L15-E26 | Subword segments and formatting characters in academic titles, medical terms, and identifiers. | L15-E27 | Proper nouns and terminology within administrative, geographical, chemical, and legal lists. |
| L15-E28 | Conditional clauses and scenarios introduced by if, when, such as, or provided that. | L15-E29 | Attributing information to sources such as articles, forum posts, or comments. |
| L15-E30 | Identifies and processes algebraic and programming variable names in equations and code. | L15-E31 | Adverbs and phrases emphasizing simplicity or exclusivity in instructions and technical definitions. |
| L15-E32 | Nouns and suffixes identifying professional roles, occupations, or organizational groups. | L15-E33 | Mandatory directives, instructions, and conditional advice in technical or formal contexts. |
| L15-E34 | Syntactic delimiters and separators in URLs, mathematical expressions, and citations. | L15-E35 | Completing comparative phrases and logical analogies using 'as' or 'like'. |
| L15-E36 | Nouns denoting abstract resources, data, or collective entities. | L15-E37 | Informational or commercial call-to-action phrases and service-related descriptive text. |
| L15-E38 | Nouns and structural keywords in formal business, technical, and biological descriptions. | L15-E39 | Punctuation, mathematical symbols, and delimiters within technical or formal contexts. |
| L15-E40 | Technical terms and components in programming code, stack traces, and chemical nomenclature. | L15-E41 | Syntactic structures that introduce clarifications, explanations, lists, or specific details. |
| L15-E42 | Proper nouns of commercial organizations, publishers, and sponsored sporting venues. | L15-E44 | Connective prepositions and verbs introducing detailed descriptions, relationships, or mathematical contexts. |
| L15-E45 | Syntactic phrases introducing specific roles, groups of people, or interrogative subjects. | L15-E46 | Tokens that express comparative, relational, or contrastive states between different entities. |
| L15-E47 | Nouns and suffixes in plural or technical contexts. | L15-E48 | Metadata field labels and navigation headers in structured or web-based text. |
| L15-E49 | Function words like 'of', 'without', 'if', and 'any' in formal or mathematical contexts. | L15-E50 | Adverbs and verbs ending in common suffixes like -ly, -ed, or -ing. |
| L15-E51 | Tokens following commas in lists of nouns, verbs, or numbers. | L15-E52 | Common adverbial and conjunctional transitions at the start of response sentences. |
| L15-E53 | Syntactic functional elements within legal disclaimers, technical reports, and boilerplate website information. | L15-E54 | Syntactic structures expressing logical explanation, definition, or significance after a demonstrative pronoun. |
| L15-E55 | Activates on logical conjunctions, disjunctions, and punctuation separating alternative or descriptive terms. | L15-E56 | Processes descriptive parenthetical metadata and standardized legal or technical citations. |
| L15-E57 | Niche compound nouns and technical terminology across diverse specialized domains. | L15-E58 | Mid-word letter combinations in proper nouns, scientific terms, and specialized names. |
| L15-E59 | Technical, statistical, and categorical nouns in formal data reporting contexts. | L15-E60 | Activates on definite/indefinite articles and descriptive adjectives within noun phrases. |
| L15-E61 | Activates on logical connectives and verbs within error messages and formal reports. | L15-E62 | Verbs and particles describing redirection, cessation, or dynamic changes in states. |
| L15-E63 | Structural breaks in documents, including section headers, separator lines, and metadata fields. | L25-E0 | Activates on numbers and quantifiers to predict subsequent units like letters or touchdowns. |
| L25-E1 | Syntactic delimiters and structural markers in code, math, and academic documents. | L25-E2 | Predicts specialized measurement and property terminology in scientific or technical contexts. |
| L25-E3 | Activates on 'j' or 'J' to predict subsequent word-forming letters. | L25-E4 | Completes specific prefixes and entity fragments in multilingual and encoded strings. |
| L25-E5 | Predicts specialized nouns in formal, legal, and technical transactional contexts. | L25-E6 | Predicts regional location and entity specific suffixes in structured geographical and organizational data. |
| L25-E7 | Prepositions and conjunctions preceding specific locations, temporal states, or fixed idioms. | L25-E8 | Transitive verbs and auxiliary verbs predicting their direct objects or complements. |
| L25-E9 | Tokens that form the first part of common compound nouns or hyphenated phrases. | L25-E10 | Syntactic completion of technical dependencies, software environments, and scientific database references. |
| L25-E11 | Scientific and technical multi-word terms, predicting subsequent components or methodological suffixes. | L25-E12 | Variables in algebraic expressions and Japanese honorific or quantifier prefix segments. |
| L25-E13 | Predicts nouns completing common idiomatic phrases, collocations, or multi-word expressions. | L25-E14 | Numerical digits in sequences, mathematical expressions, and specific naming/temporal identifiers. |
| L25-E15 | Tokens that introduce an agreement, requirement, or intent for future action. | L25-E16 | Predicts specialized technical or scientific suffix completions and compound word components. |
| L25-E17 | Syntactic delimiters and numeric sequences in code, mathematical expressions, and citations. | L25-E18 | Activates on major academic section titles and programming class or property identifiers. |
| L25-E19 | Tokens starting with 'ag' or 'mag' across multiple languages and contexts. | L25-E20 | Predicting specific continuation components of numeric citations, legal references, and mathematical equations. |
| L25-E21 | Activates on interrogative and existential markers to predict following functional modifiers. | L25-E22 | Predicts subsequent action verbs for subjects in biographical or narrative contexts. |

## ERNIE-4.5-21B-A3B

| L-E | Label | L-E | Label |
|---|---|---|---|
| L25-E23 | Proper nouns and taxonomic names, specifically focusing on first names and genus components. | L25-E24 | Scientific and mathematical notation components that trigger taxonomic or symbolic continuations. |
| L25-E25 | Tokens that introduce an infinitive phrase expressing purpose or intended action. | L25-E26 | Tokens starting with 'ac' or 'K' and related phonetic sub-fragments. |
| L25-E27 | Activates on transition points, often preceding proper nouns or function words. | L25-E28 | Mathematical and code-based delimiters for dates, formulas, and string formatting parameters. |
| L25-E29 | Syntactic delimiters and repeated punctuation used for structural formatting and item continuation. | L25-E30 | Predicts subsequent components in complex noun phrases describing materials, weaponry, and specialized equipment. |
| L25-E31 | Numerical sequences and school-related contextual phrases often involving student classification. | L25-E32 | Activates on locations or organizational entities to predict their geographic state or province. |
| L25-E33 | Completes words starting with 'al', 'sl', 'el', 'gl', or 'il' stems. | L25-E34 | Predicts specialized anatomical, mathematical, or geographical terms following definitive descriptive tokens. |
| L25-E35 | Activates on punctuation ending mathematical definitions to prompt procedural instructions. | L25-E36 | Predicts specialized organizational bodies or professional roles within specific institutional contexts. |
| L25-E37 | Activates on 'at', 'ot', or 'T' word-initial fragments and prefixes. | L25-E38 | Predicts comparison or equality operators following variables and objects in code. |
| L25-E39 | Predicts sensory or physical descriptive adjectives following linking verbs or prepositions. | L25-E40 | Predicts subsequent procedural verbs in formal, technical, or legal process descriptions. |
| L25-E41 | Technical measurements and specifications, particularly numbers followed by units of measure. | L25-E42 | Mathematical operators and structural words followed by negative numbers or math-related terms. |
| L25-E43 | Activates on plural nouns referring to groups of people or entities. | L25-E44 | Mathematical operators and punctuation within complex numeric, algebraic, or URL-encoded sequences. |
| L25-E45 | Activates on the uppercase letter 'G' when starting proper nouns and names. | L25-E46 | Sub-word segments within technical, chemical, transliterated, or code-related nomenclature. |
| L25-E47 | Linking verbs or modal phrases predicting evaluative adjectives like useful, impossible, or clear. | L25-E48 | Numerical digits within structured contexts like times, dates, and mathematical expressions. |
| L25-E49 | Activates on function words preceding entities in specific geographic or categorical contexts. | L25-E50 | Predicts conditional or causal conjunctions following specific noun or verb completions. |
| L25-E51 | Mathematical nouns and descriptors in word problems and technical legal notices. | L25-E52 | Syntactic subjects or auxiliary verbs predicting the following main verb or effect. |
| L25-E53 | Predicting specific technical terms within legal, medical, and categorical lists. | L25-E54 | Tokens that precede descriptive adjectives or psychological state completions. |
| L25-E55 | predicting structural components of legal disclaimers, licenses, and document descriptions | L25-E56 | Processes spatial and positional relationships to predict directional or locational modifiers. |
| L25-E57 | Completes specific multi-token stems like 'Qu-', 'Br-', and 'Sch-' into longer words. | L25-E58 | Tokens starting with 'An' or 'an' across multiple languages and medical terms. |
| L25-E59 | Technical identifiers, specific vehicle models, and scientific classification nomenclature. | L25-E60 | Numerical values and delimiters in mathematical, statistical, and legal citations. |
| L25-E61 | Predicts completion of stems into multi-syllabic words, often ending in suffixes. | L25-E62 | Syntactic punctuation and prepositions that initiate attribution, structural metadata, or functional transitions. |
| L25-E63 | Structural delimiters and line breaks following block closures in code and documentation. | | |

*Table 10.* `OLMoE-1B-7B` Expert labels.

**OLMoE-1B-7B**

| L-E | Label | L-E | Label |
|---|---|---|---|
| L1-E0 | Adjectival and adverbial suffixes in technical, medical, and scientific terminology. | L1-E1 | Tokens forming the suffixes of identifiers, proper names, and code-based variables. |
| L1-E2 | Mid-word and terminal suffixes within proper nouns, brands, and technical terms. | L1-E3 | Common word endings and suffixes, particularly those in scientific and proper nouns. |
| L1-E4 | Mid-word and terminal morphemes in technical, scientific, and multi-part proper nouns. | L1-E5 | Mid-word tokens and proper noun components across various specialized domains. |
| L1-E6 | Sub-word segments within proper nouns, surnames, and specialized terminology. | L1-E7 | Mid-word tokens within proper nouns, specialized terminology, or technical abbreviations. |
| L1-E8 | Mid-word tokens within proper nouns, technical terms, and complex identifiers. | L1-E9 | Activates on articles 'a', 'an', and 'the' to predict list continuations. |
| L1-E10 | Proper nouns, particularly surnames following common first names like Robert and David. | L1-E11 | Activates on specific stems or fragments, predicting common suffixes like 'Byr', 'Bes', and 'Isa'. |
| L1-E12 | Suffixes and morphemes within technical, medical, and formal organizational terms. | L1-E13 | Activates on common multi-token word endings in names and technical terms. |
| L1-E14 | Mid-word morphemes or syllables within proper nouns and specialized terminology. | L1-E15 | Markup and code attributes within technical documentation, metadata, and web headers. |
| L1-E16 | Mid-word fragments in scientific terms, medical conditions, and proper names. | L1-E18 | Activates on initial sequence tokens like numbers, URLs, and code block starts. |
| L1-E19 | Mid-word tokens and final components of proper nouns or technical terms. | L1-E20 | Mid-word sub-tokens and morphemes containing 'ant', 'and', or 'ribution'. |
| L1-E21 | Common words, names, and technical terms used in lists and biographies. | L1-E22 | Sub-word suffixes including ist, or, ner, ism, and ky. |
| L1-E23 | Scientific and technical suffixes, particularly biological and medical terminology word endings. | L1-E24 | Mid-word and end-word fragments in proper nouns and technical terms. |
| L1-E25 | Syntactic keywords and structural components in programming code and database queries. | L1-E26 | Technical sub-word components within namespaces, file paths, URLs, and specialized terminology. |
| L1-E27 | Mid-word tokens in proper nouns, particularly Dutch names and scientific terminology. | L1-E28 | Mid-word tokens in proper nouns, technical terms, and non-English names. |
| L1-E29 | Proper nouns and surnames, especially after first names or titles. | L1-E30 | Completes mid-word or multi-part proper nouns and technical terms. |
| L1-E31 | Proper nouns and technical terms split across multiple tokens. | L1-E32 | Sub-word segments within technical, medical, and scientific terminology. |
| L1-E33 | Mid-word letter clusters within medical terminology and non-English proper names. | L1-E34 | Proper names, technical identifiers, and specialized morphological sub-tokens. |
| L1-E35 | Sub-word segments within proper names, technical terms, and academic classifications. | L1-E36 | Tokens representing specific suffixes or word fragments, particularly 'y' and 'le'. |
| L1-E37 | Mid-word word fragments and suffixes in specialized terminology or proper names. | L1-E38 | Activates on internal and final syllables of proper nouns and technical terms. |
| L1-E39 | Tokens forming suffixes or components of proper nouns and technical terms. | L1-E40 | Suffixes and particles ending in 'd', 'g', or 'p'. |
| L1-E41 | Proper nouns, technical terms, and name segments in diverse contexts. | L1-E42 | Tokens forming parts of common compound terms in programming and formal naming. |
| L1-E43 | Mid-word sub-tokens in complex names, technical terms, and non-English words. | L1-E44 | Sub-word segments and suffixes within technical, scientific, or proper names. |
| L1-E45 | Tokens forming plural endings or common suffixes in academic and technical contexts. | L1-E46 | Proper nouns and alphanumeric identifiers within scientific, biographical, and technical contexts. |
| L1-E47 | Scientific and mathematical terms, symbols, and unit abbreviations within technical contexts. | L1-E48 | Proper names and surnames, especially following first names like Richard, Andrew, or Alan. |
| L1-E49 | Mid-word fragments and suffixes within polysyllabic nouns, verbs, and technical terms. | L1-E50 | Tokens forming the common Latin-derived suffix 'us' in scientific, technical, or proper names. |
| L1-E51 | Mid-word fragments and syllables within proper nouns, technical terms, and suffixes. | L1-E52 | Sub-word suffixes like 'on', 'in', and 'ator' in technical and proper nouns. |
| L1-E53 | Mid-word letter sequences in specific proper nouns, technical terms, and common determiners. | L1-E54 | Common word endings and morphological suffixes like -ous, -bright, -ling, and -ged. |
| L1-E55 | Mid-word morphemes and suffixes within technical, biological, and scientific terminology. | L1-E56 | Mid-word letter combinations and abbreviations in technical, legal, or specialized identifiers. |
| L1-E57 | Common chemical and biological word suffixes, especially -amine, -ine, and -ium. | L1-E58 | Proper nouns and capitalized name components within lists or formal contexts. |
| L1-E59 | Scientific and technical terminology within medical, biological, and software licensing contexts. | L1-E60 | Tokens concluding polite phrases and metadata fields, often preceding technical answers. |
| L1-E61 | Mid-word fragments in complex medical, scientific, and multilingual terminology. | L1-E62 | Mid-word tokens in compound words, technical terms, and proper nouns. |
| L1-E63 | Common word suffixes in proper nouns, particularly surnames and location names. | L4-E0 | Abstract concepts and terminology related to psychology, religion, and philosophy. |
| L4-E1 | Tokens relating to family members, domestic relationships, and personal life. | L4-E2 | Newlines and line breaks within lists, menus, or structured records. |
| L4-E3 | Activates on terms and identifiers within legal, patent, and formal citations. | L4-E4 | Activates on internal syllables of specialized acronyms, technical terms, and code-related identifiers. |
| L4-E5 | Syntactic function words and relative pronouns in English and German clauses. | L4-E6 | Syntactic punctuation and symbols in code, scripts, diffs, and formatted timestamps. |
| L4-E7 | Prepositions and parts of phrasal verbs within formulaic or idiomatic expressions. | L4-E8 | Adjectives and quantifiers modifying noun phrases to describe attributes or scales. |
| L4-E9 | Scientific suffixes and descriptive terminology in biological, chemical, and luxury contexts. | L4-E10 | Proper nouns and entity titles, especially surnames, sports teams, and locations. |
| L4-E11 | Tokens within the titles of video games, movies, and entertainment series. | L4-E12 | Proper nouns and titled entities, specifically movie titles, locations, and names. |
| L4-E13 | Abstract verbs and nouns describing processes, actions, or outcomes. | L4-E14 | Activates on indefinite and definite articles, predicting a following noun or adjective. |
| L4-E15 | Sub-word segments within proper nouns, especially in scientific citations and historical names. | L4-E16 | Mid-word tokens in specialized scientific, historical, or academic proper nouns. |
| L4-E17 | Technical terms in mathematics, computer science, and physics contexts. | L4-E18 | Conversational filler and meta-commentary in forum posts, emails, and online discussions. |

## OLMoE-1B-7B

| L-E | Label | L-E | Label |
|---|---|---|---|
| L4-E19 | Technical terms and keywords in web development, programming, and configuration contexts. | L4-E20 | Adjectives and descriptive terms relating to personal traits, identities, and physical properties. |
| L4-E21 | Numerical values and variables within scientific formulas, equations, and technical identifiers. | L4-E22 | Adverbs and adjectives describing temporal duration, state consistency, or relative comparison. |
| L4-E23 | Syntactic connectors and punctuation that link descriptive clauses, entities, or parentheticals. | L4-E24 | Recognizes transitional multi-word phrases and logical connectors across multiple languages. |
| L4-E25 | Syntactic terminators and delimiters in code and structured data formats. | L4-E26 | Identify descriptive prepositional phrases explaining spatial locations or technical objectives. |
| L4-E27 | Auxiliary verbs in their contracted negative forms, typically before an apostrophe. | L4-E28 | Proper nouns and specialized compound words across various niche domains. |
| L4-E29 | Technical terms and jargon related to mechanical engineering and specialized equipment. | L4-E30 | Tokens expressing negation, absence, or restriction across multiple languages. |
| L4-E31 | Technical terminology in medical research papers and clinical study methodologies. | L4-E32 | Proper nouns and technical terms in biochemistry, chemistry, and literature. |
| L4-E33 | Taxonomic citations, specifically biological nomenclature including authors, years, and punctuation. | L4-E34 | Specific model names and alphanumeric designations for vehicles and consumer electronics. |
| L4-E35 | Nouns and suffixes defining physical, geographical, or structural locations and entities. | L4-E36 | Verbs describing physical actions, movements, and dynamic transitions. |
| L4-E37 | Proper nouns and entity names, particularly focusing on people, locations, and titles. | L4-E38 | Predicative phrases describing the nature, necessity, or status of a subject. |
| L4-E39 | Syntactic structures involving repeated phrases, lists, and metadata line breaks. | L4-E40 | Tokens referring to people, including pronouns and collective nouns for human groups. |
| L4-E41 | Colons and subsequent newlines in Wikipedia disambiguation pages and structured metadata. | L4-E42 | Punctuation and conjunctions transitioning to the start of new independent clauses. |
| L4-E43 | Syntactic categories and linguistic descriptors within dictionaries, grammatical analyses, or definitions. | L4-E44 | Tokens involving legal evidence, factual claims, and logical justifications for conclusions. |
| L4-E45 | Nouns and verbs related to media releases, entertainment performances, and events. | L4-E46 | Scientific and statistical terminology within academic research abstracts and data reporting. |
| L4-E47 | General functional and technical vocabulary across diverse semantic domains and contexts. | L4-E48 | Imperative calls to action and direct instructions to the reader. |
| L4-E49 | Technical and procedural multi-word terms in legal, financial, and sporting contexts. | L4-E50 | Passive verb constructions followed by prepositions like by, under, or from. |
| L4-E51 | Syntactic connectors and delimiters in set phrases, including parenthetical numbers and hyphenated compounds. | L4-E52 | Tokens involving reciprocity or plurality, especially the word 'each' before 'other'. |
| L4-E53 | Tokens describing intense emotional reactions, sensory experiences, or physical expressions of feeling. | L4-E54 | Transitional segments including punctuation, speaker changes, and structural boundaries. |
| L4-E55 | Tokens within system configuration paths, technical parameters, and computer science terminology. | L4-E56 | Nouns denoting abstract concepts, processes, or entities within definite and possessive noun phrases. |
| L4-E57 | Infinitive and gerund phrases following verbs or prepositions like 'to' or 'for'. | L4-E58 | Adverbial and prepositional phrases describing time, location, or manner of action. |
| L4-E59 | Activates on punctuation and conjunctions that introduce or separate logical clauses. | L4-E60 | Auxiliary verbs and phrasal verbs within multi-word predicate structures. |
| L4-E61 | Punctuation separating items in code, data structures, and parenthetical lists. | L4-E62 | Nouns and terminology within specialized military, legal, governmental, or sports contexts. |
| L4-E63 | Metadata and structural elements like line breaks, punctuation, and forum headers. | L7-E0 | Technical metadata and structured parameters across code, scientific protocols, and legal headers. |
| L7-E1 | Tokens within repetitive, SEO-heavy, or poorly translated promotional text and punctuation. | L7-E2 | Syntactic structure delimiters and punctuation in programming and markup languages. |
| L7-E3 | Identify family members and relationships to predict domestic or reproductive outcomes. | L7-E4 | Numerical values and identifiers within code, mathematical notation, and data formats. |
| L7-E5 | Compound nouns and technical terms related to institutions, science, or geography. | L7-E6 | Tokens relating to persistence, redundancy, or suboptimal resource usage in task completion. |
| L7-E7 | Religious and biblical terminology, especially within scriptural passages and theological discussions. | L7-E8 | Phrases expressing potentiality, future likelihood, or conditional outcomes. |
| L7-E9 | Physical sensations, bodily conditions, and descriptions of abstract experiential states. | L7-E10 | Conditional and clarifying conjunctions or punctuation used to define exceptions or alternatives. |
| L7-E11 | Legal and academic boilerplate phrases regarding judicial reviews, citations, and procedural findings. | L7-E12 | Syntactic delimiters and sub-word units in URLs, file paths, and academic citations. |
| L7-E13 | Newlines and metadata separators within structured lists, headers, and product descriptions. | L7-E14 | Verbs in physical or metaphorical transitional states and their following prepositions. |
| L7-E15 | Adverbs and verbs describing functional processes, modifications, or quantitative changes. | L7-E16 | Plural nouns and tokens ending in 's' or pluralizing suffixes. |
| L7-E17 | Syntactic elements and identifiers in structured code and database query declarations. | L7-E18 | Tokens related to the availability, scheduling, and distribution of professional content. |
| L7-E19 | Prepositions and conjunctions introducing phrases that qualify or transition context. | L7-E20 | Tokens expressing purpose, intent, or consequence within infinitive and conditional clauses. |
| L7-E21 | Metadata delimiters, scientific units, and structural separators in structured text formats. | L7-E22 | Adjectives and descriptive phrases that precede specific nouns they characterize. |
| L7-E23 | Punctuation marks and line breaks that transition between distinct questions and answers. | L7-E24 | Syntactic function words, primarily 'of', in formal or descriptive phrases. |
| L7-E25 | Physical consumer products and materials in descriptive or instructional contexts. | L7-E26 | Nouns denoting organizational units, structural divisions, or formal categories. |
| L7-E27 | Non-ASCII characters and punctuation in multi-lingual or technical contexts. | L7-E28 | Informal conversational interjections and colloquial expressions in direct address. |
| L7-E29 | Numerical lists and paired values frequently followed by the adverb 'respectively'. | L7-E30 | Mid-sentence informational tokens following structural keywords or technical specifications. |
| L7-E31 | Proper nouns and entity segments followed by their organizational or geographic categories. | L7-E32 | Adjectives and adverbs describing scalar magnitude, intensity, or comparative measurements. |
| L7-E33 | Punctuation marks and separators within URLs, file paths, and bibliographic citations. | L7-E34 | Indefinite articles and quantifiers within common idiomatic or descriptive phrases. |

## OLMoE-1B-7B

| L-E | Label | L-E | Label |
|---|---|---|---|
| L7-E35 | Syntactic delimiters and structural whitespace within structured metadata, URLs, and code. | L7-E36 | Tokens referring to previously mentioned or following figures, sections, and hypotheses. |
| L7-E37 | Proper nouns and descriptive entities within sporting, temporal, and biographical contexts. | L7-E38 | Relational and directional prepositions and verbs in descriptive or encyclopedic statements. |
| L7-E39 | Colons and newline characters used as delimiters in structured list-based text. | L7-E40 | Contextual indicators of historical significance, record-breaking, and superlative duration. |
| L7-E41 | Punctuations and conjunctions signaling transitions in complex sentences or comparative structures. | L7-E42 | Tokens within legal case citations and academic journal abbreviations. |
| L7-E43 | Geopolitical entities and sociopolitical terms involving conflict, borders, or historical regions. | L7-E44 | Tokens describing the duration, frequency, or temporal status of states and events. |
| L7-E45 | Newlines and boilerplate punctuation in code comments and structured headers. | L7-E46 | Complex biological and technical terminology, especially within medical and laboratory research contexts. |
| L7-E47 | Proper nouns and navigational phrases related to media, politics, and news. | L7-E48 | Syntactic boundaries and logical conjunctions including punctuation, 'and', 'but', and 'who'. |
| L7-E49 | Mid-word tokens within proper names and rare surnames. | L7-E50 | Adversarial or concessive transitions emphasizing contradictions, exceptions, or unexpected persistence. |
| L7-E51 | Syntactic markers and function words connecting clauses or expressing causal relationships. | L7-E52 | Function words within formal titles of organizations, courts, and academic institutions. |
| L7-E53 | Tokens relating to professional career development, workplace skills, and organizational success. | L7-E54 | Interrogative and relative clauses inquiring about identity, manner, reason, or location. |
| L7-E55 | Physical actions involving movement, body positioning, or spatial displacement in narratives. | L7-E56 | Punctuation marks and conjunctions that transition between or conclude independent clauses. |
| L7-E57 | Anaphoric markers and adverbs that reference preceding concepts or actions. | L7-E58 | Punctuation and alphanumeric characters within complex IUPAC chemical nomenclature and score brackets. |
| L7-E59 | Technical, scientific, and medical terminology modifiers in complex noun phrases. | L7-E60 | Cognitive processes related to observation, memory, research, and understanding. |
| L7-E61 | Determiners, pronouns, and ordinal adjectives specifying objects in formal or technical descriptions. | L7-E62 | Personal and possessive pronouns, and specific human subjects or agents. |
| L7-E63 | Physical descriptive terms and anatomical features of biological species and structures. | L9-E0 | Adjectives and nouns describing qualitative states, conditions, or properties. |
| L9-E1 | Adjectives and adverbs that qualify technical, quantitative, or legal states. | L9-E2 | Passive verbs and adjectives describing states or measurements followed by prepositions. |
| L9-E3 | Negation and exclusionary terms like not, n't, never, and without. | L9-E4 | Software development identifiers and keywords in logs, code, and system outputs. |
| L9-E5 | LaTeX mathematical commands and syntax within technical and scientific notation. | L9-E6 | Mid-word fragments in proper names, special characters, and regex syntax. |
| L9-E7 | Tokens initiating parenthetical explanations or numerical values starting with leading zeros. | L9-E8 | Activates on punctuation and symbols following abbreviations, technical identifiers, or URLs. |
| L9-E9 | Sub-word segments within complex chemical nomenclature and biochemical terminology. | L9-E11 | Completes multi-token words and idiomatic phrases by predicting suffixes and related terms. |
| L9-E12 | Syntactic separators and structural markers like newlines, quotes, and URL delimiters. | L9-E13 | Activates on punctuation and formatting characters that delimit clauses, sentences, or paragraphs. |
| L9-E14 | Activates on 'the' and other determiners to predict upcoming idiomatic noun complements. | L9-E15 | Narrative actions and character dialogue transitions in descriptive storytelling. |
| L9-E16 | Punctuation marks and conjunctions at the end of clauses or sentences. | L9-E17 | Identify suspicious characters or entities to predict associated criminal or supernatural descriptors. |
| L9-E18 | Syntactic transitions and structural markers in encyclopedic, technical, and disambiguation entries. | L9-E19 | Abstract concepts related to morality, ethics, religion, and subjective human consciousness. |
| L9-E20 | Scientific and technical word fragments, particularly those involving biology, chemistry, and mathematics. | L9-E21 | Punctuation marks and conjunctions transitioning to explanatory or affirmative follow-up statements. |
| L9-E22 | Adverbs and auxiliary verbs that modify the degree or timing of a state. | L9-E23 | Contrastive and concessive conjunctions or adverbs marking shifts in reasoning or sequence. |
| L9-E24 | Proper nouns, titles, and names within specific named entities. | L9-E26 | Tokens that form semantic units related to physical hobbies, crafts, or biological terms. |
| L9-E27 | Nouns often found in metadata, website navigation, or technical documentation. | L9-E28 | Scientific and technical terminology involving measurement, properties, and analytical methods. |
| L9-E29 | Transitional verbs involving communication, cognitive actions, or completing common phrasal verbs. | L9-E30 | Coordinating conjunctions and determiners in complex parallel structures or restrictive phrases. |
| L9-E31 | Proper names and titles within formal institutional or legal designations. | L9-E32 | Prepositions and verbs indicating spatial or physical movement and relative positioning. |
| L9-E33 | Tokens that typically function as stems for common suffixes like -ive, -ion, or -ly. | L9-E34 | Structural delimiters in metadata, diffs, and formatted tabular lists. |
| L9-E35 | Proper nouns and repeated key terms in title-case or specialized terminology. | L9-E36 | Adjectives and articles preceding descriptive nouns or figurative noun phrases. |
| L9-E37 | Legal terminology related to bankruptcy, debt, and specialized regulatory procedures. | L9-E38 | Adjectives and determiners identifying the specific current subject, time, or entity. |
| L9-E39 | Syntactic punctuation and transition words in structured legal, technical, or formal text. | L9-E40 | Proper names, specifically initials and surname prefixes within formal or bibliographic contexts. |
| L9-E41 | Tokens related to leisure activities, recreational equipment, and organized social events. | L9-E42 | Punctuation and auxiliary verbs identifying research objectives or legal procedural steps. |
| L9-E43 | Prepositions and temporal markers used in complex adverbial and prepositional phrases. | L9-E44 | Quantifiers and comparative phrases specifying amount, frequency, or selection from a set. |
| L9-E45 | Logical connectives and punctuation marking comparative, conditional, or contrastive clausal transitions. | L9-E46 | Asynchronous keywords and common programming data types in software development contexts. |
| L9-E47 | Nouns and verbs related to formal processes, judgment, and institutional actions. | L9-E48 | Proper nouns and domain-specific terminology across technical, geographic, and medical fields. |
| L9-E49 | Subordinate conjunctions and relative pronouns introducing hypothetical, conditional, or explanatory clauses. | L9-E50 | Numerical ranges and relational data markers, especially hyphens and parentheses. |
| L9-E51 | Method chaining and property access via dots or closing parentheses in code. | L9-E52 | Punctuation and conjunctions that coordinate items within lists or parentheticals. |

## OLMoE-1B-7B

| L-E | Label | L-E | Label |
|---|---|---|---|
| L9-E53 | Verbs and prepositions indicating optimization, improvement, or functional enhancement of a process. | L9-E54 | Numerical separators and punctuation in statistics, phone numbers, and identifiers. |
| L9-E55 | Nouns and pronouns that conclude a semantic unit or prepositional phrase. | L9-E56 | Evaluative and epistemic phrases expressing current impressions, modality, or qualifying statements. |
| L9-E57 | Mid-word tokens and delimiters within URLs, scientific citations, and surnames. | L9-E58 | Common function words like prepositions and articles across various contexts. |
| L9-E59 | Verbs involved in idiomatic phrasal constructions and their subsequent prepositional complements. | L9-E60 | Proper names and locations from non-Western cultures, especially Asian and African. |
| L9-E61 | Syntactic functional words within standard open-source software license boilerplates. | L9-E62 | Personal pronouns and reflexive pronouns, often preceding slashes or coordinated self-references. |
| L9-E63 | Identify and complete tokens within specific entities, franchises, or proper nouns. | L11-E0 | Articles and possessives that precede nouns with specific qualifiers or attributes. |
| L11-E1 | Descriptions of physical components, manufacturing materials, and technical specifications for products. | L11-E2 | Proper nouns and capitalized first names, particularly multi-syllabic ones. |
| L11-E3 | Adjectives and adverbs that require specific prepositional or adverbial complements. | L11-E4 | Predicts continuation of references or navigational links after 'See' or 'unless'. |
| L11-E5 | Activates on specific characters to predict the second half of common abbreviations. | L11-E6 | Syntactic prepositions and verbs that predict religious, historical, or genealogical completions. |
| L11-E7 | Processes Base64-encoded strings, structured data delimiters, and specific proper name components. | L11-E8 | Completes common verb-preposition collocations and idiomatic phrasal verbs. |
| L11-E9 | Morphemes and sub-tokens within complex biological, medical, and taxonomic terminology. | L11-E10 | Activates on common function words and punctuation that initiate or link clauses. |
| L11-E11 | Activates on 'other' in contrastive phrases to predict 'hand' or 'side'. | L11-E12 | Completes specialized word stems into technical, academic, or categorical terms. |
| L11-E13 | Activates on 'obj' and programming keywords in technical code and build configurations. | L11-E14 | Metadata and categorisation tokens in Wikipedia articles and structured web URLs. |
| L11-E15 | Time-related nouns and phrases, often predicting their idiomatic or temporal continuations. | L11-E16 | Completes common idiomatic expressions and rhetorical inquiries after existential or interrogative starts. |
| L11-E17 | Proper nouns and adjectives relating to political, national, or biological entities. | L11-E18 | Indefinite pronouns and comparative expressions indicating inclusion or extent across entities. |
| L11-E19 | Predicts evaluative adjectives or descriptors following intensifying adverbs and linking verbs. | L11-E20 | Completes phrasal verbs and compound words by predicting their idiomatic suffixes. |
| L11-E21 | Auxiliary verbs and conjunctions signaling conditional outcomes, permissions, or necessities. | L11-E22 | Punctuation marks preceding sentence-starting transition words or comparative adverbs. |
| L11-E23 | Syntactic delimiters and separators in structured data, including LaTeX, UUIDs, and URLs. | L11-E24 | Coordinating conjunctions and punctuation separating items in pairs or lists. |
| L11-E25 | Physical actions, bodily parts, and descriptive movements in biblical, literary, or erotic contexts. | L11-E26 | Activates on determiners and pronouns to predict contextually relevant noun phrases. |
| L11-E27 | Syntactic punctuation and keywords that initiate structured code blocks and function calls. | L11-E28 | Tokens within formal titles, academic citations, and legal headers. |
| L11-E29 | Proper names and titles starting with the letter clusters Sch, Ar, or Kr. | L11-E30 | Syntactic markers for relative clauses and clarifying appositives that provide additional information. |
| L11-E31 | Verbs and prepositions predicting subsequent reflexive pronouns or clarifying objects. | L11-E32 | Processes prepositional phrases and set expressions to predict subsequent idiomatic components. |
| L11-E33 | Newlines and punctuation following completed sentences or code blocks. | L11-E34 | Verbs and nouns indicating relocation, omission, or placement followed by spatial prepositions. |
| L11-E35 | Predicts subsequent idiomatic or conditional completions for common multi-word phrases. | L11-E36 | Predicts a verb following a relative clause or prepositional phrase. |
| L11-E37 | Newlines and line breaks that separate distinct document sections or headers. | L11-E38 | Newlines and tokens in spiritual, astrological, or formal patent-related contexts. |
| L11-E39 | Common English idioms and dummy subject constructions beginning with 'It'. | L11-E40 | Proper nouns and titles, especially those containing conjunctions and articles. |
| L11-E41 | Common word roots and proper noun prefixes across technical and scientific domains. | L11-E42 | Completes idiomatic phrases and intensifiers by predicting highly probable subsequent nouns. |
| L11-E43 | Punctuation and numeric components within legal citations and timestamped date strings. | L11-E44 | Punctuation marks used in emoticons, citations, and technical code delimiters. |
| L11-E45 | Completes common idiomatic prepositional phrases by predicting their associated noun objects. | L11-E46 | Technical abbreviations, acronyms, and alphanumeric identifiers in scientific or computational contexts. |
| L11-E47 | Activates on LaTeX mathematical syntax, especially delimiters, fraction commands, and operators. | L11-E48 | Predicts medical and scientific research terminology following experimental or methodological descriptions. |
| L11-E49 | Adjectives or modifiers preceding specific nouns to form common multi-word phrases. | L11-E50 | Political and economic concepts related to labor, healthcare, and public policy. |
| L11-E51 | Capitalized word fragments and syllable starts, often in proper nouns or lists. | L11-E52 | Terminal punctuation and structural transitions that precede conversational responses or citations. |
| L11-E53 | Transitions between life stages, daily routines, and work-life commitments. | L11-E54 | Abstract and technical nouns, often serving as objects or conceptual subjects. |
| L11-E55 | Proper nouns within formal entities like government agencies, courts, and institutions. | L11-E56 | Activates on URL components and code delimiters to predict proper nouns. |
| L11-E57 | Numerical values including street addresses, percentages, times, and written-out numbers. | L11-E58 | Nouns or noun-phrases acting as range boundaries, typically following the word 'from'. |
| L11-E59 | Processes non-English European languages and academic citations involving multiple authors. | L11-E60 | Interpersonal markers in requests and inquiries that signal uncertainty or politeness. |
| L11-E61 | Identify indicators of patent background sections to predict 'art' and 'invention'. | L11-E62 | Punctuation and technical delimiters within URLs, IP addresses, and code namespaces. |
| L11-E63 | Auxiliary verbs and verbs describing task completion or operational states. | L13-E0 | Nouns and pronouns followed by verbs describing their state or behavior. |
| L13-E1 | Activates on prepositions and verbs that initiate specific object-modifying phrases. | L13-E2 | Indentation and newline characters in structured code across multiple programming languages. |
| L13-E3 | Predicts video game mechanics, music structures, or academic study components. | L13-E4 | Activates on the uppercase letter 'E' at the start of words. |

## OLMoE-1B-7B

| L-E | Label | L-E | Label |
|---|---|---|---|
| L13-E5 | Tokens preceding line breaks or punctuation in formal headers and lists. | L13-E6 | Software licensing boilerplate and framework-specific project structure patterns. |
| L13-E7 | Transliteration and linguistic analysis of East Asian languages and technical notation. | L13-E8 | Tokens within phrasal verbs or collocations, primarily predicting their required particle. |
| L13-E9 | Proper nouns and titles, particularly organizations, artistic works, and institutional names. | L13-E10 | Completes multi-token words by predicting the next morphological or semantic segment. |
| L13-E11 | Predicting hardware, networking, and software-specific terminology within technical contexts. | L13-E12 | Pronouns following clauses that introduce common idiomatic expressions or personal reflections. |
| L13-E13 | Predicts professional document types and formal bureaucratic procedures within employment and academic contexts. | L13-E14 | Predicts specialized movements, schools, or philosophies in arts, humanities, and religion. |
| L13-E15 | Legal prepositions and conjunctions introducing or connecting specific criminal charges. | L13-E16 | Latin-based text and scientific nomenclature containing Latin word endings and roots. |
| L13-E17 | Biological and genetic acronyms, specifically protein and gene name abbreviations. | L13-E18 | Common word prefixes and initial subword fragments like 'pre', 'de', and 'inc'. |
| L13-E19 | Descriptions of mechanical assemblies and their specific functional or structural components. | L13-E20 | Activates on prepositional or conjunctive phrases in medical research reporting, predicting methodological terms. |
| L13-E21 | Processes inflectional endings and grammatical connectors across various Romance and Germanic languages. | L13-E22 | Nouns that precede prepositional phrases, specifically identifying relationship or specification markers. |
| L13-E23 | Proper noun separators and prepositions in lists, titles, and technical identifiers. | L13-E24 | Phrasal verb and idiomatic expression completion, focusing on common prepositional phrases. |
| L13-E25 | Predicts subsequent method or property names following a dot operator in programming. | L13-E26 | Linking verbs and auxiliaries followed by role, state, or status descriptors. |
| L13-E27 | Numerical values followed by units of measurement, currency, or statistical categories. | L13-E28 | Completes multi-token technical terms, medical conditions, and specific proper names. |
| L13-E29 | Proper nouns and descriptors within genealogical, biographical, and geographical contexts. | L13-E30 | predicting judicial actions and rulings following legal subjects or procedural transitions. |
| L13-E31 | Completes technical and functional compound terms or multi-word phrases. | L13-E32 | Scientific and technical identifiers involving alphanumeric symbols, citations, or taxonomical nomenclature. |
| L13-E33 | Predicts active verbs or state-changes following subjects and auxiliary verbs. | L13-E34 | Verbs and phrases involving completing common idiomatic or phrasal verb constructions. |
| L13-E35 | Adjectives or determiners followed by specific abstract nouns describing attributes or categories. | L13-E36 | Tokens related to aviation, aerodynamics, and military air forces. |
| L13-E37 | Predicts common auxiliary verb complements and their following infinitives or objects. | L13-E38 | Syntactic subjects or modals predicting active verbs and their consequences. |
| L13-E39 | The definite article 'the' and its variants preceding specific proper names or prefixes. | L13-E40 | Predicts qualitative adjectives describing a person's behavior, style, or performance. |
| L13-E41 | Activates on objects or actions requiring temporal or spatial qualifying adverbs. | L13-E42 | Proper names of political, sports, and public figures following titles or connectors. |
| L13-E43 | Identifying vehicle and hardware brands to predict specific model names. | L13-E44 | Predicting familial or close interpersonal relationship terms following possessives or qualifiers. |
| L13-E45 | Identifies professional titles and attribution verbs preceding names in journalistic or academic contexts. | L13-E46 | Syntactic markers in structured data and legal or bibliographic citations. |
| L13-E47 | Function words and punctuation in lore-heavy summaries of fictional universes. | L13-E48 | Nouns and modifiers within specific geopolitical, institutional, or socioeconomic entities. |
| L13-E49 | Coordinating conjunctions and punctuation separating items in professional, technical, or descriptive lists. | L13-E50 | Predicts essential state or condition adjectives following linking verbs or logical conjunctions. |
| L13-E51 | Activates on prefixes 'over' and 'under' to predict the following word root. | L13-E52 | Predicting conflict-related entities such as weapons, targets, or catastrophic events. |
| L13-E53 | Predicting specific nouns in culinary, medicinal, and household care instructions. | L13-E54 | Processes punctuation and suffixes within technical, numerical, and scientific identifiers. |
| L13-E55 | Mathematical and statistical delimiters including parentheses, commas, slashes, and operators. | L13-E56 | Predicts geographical locations or entities within list-like structures and entity descriptions. |
| L13-E57 | Identify the first word of common business, economic, or technical compound terms. | L13-E58 | Predicts verbs describing intent, effect, or function after logical connectives. |
| L13-E59 | Predicting specific animal species or types based on biological or environmental context. | L13-E60 | Biological and medical descriptions of human body parts, symptoms, and neurological functions. |
| L13-E61 | Syntactic markers and keywords in programming code and phrasal verbs. | L13-E62 | Syntactic elements in code, especially shell variables, path components, and metadata fields. |
| L13-E63 | Physical features, infrastructure, and geographical elements of landscapes and properties. | L14-E0 | Predicts verbs describing reactions, consequences, or persistence following personal and behavioral indicators. |
| L14-E1 | Tokens that initiate mathematical symbols, slashes, or the prefix 'inter'. | L14-E2 | Predicting specific proper names following prepositions and conjunctions in biographical contexts. |
| L14-E3 | Predicts specialized nouns or roles following descriptors in biographical and athletic contexts. | L14-E4 | Activates on the letter 'j' in Java-related technical terms or format specifiers. |
| L14-E5 | Connective tokens linking geographic entities within administrative or location-based lists. | L14-E6 | Software frameworks and technical components within specialized computing or programming environments. |
| L14-E7 | Scientific and technical identifiers, focusing on mathematical indices and citation reference numbers. | L14-E8 | Processes grammatical endings and punctuation in Romance languages and Latin text. |
| L14-E9 | Punctuation following descriptive summaries, especially hyphens in age descriptions and period-newline sequences. | L14-E10 | Signals subjects or relative pronouns that require an upcoming functional verb. |
| L14-E11 | Syntactic separators and line breaks in structured data and formal legal documents. | L14-E12 | Identifies parenthetical citations or manufacturer details within scientific and technical descriptions. |
| L14-E13 | Hexadecimal strings, alphanumeric identifiers, and Greek text segments. | L14-E14 | Government agencies, official legal documents, and military or space missions. |
| L14-E15 | Abstract and concrete nouns that require restrictive post-modification or clarification. | L14-E16 | Phrasal verbs and idiomatic expressions involving motion or auxiliary verbs. |
| L14-E17 | Interrogative and relative pronouns including what, which, who, and that. | L14-E18 | predicting reporting verbs following subjects or auxiliary verbs in news reporting |
| L14-E19 | Completes specific multisyllabic word roots starting with common prefixes like res, int, and pl. | L14-E20 | Prepositions within idiomatic or quantitative phrases describing location, time, or state. |

## OLMoE-1B-7B

| L-E | Label | L-E | Label |
|---|---|---|---|
| L14-E21 | Physical state or positional changes involving movement, safety, and location. | L14-E22 | Proper nouns and title components that begin a specific entity name. |
| L14-E23 | Adjectives and nouns describing technical specifications, quality standards, or functional categories. | L14-E24 | Tokens forming hyphenated compound words, especially those beginning with the prefix 'self'. |
| L14-E25 | Physical athletic movements, techniques, and sports-related drills or maneuvers. | L14-E26 | Proper nouns and taxonomic names within biological descriptions and cast lists. |
| L14-E27 | Syntactic function words and common suffixes in West Germanic and North Germanic languages. | L14-E28 | Cybersecurity, cryptocurrency, and digital infrastructure technical terms and their contextual completions. |
| L14-E29 | Tokens starting with the prefix 'be' or 'Be' across various words. | L14-E30 | Predicting specialized medical procedures and complications within clinical and surgical contexts. |
| L14-E31 | Syntactic functional tokens predicting context-specific verbs such as commercial, professional, or physical actions. | L14-E32 | Legal terminology describing standard of review, jurisdictional issues, and evidentiary findings. |
| L14-E33 | Biochemical laboratory procedures involving purification, filtration, and isolation techniques. | L14-E34 | Tokens involving website navigation instructions, call-to-action links, and code object references. |
| L14-E35 | Indefinite articles preceding qualitative descriptors or idiomatic noun phrases. | L14-E36 | Analyzes technical context to predict specific software utilities and configuration syntax. |
| L14-E37 | Processes components of calendar dates to predict subsequent numerical or year markers. | L14-E38 | Scientific and physics-related noun phrases, particularly within astrophysics and quantum mechanics contexts. |
| L14-E39 | Tokens starting with the prefix 'dis' or 'Dis' to complete words. | L14-E40 | Proper nouns and named entities, particularly those beginning with 'Port', 'Power', or 'Operation'. |
| L14-E41 | Tokens involving qualitative assessment, nuanced descriptions, or communication of specific information types. | L14-E42 | Activates on terminal punctuation and emoticons followed by newline-based continuations. |
| L14-E43 | Tokens relating to infrastructure, property, transportation, and specialized commercial assets. | L14-E44 | Nouns and adjectives preceding academic citations, parentheticals, or clarifying punctuation. |
| L14-E45 | Adversative and additive conjunctions, especially 'However' and 'and', introducing new clauses. | L14-E46 | Tokens that form the first part of compound words or specific technical terms. |
| L14-E47 | The lowercase letter 'f' at the beginning of words or subwords. | L14-E48 | Auxiliary and copular verbs preceding words describing a change in state or scale. |
| L14-E49 | Proper nouns in locations, schools, and organizations requiring specific suffix completions. | L14-E50 | Predicting narrative action verbs following a subject or coordinating conjunction. |
| L14-E51 | Syntactic transitions and punctuation in liturgical, biblical, poetic, and Latin texts. | L14-E52 | Numeric values immediately preceding units of measurement, dates, or time durations. |
| L14-E53 | Prepositions and verbs within fixed idiomatic or phrasal verb expressions. | L14-E54 | Common comparative and temporal prepositions like 'as', 'than', 'after', and 'like'. |
| L14-E55 | Political conflict terms and actions related to power struggles or institutional opposition. | L14-E56 | Linking verbs and intensifiers followed by descriptive adjectives or evaluative predicates. |
| L14-E57 | Tokens that form the first part of common compound words and idioms. | L14-E58 | Syntactic structures in markup languages, programming code, and web protocols. |
| L14-E59 | predicting mechanics, stats, and character classes in tabletop and video game RPGs | L14-E60 | Plural technical nouns and punctuation within technical, scientific, or legal documentation. |
| L14-E61 | Tokens 'w' or 'st' predicting words starting with those letters. | L14-E62 | Industrial products, technical equipment, and specialized consumer goods specifications. |
| L14-E63 | Legal and formal administrative terminology, specifically involving court cases and institutional descriptions. | L15-E0 | Identify and complete substrings within URLs, IDs, and multi-part proper names. |
| L15-E1 | Structural delimiters including newlines, colons in timestamps, and technical punctuation. | L15-E2 | Abstract concepts, physical orientations, and descriptive states concluding a thought. |
| L15-E3 | Punctuation separating digits in numerical lists, population statistics, and timestamps. | L15-E4 | Modal verbs and infinitive markers facilitating action-oriented or conditional verb phrases. |
| L15-E5 | Adjectives describing geographic classifications and anatomical positions within specialized technical domains. | L15-E6 | Predicting completions for verbs and adjectives beginning with 'sh' and 'bl'. |
| L15-E7 | Syntactic functional words and common connectors in descriptive academic or technical English. | L15-E8 | Determiners, connectors, and punctuation in technical or multilingual programming contexts. |
| L15-E9 | Compound identifiers and camelCase terms in code and media titles. | L15-E10 | Predicts achievement and overcoming verbs following modal verbs, adverbs, and infinitives. |
| L15-E11 | Apostrophes following personal pronouns to predict contractions like 'm, 're, or 'd. | L15-E12 | Activates on 'and' to predict common articles like 'the' or 'a'. |
| L15-E13 | Subject pronouns and animate nouns performing or undergoing physical or habitual actions. | L15-E14 | Ecclesiastical and ritualistic descriptions, specifically regarding religious orders, clothing, and vocations. |
| L15-E15 | Auxiliary verbs and conjunctions signaling upcoming qualitative or descriptive attributes. | L15-E16 | Proper names of people, often preceded by titles or legal citations. |
| L15-E17 | Closes LaTeX mathematical environments by predicting closing braces and formatting markers. | L15-E18 | Syntactic functional tokens facilitating grammatical transitions and connecting phrases within sentences. |
| L15-E19 | Syntactic dependencies within formal documents, technical specifications, and structured data lists. | L15-E20 | Indefinite articles and determiners initiating noun phrases, often predicting descriptive adjectives. |
| L15-E21 | Determiners and adjectives preceding nouns, specifically the definite article 'the'. | L15-E22 | Common verbs and proper noun prefixes that initiate multi-word named entities. |
| L15-E23 | Tokens that typically function as prefixes or components of compound nouns. | L15-E24 | Tokens preceding numerical values in dates, times, census data, and code. |
| L15-E25 | Tokens relating to professional roles, recruitment, and institutional labor processes. | L15-E26 | Technical descriptions of mechanical components, vehicles, and engineering systems. |
| L15-E27 | Tokens within descriptions of media, games, and technology predicting related titles. | L15-E28 | Activates on demonstrative pronouns starting sentences to predict subsequent verbs. |
| L15-E29 | Tokens starting with 'fl' that initiate words related to movement or texture. | L15-E30 | Technical terms, punctuation, and keywords in programming-related questions and code documentation. |
| L15-E31 | Mid-word tokens that help complete complex or multi-syllabic Latinate terms. | L15-E32 | Syntactic function words and prepositions connecting clauses or defining structural relationships. |
| L15-E33 | Punctuation marks and symbols that signal the completion of a clause or phrase. | L15-E34 | Scientific methodology and experimental terminology in physics, chemistry, and astronomy contexts. |
| L15-E35 | Initial characters or character sequences within uppercase words, URLs, and hyphenated terms. | L15-E36 | Tokens within randomized alphanumeric strings in URLs and technical identifiers. |

## OLMoE-1B-7B

| L-E | Label | L-E | Label |
|-----|-------|-----|-------|
| L15-E37 | Tokens starting with 'as' or 'As' that form longer words. | L15-E38 | Scientific and medical descriptions identifying species classifications or disease pathologies. |
| L15-E39 | Scientific and culinary procedural descriptions involving preparation, substances, and specific ingredients. | L15-E40 | Activates on the prefix 'he' to predict subsequent word-completing suffixes. |
| L15-E41 | Activates on apostrophe-s suffixes and predicts following determiners or nouns. | L15-E42 | Proper nouns and connective words in names of official institutions and legal statutes. |
| L15-E43 | Tokens forming words starting with the 'sp' sound and related phonetic fragments. | L15-E44 | Tokens starting with the prefix 'end' or 'End' in medical or technical contexts. |
| L15-E45 | Descriptive anatomical or technical features in biological, architectural, and botanical taxonomies. | L15-E46 | Syntactic modal and conditional markers in instructional or legal documentation. |
| L15-E47 | Multilingual comma usage after clauses or within descriptive lists across multiple languages. | L15-E48 | Prepositions and punctuation introducing or connecting geographical locations and local institutions. |
| L15-E49 | Syntactic structures introducing biographical details, nationalities, and professional roles within descriptive sentences. | L15-E50 | Tokens involving the word 'just', its derivatives, and archaic or poetic phrasing. |
| L15-E51 | Financial institutions and banking terminology within investment and corporate contexts. | L15-E52 | Punctuation and uppercase functional words in list-like or technical contexts. |
| L15-E53 | Tokens starting with the letter 'b' (case-insensitive) often predicting word completions. | L15-E54 | Nouns and symbols functioning as subjects or objects in descriptive clauses. |
| L15-E55 | Determiners and adjectives preceding ordinal, temporal, or superlative sequential modifiers. | L15-E57 | Processes multi-byte characters and leading non-ASCII bytes in non-Latin scripts. |
| L15-E58 | Activates on common grammatical suffixes and functional particles across multiple European languages. | L15-E59 | Sports personnel names and professional roster or competition status descriptors. |
| L15-E60 | Punctuation marks and abbreviations specifically within formal titles, mathematical expressions, and legal citations. | L15-E61 | Activates on prepositions and conjunctions, particularly 'in', predicting following determiners or nouns. |
| L15-E62 | Forms of the verb 'to be' that precede descriptive adjectives or participles. | L15-E63 | Adverbs and auxiliary verbs in Romance languages and common English functional prepositions. |

*Table 11.* `Qwen3-30B-A3B` Expert labels.

## Qwen3-30B-A3B

| L-E | Label | L-E | Label |
|---|---|---|---|
| L4-E0 | Common multi-word idiomatic phrases and logical connectors. | L4-E1 | Scientific and technical compound terms, especially those involving hyphens or multi-token suffixes. |
| L4-E2 | Alphanumeric identifiers within biological, chemical, or technical nomenclature. | L4-E3 | Activates on common camelCase and PascalCase identifiers in code across multiple languages. |
| L4-E4 | Procedural action verbs and directional prepositions in instructional or technical contexts. | L4-E5 | Technical terms and specialized jargon followed by their abbreviations or suffixes. |
| L4-E6 | Mid-word tokens within complex medical, chemical, and scientific terminology. | L4-E7 | Japanese proper nouns, including place names, surnames, and Romanized cultural terms. |
| L4-E8 | Romanized Chinese names, geographical locations, and historical dynastic terms. | L4-E9 | Activates on specific nouns and components of fruits, vegetables, and foodstuffs. |
| L4-E10 | Syntactic structures within descriptive clauses, particularly relative clauses and qualifying prepositional phrases. | L4-E11 | Pop culture, entertainment media, and digital subculture terminology and entities. |
| L4-E12 | Scientific and academic citations, specifically identifying journals, authors, and publication metadata. | L4-E13 | Mathematical variables and geometric terms within formal proofs or scientific problem statements. |
| L4-E14 | Sub-word segments within scientific, biological, and medical terminology. | L4-E15 | Technical industrial terminology and components in engineering, manufacturing, and certification contexts. |
| L4-E16 | Activates primarily on the digit 8 and sequences of historical dates. | L4-E17 | Transitional punctuation and function words in multilingual structured text or code. |
| L4-E18 | Industrial, technical, and chemical processes involving extraction, refinement, and production. | L4-E19 | Numeric components within structured technical citations, p-values, and identification codes. |
| L4-E20 | Italian, Spanish, and French suffixes in proper nouns and names. | L4-E21 | Software components, frameworks, and identifiers in technology-related source code and documentation. |
| L4-E22 | Adverbs and intensifiers that modify the degree or manner of qualities. | L4-E23 | Sub-word segments within proper nouns, surnames, and biological nomenclature. |
| L4-E24 | LaTeX and TikZ commands, keywords, and structural metadata elements. | L4-E25 | Adverbial and temporal transitional phrases marking shifts in time or logic. |
| L4-E26 | Escape characters and backslashes in code, LaTeX, and technical notation. | L4-E27 | Nouns and noun suffixes within complex legal, technical, and formal contexts. |
| L4-E28 | PascalCase component names within C# and .NET namespaces and class declarations. | L4-E29 | Adjective suffixes and word segments that describe qualities or sensory characteristics. |
| L4-E30 | Alphanumeric character sequences within technical, medical, and scientific identifiers or codes. | L4-E31 | Mathematical symbols and operators, particularly negative signs and parentheses in expressions. |
| L4-E32 | Suffixes of Irish surnames and specific punctuation-delimited technical substrings. | L4-E33 | Scientific and statistical terminology, specifically biochemical compounds and mathematical analysis methods. |
| L4-E34 | Software components, library names, and UI-related identifiers in code and documentation. | L4-E35 | Software testing frameworks and library path dependencies in various programming languages. |
| L4-E36 | Identify and process semantic data types, character sequences, and string manipulation concepts. | L4-E37 | Proper nouns and technical terms related to software, hardware, and networking. |
| L4-E38 | Geographic locations and settlement types in structured data and Wikipedia categories. | L4-E39 | Transitive verbs and light verb constructions followed by objects or particles. |
| L4-E40 | Tokens involving removal, dismissal, resignation, surrendering, or ending of states. | L4-E41 | Processes Chinese tokens within technical instructions, software documentation, and logical explanations. |
| L4-E42 | Chinese functional particles, pronouns, and common verbs in diverse contexts. | L4-E43 | Proper nouns and technical terms ending in specific phonetic suffixes like -av, -aw, and -oy. |
| L4-E44 | Proper names and locations within Spanish and Latin American contexts. | L4-E45 | Transliterated Chinese names, official titles, and encoded Chinese text fragments. |
| L4-E46 | Proper nouns, particularly geographically or institutionally specific multi-token names. | L4-E47 | Tokens expressing states of knowledge, belief, understanding, and communicative intent. |
| L4-E48 | Informal conversational sign-offs, first-person idiomatic expressions, and acronyms in forum-style communication. | L4-E49 | Mathematical and trigonometric functions, variables, and coordinate system calculations in code. |
| L4-E50 | Nouns and modifiers within specific noun phrases and quantifying expressions. | L4-E51 | Proper nouns and adjectives relating to international organizations, geopolitics, and biogeography. |
| L4-E52 | Syntactic transition points including sentence boundaries, punctuation, and structural web markup. | L4-E53 | Capitalized initial tokens of proper nouns and technical terms. |
| L4-E54 | Technical terms and terminology in advanced mathematics, physics, and computer science. | L4-E55 | Proper nouns and poetic phrases relating to East Asian culture and mythology. |
| L4-E56 | Common transitional phrases and introductory idiomatic expressions at the start of sentences. | L4-E57 | Proper nouns and specific titles in entertainment, media, and history. |
| L4-E58 | Informational header and navigation labels in encyclopedia or blog layouts. | L4-E59 | Software licensing terms, legal boilerplate declarations, and technical configuration components. |
| L4-E60 | Completes multi-token names of organizations, entities, and specific brand titles. | L4-E61 | Proper nouns and entity components within multi-token names or institutional titles. |
| L4-E62 | Software library namespaces and technical identifiers in API-related code snippets. | L4-E63 | Tokens involving the word 'on' and numerical or possessive contexts. |
| L4-E64 | Proper nouns and scientific terms within compound words or specific entities. | L4-E65 | Processes internal suffixes within scientific names, pharmaceutical drugs, and non-English proper nouns. |
| L4-E66 | Syntactic punctuation and operators within code, mathematical expressions, and configuration files. | L4-E67 | Physical descriptions of facial expressions and character gazes or body movements. |
| L4-E68 | Activates on compound nouns and specialized terminology in both Chinese and English. | L4-E69 | Technical suffixes and multi-part identifiers in web development and programming code. |
| L4-E70 | Proper nouns and adjectives denoting ethnic, national, religious, or ideological identities. | L4-E71 | Subword components of specific identifiers in C++ code and technical headers. |
| L4-E72 | Mathematical and LaTeX symbols in expressions, equations, and variable declarations. | L4-E73 | Metadata and functional labels in interfaces, code headers, and web snippets. |
| L4-E74 | Verbs in the past tense or describing states and processes across languages. | L4-E75 | Syntactic headers and spacing in formal legal documents and license headers. |
| L4-E76 | Connective and contrastive conjunctions, punctuation, and pronouns transition between clauses. | L4-E77 | Common abstract nouns and prepositions forming the core of idiomatic phrases. |

## Qwen3-30B-A3B

| L-E | Label | L-E | Label |
|---|---|---|---|
| L4-E78 | Linux command-line utilities and system-level configuration parameters in shell environments. | L4-E79 | Tokens forming the 'd' in legal reporter abbreviations like L.Ed.2d and F.3d. |
| L4-E80 | Sub-word segments within complex biochemical names, medical abbreviations, and clinical terminology. | L4-E81 | Syntactic structures in code like SQL values, function signatures, and docstrings. |
| L4-E82 | Technical terms and components in neuroscience, bio-mechanics, and electrical signal processing. | L4-E83 | Tokens identifying the second or contrasting element in a pair or sequence. |
| L4-E84 | Python scientific library keywords, module sub-components, and specialized method identifiers. | L4-E85 | Punctuation and connecting words in structured lists, legal citations, or appositives. |
| L4-E86 | Sub-word tokens within South and Southeast Asian proper names and locations. | L4-E87 | Adjectives and adverbs describing exclusivity, sequence, or directness of states and actions. |
| L4-E88 | Sub-word segments within complex biochemical, pharmaceutical, and scientific nomenclature. | L4-E89 | Mid-word syllables in complex chemical, biological, pharmaceutical, and scientific nomenclature. |
| L4-E90 | Archaic, biblical, and formal literary language across English, Russian, and Chinese. | L4-E91 | Relational prepositions and conjunctions that introduce prepositional phrases or dependent clauses. |
| L4-E92 | Non-alphanumeric punctuation and technical symbols in code, math, and data strings. | L4-E93 | Technical and scientific word fragments, particularly within academic citations and biological terminology. |
| L4-E94 | Syntactic control structures and structural delimiters in programming code. | L4-E95 | Mid-word lowercase or uppercase character segments within proper nouns and technical terms. |
| L4-E96 | Proper nouns and specific dates within institutional, educational, and legal contexts. | L4-E97 | Professional and corporate terminology related to business operations, communication, and market research. |
| L4-E98 | Legal and formal citations, focusing on abbreviations and section markers. | L4-E99 | Closing punctuation marks following abbreviated entities, technical terms, or names. |
| L4-E100 | Mid-word tokens in proper nouns, technical terms, and rare surnames. | L4-E101 | Compound nouns and specialized terms forming multi-token entities or phrases. |
| L4-E102 | Tokens within common prepositional phrases, especially those using 'in' or 'at'. | L4-E103 | Mathematical and comparative operators describing numerical relationships, limits, and divisibility. |
| L4-E104 | Activates on boilerplate phrases and functional transitions in legal, academic, or blog text. | L4-E105 | Identify and process verbs and pronouns within common idiomatic phrases and questions. |
| L4-E106 | Activates on punctuation and conjunctions that signal clausal boundaries or parenthetical transitions. | L4-E107 | Suffixes and morphemes in formal names, legal terms, and technical terminology. |
| L4-E108 | Technical terminology and citations within patent documents and scientific literature. | L4-E109 | Mathematical terminology and educational curricula concepts, specifically regarding arithmetic and calculus. |
| L4-E110 | Sub-word segments within complex surnames and specialized proper nouns. | L4-E111 | Function words like 'of', 'the', and 'a' in prepositional phrases. |
| L4-E112 | Recognizes and processes Chinese-related proper nouns and cultural terms across scripts. | L4-E113 | Official titles and institutional roles in political, judicial, or military contexts. |
| L4-E114 | Completes specific proper nouns, technical terms, and compound words by their suffixes. | L4-E115 | Punctuation marks and line breaks that terminate sentences or segments. |
| L4-E116 | Completes common idiomatic phrases and fixed multi-word legal or descriptive expressions. | L4-E117 | Common sub-word components within technical identifiers and function names in programming code. |
| L4-E118 | Colon separators in metadata and common name suffixes ending in 'as' or 'inas'. | L4-E119 | Tokens that are immediate or near-immediate repetitions of the preceding word. |
| L4-E120 | Tokens containing the character sequence 'ro' or phonetic variants in complex words. | L4-E121 | Mid-word or suffix components of surnames, technical terms, and proper nouns. |
| L4-E122 | Taxonomic suffixes and morphemes in biological nomenclature and species common names. | L4-E123 | Code block delimiters and structural transitions like docstrings, imports, and separators. |
| L4-E124 | Mid-word fragments and syllables within foreign or specialized proper nouns. | L4-E125 | Activates on camelCase or PascalCase sub-components within software code and identifiers. |
| L4-E126 | Sub-domain segments and path parameters within URLs and package names. | L4-E127 | Acronyms and specialized suffixes in scientific, organizational, and technical nomenclature. |
| L24-E0 | Nouns and nouns acting as adjectives in technical, scientific, or linguistic contexts. | L24-E1 | Syntactic completion of formal patterns in math equations, licensing, and code. |
| L24-E2 | Attributive and reportative phrases identifying spokespeople, organizations, or news sources. | L24-E3 | Logical transitions and operations in step-by-step mathematical or algorithmic derivations. |
| L24-E4 | Biographical data specifying a person's birth year within parentheses. | L24-E5 | Activates on common programming keywords and variable assignment operators in code. |
| L24-E6 | Medical and health-related terminology, specifically clinical outcomes, symptoms, and patient demographics. | L24-E7 | Punctuation and suffixes occurring at the end of introductory or dependent clauses. |
| L24-E8 | Proper names and geographic locations within formal credits, citations, and bibliographic metadata. | L24-E9 | Tokens involving definitions, nomenclature, translations, or technical synonyms. |
| L24-E11 | Academic and professional titles of books, journals, and papers. | L24-E12 | Logical consequence and auxiliary verbs in conditional or explanatory clauses. |
| L24-E13 | Mathematical terms and symbols within formal proofs, formulas, and derivations. | L24-E14 | Identifies specialized recreational activities and specific social groups to provide related conceptual associations. |
| L24-E15 | Punctuation and conjunctions connecting independent clauses or distinct propositions. | L24-E16 | Classical Chinese literary characters, poetic vocabulary, and domain-specific Chinese terminology. |
| L24-E17 | Nouns and technical terms within descriptive product, tool, or software metadata. | L24-E18 | Demographic and service-oriented descriptions involving households, families, and commercial listings. |
| L24-E19 | Abstract and concrete nouns in both Chinese and English technical contexts. | L24-E20 | Abbreviated legal case citations, technical file paths, and mathematical expressions. |
| L24-E21 | Fragments of names, identifiers, and technical symbols within code or structured formats. | L24-E22 | Activates on non-English morphological segments to predict intra-word or syntactic continuations. |
| L24-E23 | Syntactic structures and repetitive delimiters in structured data and code blocks. | L24-E24 | Capitalized words and formal titles in legal, technical, and structured headers. |
| L24-E25 | Technical and scientific terms within specialized domain descriptions. | L24-E26 | Anaphoric and deictic references used to re-identify previously mentioned entities. |
| L24-E27 | Tokens within formal names of organizations, institutions, and legal titles. | L24-E28 | Informational text elements and technical descriptions within academic, programming, or technical contexts. |
| L24-E29 | Sub-word components of CamelCase and snake_case identifiers in code. | L24-E30 | Mid-sentence functional and comparative transitions within lists or sequential descriptions. |

## Qwen3-30B-A3B

| L-E | Label | L-E | Label |
|-----|-------|-----|-------|
| L24-E31 | Tokens within terminal commands, file system paths, and system configuration scripts. | L24-E32 | Nouns and prepositions within superlative phrases or unique classification statements. |
| L24-E33 | Numerical values and mathematical operators within arithmetic expressions and list-based math problems. | L24-E34 | Tokens within historical context describing origins, founding, and past nomenclature. |
| L24-E35 | Specific numeric or textual values within delimited sequences and ranges. | L24-E36 | Numerical ranges and temporal indicators specifying age, time, or sequence segments. |
| L24-E37 | Common morphemes and sub-tokens within software library names and technical terminology. | L24-E38 | Concrete nouns and descriptive terms within specialized technical or descriptive contexts. |
| L24-E39 | Mid-sentence content words and digits, particularly within technical, scientific, or numeric sequences. | L24-E40 | Technical scientific terms across chemistry, biology, geology, and materials science. |
| L24-E41 | Tokens within complex alphanumeric identifiers, URLs, and technical codes. | L24-E42 | Adjectival and adverbial intensifiers in superlative or emotionally charged subjective descriptions. |
| L24-E43 | Detects exclusionary, negative, or restrictive statements in technical and descriptive text. | L24-E44 | Ingredients, tools, and preparation steps in culinary and recipe instructions. |
| L24-E45 | Tokens within descriptive, structured metadata like religious verses, geographic profiles, or programming problems. | L24-E46 | activates on concrete nouns and physical objects within descriptive, narrative scenes. |
| L24-E47 | Sub-word segments within technical identifiers, software names, and mathematical notation. | L24-E48 | Mathematical operations, variables, and logical connectors within formal proofs and formulas. |
| L24-E49 | Scientific and biochemical terminology within molecular biology and proteomics contexts. | L24-E50 | Tokens relating to academic study, scientific education, and formal educational qualifications. |
| L24-E51 | Conditional logic and comparative predicates in code and technical descriptions. | L24-E52 | Nouns and technical terms within parentheses, lists, or clarifying appositive phrases. |
| L24-E53 | Syntactic verbs and particles that introduce or link predicate descriptions. | L24-E54 | Individual characters within hexadecimal strings, numeric identifiers, and mathematical constants like pi. |
| L24-E55 | Single-character variables and short identifiers in code, math, or logic. | L24-E56 | Syntactic transitions and punctuation marking the start of new clauses or explanations. |
| L24-E57 | Syntactic elements and property values in CSS, HTML, and LaTeX code. | L24-E58 | Tokens describing efficiency improvements, cost reductions, or performance optimizations. |
| L24-E59 | Activates on specific nouns or entities that represent actors, subjects, or identifiers. | L24-E60 | Tokens defining relational equivalence or identity comparisons in technical and formal contexts. |
| L24-E61 | Economic and demographic trends, statistical metrics, and market fluctuations. | L24-E62 | Tokens within formal legal headers, disclaimers, and religious liturgical texts. |
| L24-E63 | Contrastive and concessive transitions that qualify or redirect the preceding statement. | L24-E64 | Numerical digits and individual numbers within structured, technical, or statistical data. |
| L24-E65 | Syntactic transition points following punctuation, section headers, or structural document boundaries. | L24-E66 | Mathematical symbols and digits within algebraic expressions and arithmetic word problems. |
| L24-E67 | Technical or formal keywords across coding, medical, and professional contexts. | L24-E68 | Syntactic function words facilitating structural transitions within complex explanatory or technical phrases. |
| L24-E69 | Prepositions and symbols indicating directional movement, transformation, or spatial relationships. | L24-E70 | Mandarin and English verbs or determiners indicating assistance, provision, or quantity. |
| L24-E71 | Repeated boilerplate code and formulaic descriptive phrases across multiple domains. | L24-E72 | Indentation and line breaks following code block delimiters or statement terminators. |
| L24-E73 | Processes structured technical, mathematical, and descriptive data within academic or instructional contexts. | L24-E74 | Syntactic functional units describing conditions, constraints, or causal relationships in technical text. |
| L24-E75 | Narrative verbs and transition words in storytelling and dialogue sequences. | L24-E76 | Syntactic elements and connectors within philosophical, legal, or logical propositions and laws. |
| L24-E77 | Tokens related to programming control flow 'break' and communicative action verbs. | L24-E78 | Metadata and structural boilerplate tokens in technical documentation, licenses, and web forms. |
| L24-E79 | Transitional conjunctions and logical connectors that introduce list items or subsequent clauses. | L24-E80 | Tokens describing the start, progression, or completion of a multi-step process. |
| L24-E81 | Syntactic identifiers and method calls within diverse programming code contexts. | L24-E82 | Syntactic punctuation and symbols in code, especially brackets, parentheses, and assignments. |
| L24-E83 | Mathematical symbols used for multiplication and sign notation in algebraic expressions. | L24-E84 | Business, logistical, and technical operational terminology in professional or descriptive contexts. |
| L24-E85 | Metadata and interface elements for website features like comments, profiles, and reviews. | L24-E86 | Syntactic elements and identifiers in structured data, code, and configuration files. |
| L24-E87 | Personal life events, emotional states, and individual interpersonal relationships. | L24-E88 | Syntactic structures and punctuation in programming code, especially assignments and terminators. |
| L24-E89 | Numerical sequences in contact details, identifiers, and mathematical formulas. | L24-E90 | Technical terms and components related to medical conditions and mechanical systems. |
| L24-E91 | Sub-word segments within proper names and multi-token surnames. | L24-E92 | Mid-word letter clusters in names, identifiers, and uncommon technical terms. |
| L24-E93 | Chinese proper nouns, Pinyin syllables, and classical Chinese poetry characters. | L24-E94 | Numeric and symbolic identifiers in scientific formulas, software versions, and sports scores. |
| L24-E95 | Tokens describing technical specifications, parameters, and structured criteria across various domains. | L24-E96 | Newlines and separator sequences that transition between document sections or headers. |
| L24-E97 | Phrases expressing personal perspective, cognition, or communicative intent in subjective discourse. | L24-E98 | Tokens describing data transformations and operations within code comments and documentation. |
| L24-E99 | Closing punctuation marks, brackets, and following conjunctions or sentence transitions. | L24-E100 | Verbs and prepositions involving the transfer, provision, or modification of resources. |
| L24-E101 | Technical and scientific nouns describing core entities, systems, or abstract properties. | L24-E102 | Tokens involving formal transactional arrangements, service availability, and administrative scheduling. |
| L24-E103 | Technical and scientific compound terms within complex noun phrases. | L24-E104 | Transitional verbs and functional particles connecting clausal ideas or descriptions. |
| L24-E105 | Tokens within formal headers, legal boilerplate, and technical documentation templates. | L24-E106 | English and Chinese verbs or prepositions denoting interaction, processing, or physical action. |
| L24-E107 | Coordinating conjunctions and prepositions linking parallel elements or contrasting clauses. | L24-E108 | Classical religious texts, poetic proverbs, and Latin placeholder text. |
| L24-E109 | Nouns and concepts in technical, mathematical, and computational frameworks. | L24-E110 | Physical and mathematical descriptors of movement, time, and field properties. |

## Qwen3-30B-A3B

| L-E | Label | L-E | Label |
|---|---|---|---|
| L24-E111 | Mid-word tokens in proper names, technical identifiers, and code strings. | L24-E112 | Compound identifiers and technical terms in code, data structures, and bibliography. |
| L24-E113 | Predicts subsequent components in technical descriptions, measurements, and formal data reports. | L24-E114 | Mid-sentence relational and descriptive transitions in technical, scientific, or formal contexts. |
| L24-E115 | Scientific or technical definitions and descriptive appositives for entities and concepts. | L24-E116 | Common words in introductory problem statements or meta-commentary about solutions. |
| L24-E117 | Numerical digits within multi-digit identification sequences, especially patent and blog IDs. | L24-E118 | Tokens indicating scope, degree, or logical inclusivity such as also, only, both, and even. |
| L24-E119 | Technical terms in computer science, algorithms, data processing, and machine learning. | L24-E120 | Function words connecting nouns to specify properties, relationships, or possessive dependencies. |
| L24-E121 | Spatial relationships and geometric positioning within physical structures, routes, or layouts. | L24-E122 | Hexadecimal strings, alphanumeric identifiers, and specific mid-word character sequences. |
| L24-E123 | Negations, exclusion markers, and conditional constraints across natural and technical languages. | L24-E124 | Infinitive verb phrases expressing purpose or functional intent following the word 'to'. |
| L24-E125 | Possessive markers and plural possessive apostrophes in English and similar grammatical markers. | L24-E126 | Structured metadata fields and technical key-value separators in code and email headers. |
| L24-E127 | Proper nouns and alphanumeric identifiers containing specific suffixes, numbers, or internal symbols. | L44-E0 | Coordinating conjunctions linking paired adjectives, verbs, or words in common idiomatic phrases. |
| L44-E1 | Numerical units of measurement, abbreviations, and their subsequent completions. | L44-E2 | Legal citations and formal identifiers involving numbers, abbreviations, and case law. |
| L44-E3 | Sports statistics and time-duration measurements followed by specific numeric or action completions. | L44-E4 | Transliterating and translating Chinese names, terms, or text between romanization and characters. |
| L44-E5 | Common adverbial or adjectival prefixes that initiate compound words. | L44-E6 | Metadata and details related to creative works like films, music, and performances. |
| L44-E7 | Numerical components and symbols within legal statute and code citations. | L44-E8 | C-style printf format strings, format specifiers, and escaped newline characters. |
| L44-E9 | Proper nouns and dates within British, Irish, and Greek biographical contexts. | L44-E10 | Geographical locations and infrastructure in China, particularly provincial divisions and transport networks. |
| L44-E11 | Proper nouns and entity names containing the 'man' syllable or fashion-related brands. | L44-E12 | Syntactic structure and geographic entities in Iranian village and district descriptions. |
| L44-E13 | Passive verbs and participles requiring specific prepositional complements like 'at', 'with', or 'to'. | L44-E14 | 3D graphics, shader programming, and geospatial data processing terminology. |
| L44-E15 | Tokens within formal names of academic, legal, or governmental organizations and journals. | L44-E16 | Mathematical and scientific symbols including LaTeX operators, chemical bonds, and relational delimiters. |
| L44-E17 | Statistical and financial terms related to mathematical functions, ratios, and trends. | L44-E18 | Predicts technical lifecycle actions like deletion, compilation, assignment, and invalidation. |
| L44-E19 | Proper nouns and alphanumeric sequences like call signs, names, and numbers. | L44-E20 | Tokens preceding nouns describing drawbacks, difficulties, or negative consequences. |
| L44-E21 | Processes natural language descriptions and technical comments in code documentation. | L44-E22 | Architectural features, property details, and real estate amenities. |
| L44-E23 | Predicts common suffixes or word completions for technical and formal stems. | L44-E24 | Sub-word segments and prefixes within complex chemical and biochemical nomenclature. |
| L44-E25 | Syntactic structures and bracketed identifiers in technical, legal, and programming contexts. | L44-E26 | Indonesian and Malaysian geographic locations, administrative divisions, and related entities. |
| L44-E27 | Military and hierarchical organizations, predicting specific unit types and ranks. | L44-E28 | Activates on common programming methods and functions to predict upcoming opening parentheses. |
| L44-E29 | Tokens representing components of dates and times in various data formats. | L44-E30 | Interrogative phrases and question structures in technical, mathematical, and conversational contexts. |
| L44-E31 | Code line terminators followed by indentation or newlines across multiple programming languages. | L44-E32 | Handling exceptions and error control flow across multiple programming languages. |
| L44-E33 | Numerical comparisons and limit conditions such as bonuses, thresholds, and ranges. | L44-E34 | Processes negative modifiers and tokens that initiate negated or notable descriptors. |
| L44-E35 | Numerical components in version numbers, section citations, and mathematical expressions. | L44-E36 | Network configuration parameters, protocol headers, and technical infrastructure identifiers. |
| L44-E37 | Syntactic structures involving auxiliary verbs followed by specific passive or idiomatic participles. | L44-E38 | Business terminology prediction focusing on organizational roles, marketing strategies, and corporate workflows. |
| L44-E39 | Sports tournament records, athlete biographies, and competition results involving specific event categories. | L44-E40 | Scientific and technical terminology involving physics, material science, and numerical sequences. |
| L44-E41 | Technical terms and code syntax, focusing on domain-specific identifiers and operators. | L44-E42 | Activates on prefixes to complete multisyllabic or Latinate academic words. |
| L44-E43 | Predicts mathematical and graph-theoretical properties or components of defined objects. | L44-E44 | Identify and predict words related to family members and kinship relationships. |
| L44-E45 | Modal verbs and negatives preceding common infinitive or idiomatic complements. | L44-E46 | Numerical digits and individual letters within technical, medical, or mathematical identifiers. |
| L44-E47 | Predicts subsequent items in alphanumeric, musical, or technical sequences. | L44-E48 | Determiners and adjectives followed by specific nouns like 'end', 'year', 'point', or 'way'. |
| L44-E49 | Bioinformatics terminology and specific database names within scientific and technical contexts. | L44-E50 | Industrial machinery and thermodynamic systems involving heating, cooling, and milling. |
| L44-E51 | Syntactic delimiters and structural separators in code, templating, and documentation. | L44-E52 | Tokens connecting locations to their administrative regions, particularly in Germanic countries. |
| L44-E53 | Commas following city names to predict state or regional abbreviations. | L44-E54 | Prepositions and particles within idiomatic phrases that predict common following articles. |
| L44-E55 | Syntactic structures and boilerplate in typed programming languages like Go, C++, and VBA. | L44-E56 | Predicts and identifies biological names, species, and domestic animal breeds. |
| L44-E57 | Mathematical operators and special characters within symbolic expressions and shell commands. | L44-E58 | Numerical digits in sequences representing measurements, codes, or quantitative values. |
| L44-E59 | Tokens inside parentheses, specifically citations, metadata, area codes, and news credits. | L44-E60 | Spanish-language geographical locations, historical figures, and related terminology or dates. |
| L44-E61 | Mathematical symbols, code operators, and auxiliary verbs in structured or formal text. | L44-E62 | Completes common multi-token words and compound terms by predicting their second half. |

## Qwen3-30B-A3B

| L-E | Label | L-E | Label |
|---|---|---|---|
| L44-E63 | Scientific computing code and documentation, particularly linear algebra libraries and matrix operations. | L44-E64 | Tokens preceding abbreviations in parentheses or specific components of multi-word titles. |
| L44-E65 | Structural keywords and punctuation that initiate or transition code and markdown blocks. | L44-E66 | Latin words and biological nomenclature, focusing on morphological endings and suffixes. |
| L44-E67 | Tokens within technical non-English contexts, including mathematical notation and agglutinative languages. | L44-E68 | Syntactic function words and logical connectors in mathematical and technical definitions. |
| L44-E69 | Predicts geographical sub-regions or state abbreviations from location contexts and prepositions. | L44-E70 | Tokens within file paths, URLs, and build system file extensions. |
| L44-E71 | Prepositions and verbs used within prepositional phrases or idiomatic qualifying expressions. | L44-E72 | Completes common idiomatic compounds and technical collocations. |
| L44-E73 | Subordinating conjunctions and adverbs that introduce causal or conditional dependent clauses. | L44-E74 | Technical and scientific nouns describing processes, systems, and research methodologies. |
| L44-E75 | Chinese adverbial modifiers and modal verbs indicating possibility, manner, or extent. | L44-E76 | Hardware-related identifiers, specifically microcontroller registers, PCB designators, and embedded software components. |
| L44-E77 | Predicts specialized class names and types within nested software package structures. | L44-E78 | Activates on Chinese historical names, dates, and locations to predict related Sinosphere entities. |
| L44-E79 | Scientific methodology transitions from experimental conditions to findings and analysis. | L44-E80 | Phrasal verb components and prepositions indicating direction, position, or completion. |
| L44-E81 | Syntactic connectors linking abstract social, moral, or professional concepts and values. | L44-E82 | Internal sub-word fragments of scientific names and specialized proper nouns. |
| L44-E83 | International organizational bodies and complex legal or regulatory frameworks. | L44-E84 | Technical terms in semiconductor manufacturing, mechanical engineering, and printing technology. |
| L44-E85 | Chinese pronouns, punctuation, and poetic structures in mixed-language or instructional contexts. | L44-E86 | Identify biological or pharmacological terms to predict related Chinese translations or characters. |
| L44-E87 | Completes common multi-character nouns and polite terminal verb phrases in East Asian languages. | L44-E88 | Pronouns and functional words predicting imminent verbs or auxiliary verbs. |
| L44-E89 | Predicting specialized medical terminology related to surgery, obstetrics, and medical procedures. | L44-E90 | Activates on punctuation ending mathematical premises to predict the next instructional step. |
| L44-E91 | Standard library and popular package module members or import statements. | L44-E92 | Technical terms and components related to audio, optics, electronics, and digital signals. |
| L44-E93 | CJK characters and punctuation transitioning between natural language and code comments. | L44-E94 | Wiki-style section headers and structured meta-data fields ending a content block. |
| L44-E95 | Scientific and astrophysical terminology completion within technical physics research contexts. | L44-E96 | Completes specific prefixes and mid-word fragments in academic and technical terminology. |
| L44-E97 | Software development configuration keys, tool-specific commands, and package dependency path components. | L44-E98 | Predicts software products and hardware brands within lists or descriptive technical contexts. |
| L44-E99 | Predicts modal verbs or auxiliary verbs following subjects in formal and legal texts. | L44-E100 | Processes internal word structures and components in Romance and Balto-Slavic languages. |
| L44-E101 | Syntactic fragments and morphological markers in Russian and technical multilingual strings. | L44-E102 | Identifies and processes URL components, file extensions, and path separators. |
| L44-E103 | Scientific citations, LaTeX formatting commands, and specialized prefix completion. | L44-E104 | Proper nouns and locations in East Asian contexts, specifically Japan and Korea. |
| L44-E105 | Numerical digits within years, dates, and technical versioning strings. | L44-E106 | Data structure and query terminology within code and database contexts. |
| L44-E107 | Predicting specific names following professional titles, roles, or connective punctuation. | L44-E108 | Automotive terminology, specifically car brands, models, specifications, and interior components. |
| L44-E109 | Predicts subsequent mathematical variables or physical coordinates following operators and delimiters. | L44-E110 | Nouns and adjectives preceding comparative or conditional conjunctions like 'as' and 'so'. |
| L44-E111 | Numerical sequences and identifiers within code, mathematical expressions, and dates. | L44-E112 | Syntactic operators and identifiers within code structures and markup tags. |
| L44-E113 | Phrasal completion of 'to' in fixed idioms like 'to do' or 'to reason'. | L44-E114 | Predicts grammatical suffixes for Russian and Slavic verbal and nominal stems. |
| L44-E115 | Classical Chinese poetry, idioms, and formal terminology related to administrative governance. | L44-E116 | Predicts nouns for tools, clothing, and household objects within descriptive lists. |
| L44-E117 | Activates on Chinese tokens related to programming documentation, instructions, and system operations. | L44-E118 | Processes and predicts tokens within complex medical conditions and disease terminology. |
| L44-E119 | Technical terms and formal department names in biological, medical, and governmental contexts. | L44-E120 | Legal, financial, and administrative terms, especially relating to official records or investigations. |
| L44-E121 | Identifying formal textual structures to predict academic or technical classification terms. | L44-E122 | predicting descriptive physical attributes and bodily parts in sexual or emotive contexts |
| L44-E123 | Activates on structure members and field accessors in code and data serializations. | L44-E124 | Chinese vocabulary and punctuation within technical, academic, or bilingual contexts. |
| L44-E125 | Syntactic structures and property definitions in code, specifically UI and mobile development. | L44-E126 | Predicts geographical proper nouns following cardinal and ordinal direction prefixes. |
| L44-E127 | Technical command-line syntax, hexadecimal constants, and XML/Git metadata formatting. | | |

