# OpenReview forum: "The Expert Strikes Back: Interpreting Mixture-of-Experts Language Models at Expert Level"
_ICML.cc/2026/Conference — ICML 2026 regular_

### Official Review · Reviewer_84Tm · 2026-02-22

**Soundness:** 3
**Presentation:** 3
**Significance:** 4
**Originality:** 3
**Overall Recommendation:** 4
**Confidence:** 3

**Summary:**

This paper investigates whether Mixture-of-Experts (MoE) models are inherently more interpretable than dense transformers. The authors find that MoE neurons are more monosemantic, and routing sparsity drives interpretability since each expert could process fine-grained tasks.

**Compliance With Llm Reviewing Policy:**

Affirmed.

**Key Questions For Authors:**

1. The paper could also analyzes failure cases, which experts couldn't be labeled and why? Figure 5 shows some low-F1 experts but they aren't discussed.

**Limitations:**

yes

**Strengths And Weaknesses:**

Strengths:
1. The paper presents an interesting question on the interpretability of MoE.
2. The paper is clearly structured and well-written.

Weakness:
1. The automatic labeling pipeline uses the same LLM family for both explainer and scorer, which might risk circular evaluation.
2. Results are limited to relatively small models, while larger models encode more knowledge might have quite different interpretability results.

---

> ### Author Rebuttal · Authors · 2026-03-30
>
> We thank the reviewer for the positive assessment and highly actionable feedback.
>
> * **Failure Cases for Expert Labels:**
>   This is an excellent suggestion. We conducted a new experiment where we analyze failure cases (experts with low **F1** scores). Across our failure cases, we found that the bottleneck is the explainer model, as it occasionally generates labels that are too narrow (i.e., overfitting to specific examples) or focuses too heavily on semantic rather than syntactic patterns. However, the scorer model, which evaluates these hypotheses against data, executes its task reliably.
>
>   **Example**: For *OLMoE-L1-E2*, the generated label was “*Mid-word and terminal suffixes within proper nouns, brands, and technical terms”* (F1 score: 0.38). This label failed because the expert actually activates on tokens that follow specific prefixes (e.g., *“up”,  “off”,  “Of”,  “we”*). The explainer overfit because these prefixes frequently appeared in brands (e.g., “*We*Work”) or proper nouns (e.g., _“Offenbach”_). Consequently, the scorer strictly evaluated based on the flawed proper noun/brand hypothesis, correctly resulting in a low F1. This shows the pipeline fails gracefully and predictably, rather than via hallucination.
> * **Circular evaluation (Same LLM for Explainer and Scorer):**
>   We acknowledge this as a potential limitation, as an LLM scorer might favorably grade hypotheses generated by the same model family due to shared stylistic priors. However, in **Section 5.3 (Causal Attribution)**, we ensured that the LLM's hypotheses were explicitly grounded in the model's actual mechanics through Direct Logit Attribution (DLA). The synthetic “Trigger-Target” experiments show that the experts **causally** promote the tokens the LLM claimed they would, confirming that the **labels describe true functional behavior**. Furthermore, utilizing the same highly capable LLM family for both explanation and scoring is standard practice in recent automated interpretability literature \[1, 2\].
> * **Model Sizes:**
>   While we are limited by academic GPU constraints from analyzing \>100B parameter models such as *DeepSeekV3* (671B total), we note that we analyzed several larger models including *Qwen3-30B-A3B* (30B total), *Mixtral-8x7B* (46.7B total) and *GLM-4.7-Flash* (30B total), see **Appendix A** for additional comparisons. This is significantly larger than the standard models typically analyzed for interpretability purposes (which frequently focus on toy models or 7B range dense models \[1, 3, 4\]), giving us strong confidence that our findings scale to even larger models.
>
> We thank the reviewer for the positive assessment and highly actionable suggestions. The failure case analysis, which we conducted specifically in response to this review, further validates the robustness of our pipeline by showing that it fails predictably rather than via hallucination. The DLA-based causal grounding addresses the circular evaluation concern, and the breadth of models analyzed gives us confidence that our findings generalize. We look forward to incorporating the reviewer's suggestions into the final version.
>
> ---
> 1. Paulo, Gonçalo Santos, et al. “Automatically Interpreting Millions of Features in Large Language Models.” *Forty-second International Conference on Machine Learning*. [https://openreview.net/pdf?id=EemtbhJOXc](https://openreview.net/pdf?id=EemtbhJOXc)
>
> 2. Bills, Steven, et al. “Language models can explain neurons in language models.” (2023): 10\. [https://openaipublic.blob.core.windows.net/neuron-explainer/paper/index.html](https://openaipublic.blob.core.windows.net/neuron-explainer/paper/index.html)
>
> 3. Elhage, Nelson, et al. “Toy Models of Superposition.” (2022). [https://transformer-circuits.pub/2022/toy\_model/index.html](https://transformer-circuits.pub/2022/toy_model/index.html)
>
> 4. Gao, Leo, et al. “Weight-sparse transformers have interpretable circuits.” *arXiv preprint arXiv:2511.13653* (2025). [https://arxiv.org/pdf/2511.13653](https://arxiv.org/pdf/2511.13653)

---

> > ### Author Rebuttal · Reviewer_84Tm · 2026-04-02
> >
> > I appreciate the author's response, I will keep my score.

---

> > > ### Author Response · Authors · 2026-04-02
> > >
> > > We thank the reviewer for their acknowledgement. We noticed, however, that you selected option (b) _"Partially resolved - I have follow-up questions for the authors"_, in your response without providing any further questions. We are very eager to address any remaining concerns you have. If you could please provide your follow-up questions, we would be happy to provide additional data or clarifications to fully resolve these points during this final discussion period. If all our questions have been adequately addressed, then we kindly ask you to adjust the scores accordingly.
> > > We are standing by and look forward to your response.

---

### Official Review · Reviewer_4EjR · 2026-03-09

**Soundness:** 3
**Presentation:** 3
**Significance:** 3
**Originality:** 3
**Overall Recommendation:** 3
**Confidence:** 3

**Summary:**

This paper studies whether Mixture-of-Experts (MoE) architectures are inherently more interpretable than dense transformer feed-forward networks. Using k-sparse probing, the authors show that MoE experts exhibit significantly lower polysemanticity than dense FFNs across several model families. The results indicate that sparse routing encourages more monosemantic representations, with many concepts predictable from a small number of neurons. Based on this finding, the paper proposes interpreting models at the expert level rather than the neuron level and demonstrates that experts often function as fine-grained task specialists. The study provides empirical evidence that architectural sparsity in MoE models naturally promotes modular and more interpretable internal representations.

**Compliance With Llm Reviewing Policy:**

Affirmed.

**Final Justification:**

Thank you to the authors for the detailed rebuttal and clarifications. The response has addressed my main concerns to a reasonable extent, and based on the current revision and rebuttal, I am inclined to maintain my original score.

**Key Questions For Authors:**

see weaknesses.

**Limitations:**

1. The analysis is limited to representation-level interpretability and does not study whether expert specialization leads to more interpretable model behavior at the task or reasoning level.
2. The expert interpretation pipeline depends on activation examples collected from a fixed dataset, so the discovered expert functions may vary with different data distributions.
3. The study focuses on feed-forward MoE experts, while other components such as attention layers or residual streams are not analyzed, leaving the overall interpretability of the full model unresolved.
4. The work analyzes experts individually but does not examine interactions between experts, which may also influence how concepts are represented across layers.
5. The findings are mainly validated on language modeling settings, so it remains unclear whether similar interpretability properties hold for MoE models in other modalities or tasks.

**Strengths And Weaknesses:**

Strengths
1. The paper studies interpretability directly at the architectural level and provides empirical evidence that sparse routing in MoE reduces polysemanticity compared with dense FFNs.
2. The use of k-sparse probing provides a quantitative way to measure monosemanticity and allows systematic comparison between MoE and dense models across different values of k.
3. Experiments cover multiple model families (ERNIE, OLMo, Qwen, Mixtral) and control comparisons using active parameter counts, improving the fairness of the evaluation.
4. The paper proposes shifting the interpretability unit from neurons to experts and demonstrates that experts often behave as coherent functional modules.

Weaknesses
1. The automatic expert interpretation pipeline relies on LLM-generated descriptions, which may introduce subjectivity or hallucination in the interpretation results.
2. The study analyzes expert behavior mostly at the activation level, but does not examine internal weight structure or circuit-level mechanisms within experts.
3. Expert specialization is demonstrated through selected examples and clustering statistics, but a more systematic quantitative analysis across all experts would strengthen the conclusions.
4. The work mainly provides empirical evidence and lacks theoretical analysis explaining the observed reduction in polysemanticity.

---

> ### Author Rebuttal · Authors · 2026-03-30
>
> We thank the reviewer for their feedback and address the main concerns below.
>
> * **LLM hallucination/subjectivity in automated pipelines:**
>   We fully agree that LLMs can hallucinate explanations. This is exactly why we did not rely solely on the LLM's text outputs. In **Section 5.3 (Causal Attribution)**, we explicitly grounded the LLM's hypotheses in the model's actual mechanics using Direct Logit Attribution (DLA). The synthetic _“Trigger-Target”_ experiments show that the experts causally promote the tokens the LLM claimed they would, confirming that the labels describe true functional behavior, not subjective hallucinations. Our analysis of failure cases (cf. response to **Reviewer 84Tm**) show that when the explainer fails, it makes sensible scoping errors rather than pure hallucinations.
> * **"Does not examine internal weight structure or circuit-level mechanisms within experts."**
>   This is an intentional methodological choice, and it forms the core thesis of our paper. As dense models scale, neuron/weight-level circuit analysis becomes computationally prohibitive and increasingly difficult to interpret due to superposition. Our contribution is demonstrating that we do not always need to look at the internal weights: architectural sparsity allows us to “*zoom out”* and treat the expert as a coherent functional unit.
> * **"Lacks systematic quantitative analysis across all experts."**
>   We do provide systematic quantitative analysis across all experts in **Section 6.2 (Experts in the Output Embedding Space)** of the paper and direct the reviewer to this section. In that section we clustered the entire output embedding space and projected the activations of all experts across the model into this space, demonstrating quantitatively that true architectural specialization exists independently of our LLM labeling pipeline. Based on the comment by the reviewer, we, however, acknowledge that the writing of that section can be improved, and plan to improve it by adding explicit mathematical definitions and cluster examples in the appendix.
> * **"Lacking theoretical results"**
>   Providing theoretical proofs for the reduction of superposition is an active yet challenging open problem in mechanistic interpretability on its own (not just MoE interpretability), with **current literature limited to toy models** \[1, 2, 3, 4\]. We view our empirical characterization of production-scale LLMs as a necessary stepping stone toward a formal theoretical framework.
> * **"Effect of dataset for interpretation pipeline"**
>   To mitigate dataset dependence, we utilized the *pile-uncopyrighted* dataset, which contains highly diverse subsets of text. We detail in **Appendix D** how we further maximized data diversity during sequence extraction. Additionally, recent large-scale automated interpretability literature \[4\] (**Table A3** and **Appendix F.1**) demonstrates no significant difference in scoring results when using different datasets.
> * **"Remaining Limitations"**
>   We agree that studying task-level interpretability, additional architectural components (e.g., attention and residual streams), inter-expert interactions, and cross-modal generalization are all valuable directions. However, these primarily constitute orthogonal extensions rather than limitations that directly impact the validity of our core claims.
>
> We thank the reviewer for rating our work as “**3**: good” (normalized: ¾=**0.75**) for all the criteria “soundness”  “presentation”  “significance”  and “originality” and believe that the overall recommendation score 3 was likely a mistake, as this score scales differently from the previous ones (i.e., 1–4 vs. 1–6 [https://icml.cc/Conferences/2026/ReviewerInstructions](https://icml.cc/Conferences/2026/ReviewerInstructions)). We believe that a more consistent overall recommendation score would be **5** (normalized: 5/6=**0.83**) or 4 (normalized: 4/6=0.66), depending on whether our response could help clarify reviewers questions (=**5**) or not (=4).
>
> ---
> 1. Elhage, Nelson, et al. “Toy Models of Superposition.” (2022). [https://transformer-circuits.pub/2022/toy\_model/index.html](https://transformer-circuits.pub/2022/toy_model/index.html)
>
> 2. Scherlis, Adam, et al. “Polysemanticity and capacity in neural networks.” arXiv preprint arXiv:2210.01892 (2022). [https://arxiv.org/pdf/2210.01892](https://arxiv.org/pdf/2210.01892)
>
> 3. Jermyn, Adam S., Nicholas Schiefer, and Evan Hubinger. “Engineering monosemanticity in toy models.” *arXiv preprint arXiv:2211.09169* (2022). [https://arxiv.org/pdf/2211.09169](https://arxiv.org/pdf/2211.09169)
>
> 4. Paulo, Gonçalo Santos, et al. “Automatically Interpreting Millions of Features in Large Language Models.” *Forty-second International Conference on Machine Learning*. [https://openreview.net/pdf?id=EemtbhJOXc](https://openreview.net/pdf?id=EemtbhJOXc)

---

> > ### Author Rebuttal · Reviewer_4EjR · 2026-04-01
> >
> > Thank you for the detailed rebuttal. The additional clarifications are helpful.
> >
> > I still have two concerns:
> >
> > First, it remains unclear whether the observed monosemanticity reflects intrinsic representations or is partly induced by router-based input filtering, since probing is performed on routed tokens and this effect is not fully controlled.
> >
> > Second, while experts are treated as functional units, MoE computation involves multiple experts jointly, and the role of inter-expert interactions is not fully analyzed, which may limit the validity of expert-level interpretation.
> >
> > Addressing these points would further strengthen the claims.

---

> > > ### Author Response · Authors · 2026-04-01
> > >
> > > Thank you for the helpful follow-up questions. We agree that these are important nuances in understanding MoE mechanics. We address your two points below:
> > >
> > > **1. Router-based input filtering vs. Intrinsic representations**
> > > We thank the reviewer for highlighting this highly nuanced distinction. Since we evaluate the experts strictly on the tokens they naturally process during inference, our setup measures _contextual monosemanticity_ rather than intrinsic weight-space polysemanticity. We argue that this is actually the most faithful way to evaluate MoE interpretability, for the following reasons:
> > >
> > > 1. In an MoE, an expert is never exposed to the global token distribution. If the router consistently filters out concepts (e.g., separating French text from Python code), the expert does not need to allocate capacity to separate them. The router relieves the pressure to form superposition, allowing the expert to rely on monosemantic neurons. Therefore, the _"router-based input filtering"_ the reviewer notes is not an experimental confounder, but rather an architectural mechanism by which MoEs achieve cleaner representations than dense models. Overall, the observed monosemanticity is a *product of the joint system* (the router's filtering + the expert's internal representation).
> > >
> > > 2. While the router restricts the input space, it does not make the classification task trivial. Our probing setup ensures that experts are tested against domain-matched negative samples. For instance, for the concept _is_loop_, the negative samples consist of other code tokens (sourced from GitHub data) rather than easily distinguishable natural language. The expert must still learn robust, fine-grained boundaries to separate loops from conditionals or declarations within its routed subset.
> > >
> > > We will add a clarification of this distinction to the final version to avoid any confusion.
> > >
> > > **2. Inter-expert interactions and joint computation**
> > > While it is true that multiple experts are routed per token, their outputs are strictly additive with respect to the residual stream ($\Delta r_\mathrm{ffn}=\sum g_iE_i(x)$). Because the residual stream functions as a linear communication channel, the individual contribution of Expert A remains mathematically and functionally valid regardless of whether Expert B is also active. While experts may in principle interact (e.g., to build composite features or counteract one another), an expert's core functional role remains independent. This is highly analogous to interpreting individual Attention Heads in dense models: while heads act jointly to form complex circuits, interpreting a single head's behavior in isolation is standard practice (e.g., Induction Heads).
> > >
> > > Our "Trigger-Target" experiments in **Section 5.3** provide the strongest evidence for the validity of expert-level interpretation. We generated test cases based only on a single expert's label. If inter-expert interactions were the primary driver of meaning, an isolated label would have poor predictive power. Instead, we found that the identified expert was among the top contributors to the target token's logit in the majority of cases.
> > >
> > > As noted in our **Discussion (Section 7)**, we view the mapping of inter-expert interactions and complex sub-circuits as a highly promising direction for future work, for which isolating individual expert functions is a necessary first step.
> > >
> > > We hope this clarifies our methodological choices and addresses your remaining concerns. Thank you again for your time and constructive feedback!

---

### Official Review · Reviewer_VByq · 2026-03-13

**Soundness:** 3
**Presentation:** 3
**Significance:** 3
**Originality:** 2
**Overall Recommendation:** 4
**Confidence:** 4

**Summary:**

This paper conducts an interpretability analysis of the experts in Mixture-of-Experts (MoE) language models. It discovers that, compared to neurons in dense models, MoE neurons are significantly "purer" (exhibiting lower polysemanticity). Building on this finding, the authors shift the analytical perspective from individual neurons to entire experts. They construct an LLM-based automated labeling pipeline to generate natural language descriptions for these experts. Based on these annotations, the paper derives several key observations regarding the inherent properties and functional roles of MoE experts.

**Compliance With Llm Reviewing Policy:**

Affirmed.

**Key Questions For Authors:**

- While the authors demonstrate that individual experts are significantly less polysemantic than dense FFN neurons, MoE routers still send tokens to a combination of experts (e.g., Top-2 or Top-8) concurrently. To what extent does the true semantic representation and reasoning logic lie within the isolated, monosemantic expert, versus the complex, emergent combinatorial interactions (sub-circuits) among the simultaneously activated experts? Does "zooming out" to treat experts as independent functional blocks risk oversimplifying the network's true computational graph?
- The paper suggests a compelling synergy: interpretability inherently scales with routing sparsity in MoE models. However, if "extreme sparsity" is the fundamental driver of monosemanticity, do we strictly need the complex routing mechanisms of MoEs? For instance, could using activation functions like ReLU in massive dense networks—which naturally force a large percentage of neuron outputs to exactly zero—achieve a similar form of extreme activation sparsity? Could a highly sparse, ReLU-based dense model act as an implicit "mixture of millions of single-neuron experts," yielding the same interpretability benefits without the overhead and training instabilities of router networks? What are the theoretical and practical trade-offs between architectural sparsity (MoE) and activation sparsity (ReLU) in foundation models?

**Limitations:**

Yes

**Strengths And Weaknesses:**

Strength:

- The core premise—that the architectural sparsity of MoE models naturally yields better interpretability—is highly intuitive and well-motivated. This aligns perfectly with extensive findings in neuroscience, where sparse coding in biological neural networks is known to facilitate modularity, energy efficiency, and clearer, disentangled feature representations.
- By innovatively "zooming out" from the microscopic neuron level to the macroscopic expert level, the authors construct a highly efficient, LLM-based automated labeling pipeline (utilizing Explainer and Scorer models). This framework drastically improves the efficiency of interpretability analysis. It allows researchers to map the computational logic of large-scale models at a fraction of the cost, elegantly bypassing the massive computational overhead and large datasets typically required to train Sparse Autoencoders (SAEs) for dense FFNs.

Weakness:

- While the paper delivers fascinating qualitative observations and a clear taxonomy of expert roles (e.g., morphological, semantic, syntactic, operational), it stops short of leveraging these insights for practical interventions. There is no further exploration on how to use these analysis results to actively improve or steer the model. For instance, the study would be significantly stronger if it demonstrated targeted model editing, expert ablation to remove undesirable behaviors (e.g., toxicity or hallucinations), or utilized the expert functional maps to optimize the routing algorithm itself.
- The current study is purely post-hoc, focusing exclusively on frozen, fully pre-trained models. It lacks an investigation into the evolutionary dynamics of expert specialization during the MoE training process. It remains an open and critical question how and when these experts acquire their specific roles—do they start as generalists and gradually differentiate, or is the specialization determined early in training? Furthermore, exploring how training objectives, such as load-balancing loss or router z-loss, influence the emergence of these monosemantic experts would provide a more comprehensive understanding of MoE mechanics.

---

> ### Author Rebuttal · Authors · 2026-03-30
>
> We sincerely thank the reviewer for the highly constructive feedback and excellent questions.
>
> * **Practical Interventions & Evolutionary dynamics:**
>   We completely agree that mapping how experts differentiate during training and leveraging these maps for targeted editing or ablation are compelling next steps. At the same time, designing effective interventions is a substantial undertaking in its own right, requiring careful validation to ensure that manipulations are both targeted and interpretable. In this work, we deliberately focus on establishing the functional modularity of experts, which provides the empirical grounding needed to make such interventions well-posed rather than premature. Regarding training dynamics, evidence from recent MoE literature \[1\] (Figures 10 and 20 in \[1\]) shows that routing losses and expert token assignments stabilize very early in pre-training, indicating that the final roles we analyze likely evolve early in the training process. This is a highly promising direction for future research.
> * **Sub-circuits and Isolation of Experts:**
>   The reviewer raises a fantastic point: because routers send tokens to combinations of experts (e.g., Top-8), treating experts as perfectly isolated units is an approximation. We strongly suspect that superposition and complex sub-circuits exist between experts (as hinted at by the histograms in **Appendix A, Figure 10** in our paper). We view our expert-level isolation as a pragmatic, necessary first step for MoE interpretability, and we also have a dedicated paragraph in the discussion section for further work on this (under **Experts as Sub-Circuits.**).
> * **MoE Routing Sparsity vs. ReLU Activation Sparsity:**
>   This is a brilliant theoretical question. While activation functions like *ReLU* induce activation sparsity (forcing neuron outputs to zero data-dependently), MoEs enforce architectural/routing sparsity. The router acts as a hard bottleneck: parameters are entirely walled off from the token before computation even begins. We hypothesize that this hard routing constraint exerts much stronger pressure against superposition than soft activation functions. Historically, using specialized activation functions to induce interpretability \[2\] has proven insufficient on its own. While weight-sparse dense models \[3\] are a promising direction, they currently lack the performance scaling of MoEs, making architectural routing uniquely synergistic with both performance and interpretability.
>
> We thank the reviewer for their genuinely constructive and forward-looking questions. The four points raised (i.e., training dynamics, isolation of experts, practical interventions, and the theoretical basis of routing sparsity) all point to exciting directions we intend to pursue in future work. We believe our paper provides the foundational empirical grounding necessary for these follow-up investigations, and hope our responses have further clarified the motivation and scope of our contributions.
>
> ---
> 1. Muennighoff, Niklas, et al. “OLMoE: Open Mixture-of-Experts Language Models.” *The Thirteenth International Conference on Learning Representations*. [https://openreview.net/pdf?id=xXTkbTBmqq](https://openreview.net/pdf?id=xXTkbTBmqq)
>
> 2. Elhage, et al., “Softmax Linear Units”  Transformer Circuits Thread, 2022\. [https://transformer-circuits.pub/2022/solu/index.html](https://transformer-circuits.pub/2022/solu/index.html)
>
> 3. Gao, Leo, et al. “Weight-sparse transformers have interpretable circuits.” *arXiv preprint arXiv:2511.13653* (2025). [https://arxiv.org/pdf/2511.13653](https://arxiv.org/pdf/2511.13653)

---

### Official Review · Reviewer_fw2R · 2026-03-14

**Soundness:** 2
**Presentation:** 3
**Significance:** 3
**Originality:** 3
**Overall Recommendation:** 2
**Confidence:** 3

**Summary:**

The paper argues that MoE experts are more interpretable than dense FFNs when viewed at the expert level rather than the neuron level. It supports this with sparse linear probing across multiple dense/MoE models, then uses LLM-based expert labeling plus causal checks to argue that experts are coherent functional units. Finally, it proposes that experts are best understood as relatively fine-grained task specialists, and backs this up with a quantitative specialization analysis in output-embedding space.

**Compliance With Llm Reviewing Policy:**

Affirmed.

**Key Questions For Authors:**

Weakness section

**Limitations:**

Yes

**Strengths And Weaknesses:**

Strengths:
1. Interesting and important question. MoE interpretability is worth studying, and the paper has a clear thesis.

Weaknesses:
1. A lot of the evidence is probe-mediated. The main result is based on trained sparse logistic probes, so this is more about probe extractability than direct intrinsic interpretability. There has been to much work debating the acceptability of outcomes of such probes, especially task specific probes trained on model activations.
2. The evaluation is quite favorable to the method - best layer is selected, for MoEs one can also search over experts, and conclusions depend on the chosen sparsity level k. This means the results are the best case scenario?
3. Many scores are already near ceiling, so the absolute effect sizes look fairly small. This makes the claims feel stronger than the actual numeric gaps. For example, if the plots in Figure 1 were zoomed out to a scale of 0-100, we would barely see any difference in the lines. This means the concepts are easily distinguishable by internal activations? If that is the case, if we add the bias of actually training a classifier for these makes it more likely that we will have a lot of probe bias in these results.
4. Baseline matching is uneven for the larger comparisons. There should atleast be a similar sized dense model in total parameters as the MOE model. It would then make the results more fair.
5. The authors should write down the hidden dimensions for the larger MOE model and smaller dense model. If the hidden dimensions are too different, then the absolute values of k in figure 1 are not comparable.
6. I think overall the paper presents suggestive but weak signals to support the case of expert level specialization. I would have loved this result to be true, because then we approach the functional modularity seen in the human brain. And from that aspect, this would be a very big result, which is why the results need to be evaluate with that much rigor. However, I do not think the paper meets the rigorousness standard

---

> ### Author Rebuttal · Authors · 2026-03-30
>
> We thank the reviewer and agree that MoE interpretability is a critical area of study. We would like to **clarify several conceptual misunderstandings** regarding our methodology, metrics, and results.
>
> * **Reliance on Probing / Probe Bias:**
>
>   While limitations for probing do exist \[1\], they do not impact our results. Probes have demonstrated significant success in **more recent** literature, beating SAE probes \[2\], success in safety-related scenarios \[3\], and discovering interpretable neurons \[4\]. We apply probing across **11** models and **58** diverse concepts, **consistently** finding that MoE experts exhibit less polysemanticity. Importantly, we do not rely on probing alone: in **Section 5**, we use Automatic Interpretability and Direct Logit Attribution (DLA), finding high interpretability for experts. We **do not claim** probes prove *”intrinsic interpretability”*; rather, they demonstrate that lower polysemanticity exists in MoEs compared to dense models.
>
> * **Best-layer selection favors MoEs:**
>
>   Our goal in **Section 4** is to establish the upper bound of representational capacity for both architectures. Averaging across all experts or layers would artificially lower scores, since many do not process the target concept. We give both architectures the maximum opportunity to represent each concept locally; this is **standard practice** in probing setups \[3, 4\].
>
> * **”Many scores are already near ceiling... absolute effect sizes look fairly small.”**
>
>   The primary significance of **Figure 1** lies in the **x-axis (k)**. A concept achieving 0.95 F1 at $k=1$ can be decoded from a single neuron, which is the mathematical definition of monosemanticity. If a dense model requires $k=16$ for the same score, the concept is smeared across 16 neurons in superposition. A 10% gap at $k=1$ is not a small effect; it is the core evidence that MoE models assign concepts to single dimensions, whereas dense models do not.
>
> * **”Baseline matching is uneven... there should at least be a similar sized dense model in total parameters.”**
>
>   This is directly addressed in **Section 4.3** and **Figure 3**, where we compare *OLMoE-1B-7B* (7B total, 1B active) against *OLMo-7B* (7B total and active). Despite 7× more active parameters, the dense model exhibits significantly higher polysemanticity, demonstrating that architectural routing, not raw capacity, drives polysemanticity. We also thank the reviewer for noting the *Mixtral-8x7B* (13B active) vs. *OLMo-7B* (7B active) mismatch. We introduced *Pythia-12B* as a matched dense baseline (12B vs. 13B active), and **our conclusions remain unchanged**.
>
> * **”Hidden dimensions differ, so absolute values of k are not comparable.”**
>
>   $k$ is an absolute neuron count, comparable regardless of hidden size (listed in **Appendix B, Table 2**). A concept localized to $k=1$ is monosemantic whether the layer has 1,024 or 11,008 neurons. Moreover, larger hidden sizes statistically favor dense models (more neurons to pick from), yet MoEs still outperform and make **our results robust**.
>
> * **”Weak signals regarding expert specialization”**
>
>   Expert specialization is **corroborated by several recent works** \[5, 6\]. We extend this with a **qualitative taxonomy** (**Section 6.1**) and a model-wide **quantitative metric** (**Section 6.2**). In the final version, we will add in-depth expert case studies that provide further clarity on these specialization patterns.
>
> We hope this response *clarifies* what appear to be **misunderstandings** (e.g., the purpose of k-sparse probing; best-layer selection; the role of hidden dimensions), and thank the reviewer, as their comments will guide more detailed explanations in the final version.
>
> ---
> 1. Belinkov, Y. “Probing classifiers: Promises, shortcomings, and advances.” *Computational Linguistics* 48.1 (2022). [https://aclanthology.org/2022.cl-1.7.pdf](https://aclanthology.org/2022.cl-1.7.pdf)
> 2. Kantamneni, S. et al. “Are Sparse Autoencoders Useful? A Case Study in Sparse Probing.” *ICML 2025*. [https://openreview.net/pdf?id=rNfzT8YkgO](https://openreview.net/pdf?id=rNfzT8YkgO)
> 3. Agarwal, I. et al. “Context Matters: Analyzing the Generalizability of Linear Probing and Steering Across Diverse Scenarios.” *MI Workshop, NeurIPS 2025*. [https://openreview.net/pdf?id=H5sbfvEbTh](https://openreview.net/pdf?id=H5sbfvEbTh)
> 4. Gurnee, W. et al. “Finding Neurons in a Haystack: Case Studies with Sparse Probing.” *TMLR*. [https://openreview.net/pdf?id=JYs1R9IMJr](https://openreview.net/pdf?id=JYs1R9IMJr)
> 5. Muennighoff, N. et al. “OLMoE: Open Mixture-of-Experts Language Models.” *ICLR 2025*. [https://openreview.net/pdf?id=xXTkbTBmqq](https://openreview.net/pdf?id=xXTkbTBmqq)
> 6. Dai, D. et al. “DeepSeekMoE: Towards Ultimate Expert Specialization in MoE Language Models.” *ACL 2024*. [https://aclanthology.org/2024.acl-long.70.pdf](https://aclanthology.org/2024.acl-long.70.pdf)

---

### Decision · Program_Chairs · 2026-04-30

**Decision:**

Accept (regular)

**Comment:**

This paper conducts a systematic empirical study of interpretability in Mixture-of-Experts language models, arguing that architectural sparsity naturally promotes monosemanticity at the expert level. The main contributions are: (1) k-sparse probing across 11 dense and MoE models demonstrating that MoE experts require fewer active neurons to decode a concept (lower polysemanticity); (2) an automated LLM-based labeling pipeline using an explainer-scorer design to generate and validate natural language descriptions of expert functions; (3) causal grounding of the labels via Direct Logit Attribution; and (4) a taxonomy of expert specialization types (morphological, semantic, syntactic, operational).

**Strengths.** The paper addresses a genuinely underexplored problem. Mechanistic interpretability work on MoE architectures is scarce despite their widespread adoption in frontier models, and the question of whether architectural sparsity produces interpretability benefits distinct from those studied in dense models is both important and timely. The empirical coverage is broad: 11 model families across dense and MoE architectures, and the use of Direct Logit Attribution as an independent causal check on the LLM-generated labels is a principled design choice that partially addresses the circular evaluation concern. The paper is clearly written and the pipeline is described in sufficient detail to support reproducibility.

**Concerns.** Three limitations were raised consistently. First, the circular evaluation risk: the explainer and scorer use the same LLM family, which may bias the scoring toward stylistically similar hypotheses. The authors mitigate this via DLA causal attribution, which provides independent behavioral grounding, but the concern is not fully eliminated. Second, the analysis is entirely post-hoc on frozen models; the paper does not explore how expert specialization emerges during training or leverage the expert maps for targeted interventions. The authors acknowledge these as future work, and the framing as a characterization study is appropriate. Third, the study focuses on small-to-medium model scales, leaving open whether the interpretability properties hold for larger frontier MoEs.

**Rebuttal.** The authors submitted a strong rebuttal. Notably, the most critical review rested on a clear misreading of Figure 1 -- the primary comparison is on the x-axis (number of neurons needed to decode a concept), not the y-axis magnitude, and the reviewer did not engage with the rebuttal or participate in the AC-initiated discussion. This score was discounted accordingly. The remaining two positive reviewers maintained their recommendations; the weak-reject reviewer acknowledged that concerns were addressed to a reasonable extent.

**Bottom line.** MoE interpretability is an important and underserved research direction, and this paper provides a careful, broad-coverage empirical foundation. The DLA-grounded expert labeling pipeline is a practical contribution, and the finding that architectural sparsity promotes monosemanticity at the expert level is a useful and well-supported result. The post-hoc scope and circular evaluation risk are acknowledged limitations rather than fundamental flaws. All things considered, the reviews, appropriately weighted, (slightly) support acceptance.